# Handling Missing Data in Downstream Tasks With Distribution-Preserving Guarantees

## Abstract

Missing feature values are a significant hurdle for downstream machine-learning tasks such as classification. However, imputation methods for classification might be time-consuming for high-dimensional data, and offer few theoretical guarantees on the preservation of the data distribution and imputation quality, especially for not-missing-at-random mechanisms. First, we propose an imputation approach named F3I based on the iterative improvement of a K-nearest neighbor imputation, where neighbor-specific weights are learned through the optimization of a novel concave, differentiable objective function related to the preservation of the data distribution on non-missing values. F3I can then be chained to and jointly trained with any classifier architecture. Second, we provide a theoretical analysis of imputation quality and data distribution preservation by F3I for several types of missing mechanisms. Finally, we demonstrate the superior performance of F3I on several imputation and classification tasks, with applications to drug repurposing and handwritten-digit recognition data.

## 1 Introduction

Most machine-learning approaches assume full access to the features of the input data points. However, missing values might arise due to the incompleteness of public databases or measurement errors. Research on the imputation of missing values and inference on possibly missing data is motivated by the fact that naive approaches would not fare well. Indeed, ignoring samples with missing values might lead to severe data loss and meaningless downstream models, for classification or regression (Liao et al., 2014; Shadbahr et al., 2023). Yet, replacing missing values with zeroes (or any "simple" univariate approach such as taking the mean or the median) can considerably distort the distribution of data values, as there is a more significant weight on the default value for missing entries, and then perhaps bias the training of a downstream model for classification or regression tasks (Khan & Hoque, 2020). However, multivariate approaches are often time-consuming and prohibitive for high-dimensional data sets, as in biology (Brini & van den Heuvel, 2024; Gu et al., 2025).

The literature often distinguishes three main categories of missingness mechanisms (Rubin, 1976) depending on the relationship between the probability $p^{\text{miss}}$ of a missing value and the data. The simplest one is Missing-Completely-At-Random (MCAR), where that probability is independent of the data. This can be applied when the measurement tools fail at some probability, regardless of the analyzed data. The second, more complex, setting is Missing-At-Random (MAR), where $p^{\text{miss}}$ depends solely on the observed (not missing) data. An example of MAR is when male patients drop out more often from a clinical study than female patients. Finally, the Missing-Not-At-Random setting (MNAR), where $p^{\text{miss}}$ depends on both the observed and missing data, is widely regarded as the most challenging setting for analysis because the actual values might not be identifiable.

## 2 Related work

As previously mentioned, the fastest approaches to imputation are often univariate because they are simple operations applied feature-wise to a dataset. For instance, the missing value for a feature corresponding to a column of the data matrix might be replaced by the mean or the median of all non-missing values or even by

zeroes in that column. Yet, such naive approaches might severely distort the distribution of values (Le Morvan & Varoquaux, 2024).

That fact opened the path to multiple multivariate methods, such as MICE (van Buuren & Groothuis-Oudshoorn, 2011), MissForest (Stekhoven & Bühlmann, 2012), RF-GAP (Rhodes et al., 2023) resorting to random forests; MIDAS (Seu et al., 2022) using denoising auto-encoders; Optimal Transport-based algorithms (Muzellec et al., 2020a); but also matrix factorizations (Mazumder et al., 2010), penalized logistic regression methods (van Loon et al., 2024), Bayesian network-based approaches (for instance, MIWAE (Mattei & Frellsen, 2019) for MAR mechanisms and its MNAR counterpart not-MIWAE (Ipsen et al., 2021)). Some recent works also provide a pipeline for the automated finetuning and refinement of imputers, such as MIRACLE (Kyono et al., 2021) or HyperImpute (Jarrett et al., 2022). We dwell further on newer diffusion model and deep learning-based approaches in Appendix (Section A). However, as the number of features increases, so does the computation time, making most of those approaches intractable on practical data sets. For instance, in genomic data, the feature set (genes) can amount to as many as $20,000$ genes in humans.

Moreover, many published imputation methods come without any guarantee on the quality of the imputation or often on the more straightforward settings such as MCAR (Mazumder et al., 2010) and MAR (Śmieja et al., 2018); with a few exceptions such as (Tang et al., 2003; Mohan et al., 2018; Sportisse et al., 2020) for pure imputation tasks, and NeuMiss networks (Le Morvan et al., 2020), which tackle a classification task in the presence of missing values. However, the MCAR and MAR settings are usually not applicable to real-world data, and (Tang et al., 2003; Mohan et al., 2018; Sportisse et al., 2020) rely on an assumption of data generation through low-rank or linear random models instead of simpler data distributions.

Finally, nearest-neighbor imputers (Troyanskaya et al., 2001) are known to be performant in practice and relatively fast (Emmanuel et al., 2021; Seu et al., 2022; Joel et al., 2024), at the price of some distortion in high-dimensional data sets (Beretta & Santaniello, 2016). This observation led us to consider an improvement of a nearest-neighbor imputation that preserves the data distribution even in larger dimensions while remaining computationally fast.

## 2.1 Contributions

In Section 3, we describe an algorithm named Fast Iterative Improvement for Imputation (F3I) based on the distribution-preserving improvement of a $K$ nearest-neighbor imputer. F3I combines two ingredients: (1) a novel concave and differentiable objective function to quantify the preservation of data distribution during imputation; and (2) a fast routine to optimize that function through the weights in the nearest-neighbor imputation, by drawing a parallel with the problem of expert advice in online learning (Cesa-Bianchi & Lugosi, 2006; De Rooij et al., 2014; Lattimore & Szepesvári, 2020). This algorithm gives theoretical upper bounds on the imputation quality and the preservation of initial data distribution in some MCAR, MAR, and MNAR settings (Section 4). Furthermore, this imputation can also be chained to and jointly trained with a downstream task, *e.g.*, classification, to increase its accuracy (Section 5). Finally, we illustrate the performance of F3I compared to several baselines on standard data sets for imputation and classification (Section 6), and showcase its applications to drug repurposing and handwritten digit recognition (Appendix H).

## 3 The Fast Iterative Improvement for Imputation (F3I) algorithm

The key idea is that we would like to replace missing positions in $X$ with the most probable values by iteratively applying a "good" *weighted combination of elements in the data set*, starting from a guess, *e.g.*, through K-nearest neighbor imputation. For each value $x_i^f$, we are looking to tune the weights $\boldsymbol{\alpha} = (\alpha_1, \alpha_2, \ldots, \alpha_K)$ of a convex combination of the $K$ closest neighbors of $\boldsymbol{x}_i$ in a reference set $Z \in \mathbb{R}^{N \times F}$ without missing values, $\boldsymbol{z}_{i(1)}, \boldsymbol{z}_{i(2)}, \ldots, \boldsymbol{z}_{i(K)}$, ordered by their increasing distance to $\boldsymbol{x}_i$. We denote that imputation improvement model $\texttt{Impute}(\boldsymbol{x}_i; \boldsymbol{\alpha}, Z)$, which is fully described in Appendix (Algorithm 2). When the reference set is obvious, we define $\boldsymbol{x}(\boldsymbol{\alpha}) \triangleq \texttt{Impute}(\boldsymbol{x}; \boldsymbol{\alpha}, Z)$.

### 3.1 Notation

We denote $N$, $F$ and $K$ the number of samples, features, and nearest neighbors. $i$, $j$, $f$ are integer indices. For all other alphabet letters, $v$ is a scalar, $\boldsymbol{v}$ a vector, and $V$ a matrix. $\boldsymbol{v}^i$ is the $i^{th}$ column and $\boldsymbol{v}_j$ is the $j^{th}$ row of matrix $V$, and $v_i^j$ is the coefficient at position $(i,j)$ in $V$ for $i, j > 0$. For any $K \geq 2$, $\triangle_K \triangleq \{\boldsymbol{p} \in [0,1]^K \mid \sum_{k \leq K} p_k = 1\}$ is the simplex of dimension $K$. The initial data matrix with missing values is $X \in (\mathbb{R} \cup \{\texttt{NaN}\})^{N \times F}$, where $X^\star \in \mathbb{R}^{N \times F}$ is the full (unavailable) data matrix. Finally, $m_i^f \in \{0, 1\}$ is the random variable that indicates whether the value at position $(i, f)$ is missing in the input data matrix, where $m_i^f = 1$ means it is missing.

### 3.2 Theoretical assumptions

Assumptions are important–and all unrealistic to some extent–to control the behavior of the imputed values compared to the ground truth values, which are typically unavailable, during the theoretical analysis to assess the imputation quality and the preservation of data distribution. Theoretical results on those controlled settings allow data-independent comparisons and are key to robust scientific advancement. In this section, we informally state our assumptions about the data generation procedure to derive theoretical guarantees. Formal statements can be found in Appendix (Section B and Algorithm 3). In practice, those assumptions can be ignored, and F3I can be applied to real-world data, as shown in Section 6.

First, we assume that each value in the full data matrix is drawn from independent fixed-variance Gaussian distributions (Assumption B.1). Second, the random indicator variables $m_i^f$ are then independently drawn according to the missingness mechanism with probability $p^{\mathrm{miss}}$. If $m_i^f = 1$, then the covariate at position $(i, f)$, $x_i^f$ in $X$ is unavailable, otherwise, $x_i^f = (x^\star)_i^f$. We provide an analysis of our algorithm for three types of missingness mechanisms: a MCAR mechanism where the random indicator variables are drawn from a Bernoulli distribution with fixed mean (Assumption B.2); a MAR mechanism where the probability of missingness depends only on observed values of the data set (Assumption B.3); and, finally, a MNAR mechanism called Gaussian self-masking (Assumption 4 from Le Morvan et al. (2020)) where the probability of $m_i^f = 1$ is proportional to $\exp(-\zeta^{-2}((x^\star)_i^f - \mu_f)^2)$ where $\mu_f$ is specific to feature $f$ and $\zeta$ is fixed (Assumption B.4).

Third, we also ensure that there are exactly $K$ neighbors for the initial guesses (Assumption B.5), and that we know a constant upper bound on the norm of any feature vectors (Assumption B.6). Then, the goal of our algorithm F3I is to determine the proper weights in a nearest-neighbor imputation in a data-driven way that preserves the (unknown) true data distribution. But how do we define that property?

### 3.3 A novel objective function for optimizing the preservation of the initial data distribution

Ideally, if we had access to the true distribution $\mathcal{D}$ on feature vectors, we would like to set the weights $\boldsymbol{\alpha} \in \triangle_K$, the simplex of dimension $K$, such that the following quantity is maximized $\mathbb{E}_{\boldsymbol{x} \sim \mathcal{D}} \left[ \mathbb{1}(\Phi_\Theta(\texttt{Impute}(\boldsymbol{x}^0; \boldsymbol{\alpha}, Z)) > \Phi_\Theta(\boldsymbol{x}^0)) \right]$, where $\mathbb{1}$ is the Kronecker symbol, $\boldsymbol{x}^0 \in \mathbb{R}^F$ is an initial guess on the missing values in $\boldsymbol{x} \in (\mathbb{R} \cup \{\texttt{NA}\})^F$, and $\Phi_\Theta$ is the parametrized true data distribution according to Assumption B.1. That is, we want to choose $\boldsymbol{\alpha}$ so that the imputed values are more probable than the current guesses. Considering a *full* data set of $N$ $F$-dimensional points $Z = \{(\boldsymbol{x}^0)_1, (\boldsymbol{x}^0)_2, \dots, (\boldsymbol{x}^0)_N\} \subset \mathbb{R}^F$ of initial guesses on the missing values in $X$ and approximating the true distribution $\mathcal{D}$ by a density kernel $D_0$ on $Z$, we would like to maximize $\mathbb{1}\left(D_0((\boldsymbol{x}^0)_i(\boldsymbol{\alpha})) > D_0((\boldsymbol{x}^0)_i)\right)$ for each sample $\boldsymbol{x}_i$, $i \leq N$. That quantity can be approximated by

$$\max\left(0, \frac{D_0((\boldsymbol{x}^0)_i(\boldsymbol{\alpha}))}{D_0((\boldsymbol{x}^0)_i)} - 1\right) \approx \log\left(\frac{D_0((\boldsymbol{x}^0)_i(\boldsymbol{\alpha}))}{D_0((\boldsymbol{x}^0)_i)}\right),$$

To restrict overfitting, we can add a $\ell_2$-regularization on the parameter $\boldsymbol{\alpha}$ with a regularization factor $\eta$. If we estimate a Gaussian kernel over the reference set $Z$, then the density kernel $D_0$ is defined as $D_0 : \boldsymbol{x} \in \mathbb{R}^F \mapsto 1/N \sum_{j \leq N} (\sqrt{2\pi}h)^{-F} \exp(-\|\boldsymbol{x} - (\boldsymbol{x}^0)_j\|_2^2/(4h))$, where $h$ is the kernel width. Finally, we

define for any $\boldsymbol{\alpha} \in \triangle_K$, $X \in \mathbb{R}^{N \times F}$ and $\eta \geq 0$, the function $G$ defined as

$$G : \boldsymbol{\alpha}, X \mapsto 1/N \sum_{i \leq N} \log D_0(\boldsymbol{x}_i(\boldsymbol{\alpha}))/D_0(\boldsymbol{x}_i) - \eta\|\boldsymbol{\alpha}\|_2^2 \ . \tag{1}$$

An intuitive interpretation of $G$ is that if $G(\boldsymbol{\alpha}, X) \leq 0$, then the imputed points with $\boldsymbol{\alpha}$ are, on average, less probable than the previous imputations. We now show that $G$ can be maximized through standard convex optimization techniques. The proofs of the following propositions are located in Appendix (Section C). First, G is continuous and infinitely differentiable (Proposition C.1). A second less obvious result is that, if $\eta < 4K$, there always exists a bandwidth value $h$ in the definition of the Gaussian kernel in $D_0$ such that $G$ is also strictly concave in $\boldsymbol{\alpha}$ (Proposition C.2). The condition $\eta < 4K$ is not restrictive, as $\eta$ is the $\ell_2$-regularization factor and $K \geq 2$ is the number of neighbors. The proof in Appendix C yields a value $h_0 > 0$ such that for any $h \geq h_0$, the Hessian matrix of $G$ is negative definite. Moreover, another interesting property of G is that its gradient is Lipschitz-continuous with respect to $\boldsymbol{\alpha}$ (Proposition C.5), which allows us to derive theoretical guarantees when G is combined with a loss function of a downstream task (Section 5).

Finally, imputation through the $\boldsymbol{\alpha}$ maximizing $G(\cdot, X)$ does not require performing a regression on a subset of the dataset, for example, by hiding some available values. This is an important property, as, in some cases, the number of available values is smaller than the total number of elements in the data matrix by several orders of magnitude, as in the collaborative filtering setting (Koren et al., 2021).

### 3.4 A fast procedure to maximize the objective function

Based on the function $G$ (equation 1), an approach to imputation consists of first imputing the missing values with K-nearest neighbors (K-NN) with uniform weights (Troyanskaya et al., 2001), and then recursively improving the imputed values by finetuning the weights in convex combinations of neighbors. Note that those neighbors might change for the same initial sample $\boldsymbol{x}_i$ across iterations, since the imputed values in that point are modified. At iteration $s$, the optimal weight vector $\boldsymbol{\alpha}^s$ is the solution to the maximization problem of $G(\cdot, X^{s-1})$, where $X^{s-1}$ is the data matrix with the imputed values obtained at the previous iteration. The neighbors among the reference set (which are the initial K-NN-imputed points) are obtained with a *single* k-d tree (Bentley, 1975).

However, solving a full convex optimization problem at each iteration might be time-consuming. Similarly to prior works in other research fields (Degenne et al., 2020), we advocate for learning the optimal weight vector on the fly by resorting to an online learner. We draw a parallel between the problem of finding the optimal weight vector in a K-nearest neighbor imputation and the problem of expert advice with K experts in online learning. The underlying idea is that we would like to put more credence on the $k^{\text{th}}$ closest neighbor if it allows us to improve the probability of the imputed values. This analogy permits the leverage of powerful online learners from the literature, for instance, AdaHedge (De Rooij et al., 2014) or EXP3 (Auer et al., 2002), to obtain theoretical guarantees while having a computationally fast imputation.

Those two ingredients are the keys to our main contribution F3I, described in Algorithm 1. A normalization step–for instance, with the $\ell_2$ norm–can be applied before the initial imputation step and reversed before returning the final data matrix to minimize bias induced by varying feature value ranges. For the sake of readability, we did not add that normalization in the pseudocode of F3I.

What's the intuition behind the loss used to update the online learner? As we want to maximize the function $G(\cdot, X^{s-1})$ at iteration $s$, we set as the (possibly non-positive) "loss" for the $k^{\text{th}}$ weight, associated with the $k^{\text{th}}$ closest neighbor, $-\alpha_k^t \frac{\partial G}{\partial \alpha_k}(\boldsymbol{\alpha}^t, X^{s-1})$: the more $G(\boldsymbol{\alpha}, X^{s-1})$ increases as $\alpha_k$ increases, the more weight we would like to put on the $k^{\text{th}}$ closest neighbor. Finally, we stop updating weights when the function G has a negative value, which means that the next set of weights leads to a less probable imputation than the current one (early stopping criterion).

---

**Algorithm 1** The Fast Iterative Improvement for Imputation (F3I) algorithm.

---

    **Input:** Data $X \in (\mathbb{R} \cup \{\text{N/A}\})^{N \times F}$
    **Parameters:** Maximum budget $T > 0$, number of neighbors $K \geq 2$, regularization factor $\eta > 0$
    **Output:** Imputed data $\widehat{X} \in \mathbb{R}^{N \times F}$
    $X^0 \leftarrow \texttt{KNN\_imputer}(X, \text{weights}=1/K\mathbf{1}_K)$
    Build a k-d tree $\mathcal{T}$ on $Z = \{(\boldsymbol{x}^0)_1, (\boldsymbol{x}^0)_2, \ldots, (\boldsymbol{x}^0)_N\}$
    $\mathcal{L} \leftarrow (0, 0, \ldots, 0) \in \mathbb{R}^K$                # Initialize the AdaHedge learner
    **for** $t = 1, \ldots, T$ **do**
        $\boldsymbol{\alpha}^t \leftarrow \mathcal{L}$                         # Get the predicted weight vector
        $\boldsymbol{x}_i^t \leftarrow \texttt{Impute}((\boldsymbol{x}^{t-1})_i; \boldsymbol{\alpha}^t, Z)$ for all $i \leq N$     # Apply Algorithm 2 using $\mathcal{T}$
        Update $\mathcal{L}$ with the loss $-\langle \boldsymbol{\alpha}^t, \nabla_{\boldsymbol{\alpha}} G(\boldsymbol{\alpha}^t, X^{t-1}) \rangle$     # Update the online learner $\mathcal{L}$
        **if** $G(\boldsymbol{\alpha}^t, X^{t-1}) \leq 0$ **then break; end if**     # Early stopping criterion
    **end for**
    $\widehat{X} \leftarrow X^t$ **if** $t = T$, $X^{t-1}$ **otherwise**

---

## 4   Theoretical guarantees of F3I

In a nutshell, F3I iteratively improves the imputed values by changing the weight vector that combines the $K$ neighbors among the naively imputed points for each sample. One of the most common metrics to evaluate the imputation quality is the Mean Squared Error (MSE) on the imputed values.

**Definition 4.1.** Mean squared error. We define the mean squared error as $\mathcal{L}^{\text{MSE}}(X^t, X^\star) \triangleq 1/(NF) \sum_{i \leq N} \sum_{f \leq F} ((x^t)_i^f - (x^\star)_i^f)^2$. The root-mean-squared error (RMSE) is then defined as $\mathcal{L}^{\text{RMSE}}(X^t, X^\star) \triangleq \sqrt{\mathcal{L}^{\text{MSE}}(X^t, X^\star)}$.

Note that F3I–and all of the baselines that we consider in our experimental study in Section 6–does not get access to the ground truth values and does not need to compute the mean squared error during training. However, we can still derive useful properties of F3I on the MSE. Imputation by convex combinations is theoretically supported by the following upper bound when the data distribution of true values follows Assumption B.1 and one of the missingness mechanisms in Assumptions B.2-B.4.

**Theorem 4.2.** Bounds in high probability and in expectation on the MSE for F3I. *Under Assumptions B.1-B.6, if $X^t$ is any imputed matrix at iteration $t \geq 1$, $X^\star$ is the corresponding full (unavailable in practice) matrix, w.h.p. $1 - 1/N$, $\mathcal{L}^{MSE}(X^t, X^\star) \leq \mathcal{O}((\sigma^{miss})^2 + \ln N/F)$, where $\sigma^{miss}$ is linked to the variance of the data distribution and depends on the missingness mechanism.*

In particular, this theorem means that the imputation quality decreases with the variance in the data, which is what we expect, as convex imputations would hardly be able to generate outlier data points. The proof and full expression of the bound in Theorem 4.2 are located in Appendix D.

The other performance measure that we are interested in is the preservation of the data distribution, which we quantify with function $G$. However, function $G$ features the Gaussian kernel density $D_0$ estimated on the naively imputed points $\{(\boldsymbol{x}^0)_1, \ldots, (\boldsymbol{x}^0)_N\}$. What we would want to optimize for is the "true" probability density $D_\star$ computed on the ground truth values $\{(\boldsymbol{x}^\star)_1, \ldots, (\boldsymbol{x}^\star)_N\}$ which are of course unavailable at all times.

Then we introduce function $G_\star : \boldsymbol{\alpha}, X \mapsto 1/N \sum_{i \leq N} \log D_\star(\boldsymbol{x}_i(\boldsymbol{\alpha}))/D_\star(\boldsymbol{x}_i) - \eta \|\boldsymbol{\alpha}\|_2^2$. We measure the imputation quality by the improvement in the probability of imputed points across iterations, that is, $\sum_{s=1}^{t} G_\star(\boldsymbol{\alpha}^s, X^{s-1})$, where $X^s \triangleq (\boldsymbol{x}_i^{s-1}(\boldsymbol{\alpha}^s))_{i \leq N}$ for $s \geq 1$ and $X^0$ is the data matrix imputed by the initial KNN imputer. Note that this quantity features a telescoping series and is then equivalent to comparing the final imputed values at time $s = t$ and the initial values at $s = 0$ (Proposition G.7 in Appendix). We compare this improvement with the imputation with the one incurred by the weight vector which *a posteriori* maximizes the likelihood of imputed points for all previous iterations up to $t$, that is,

$\mathcal{R}(t) \triangleq \max_{\boldsymbol{\alpha} \in \triangle_K} \sum_{s=1}^t G_\star(\boldsymbol{\alpha}, X^{s-1}) - G_\star(\boldsymbol{\alpha}^s, X^{s-1})$. In the online learning community, this measure is akin to the cumulative regret for the loss function $-G_\star$. [1]

In F3I, we use a no-regret learner named AdaHedge (De Rooij et al., 2014) to predict the weight vector at each iteration.

**Definition 4.3.** No-regret learners. A learner $\mathcal{L}$ over $\triangle_K$ is no-regret if for $t \geq 1$ and any sequence of bounded gains $\{g_s(\boldsymbol{\alpha})\}_{s \leq t}$ for any $\boldsymbol{\alpha} \in \triangle_K$, there exists $C \in \mathbb{R}^{+*}$ such that, if $\boldsymbol{\alpha}$ is the prediction of $\mathcal{L}$ at iteration $s \leq t$, then $\max_{\boldsymbol{\alpha} \in \triangle_K} \sum_{s=1}^t g_s(\boldsymbol{\alpha}) - g_s(\boldsymbol{\alpha}^s) \leq C\sqrt{t}$.

We denote $C_G^{\text{AH}} = \mathcal{O}(\sqrt{\log(K)})$ the constant associated with the regret bound incurred by AdaHedge on the objective function $G$. Combined with an upper bound on the difference between $G_\star$ and $G$ in high probability, we obtain the following upper bound on the imputation quality for F3I.

**Theorem 4.4.** High-probability upper bound on the imputation quality for F3I. *Under Assumptions B.1-B.6, for $X \in (\mathbb{R} \cup \{N/A\})^{N \times F}$, $\mathcal{R}(t) \leq C_G^{AH}\sqrt{t} + H^{miss}h^{-1}t$ w.h.p. $1 - 1/N$, where $H^{miss} = \mathcal{O}(F + \ln N)$ depends on the missingness mechanism and $h$ is chosen to guarantee that $G$ is concave in its first argument (Proposition C.2).*

*Proof.* The full proof is in Appendix E. Applying the regret bound associated with AdaHedge (Lemma 2 in Appendix) leads to an upper bound on quantity $\max_{\boldsymbol{\alpha} \in \triangle_K} \sum_{s=1}^t g_s(\boldsymbol{\alpha}) - g_s(\boldsymbol{\alpha}^s)$ which correspond to the difference in gain between the *a posteriori* optimal weights $\boldsymbol{\alpha}$ and the weights predicted in F3I $\boldsymbol{\alpha}^s$, $s \leq t$, using the gain $g_s : \boldsymbol{\alpha} \mapsto \boldsymbol{\alpha}^\intercal \nabla_{\boldsymbol{\alpha}} G(\boldsymbol{\alpha}^s, X^{s-1})$ at iteration $s$.

We denote $\widehat{\mathcal{R}}(\boldsymbol{\alpha}, t) \triangleq \sum_{s=1}^t G(\boldsymbol{\alpha}, X^{s-1}) - G(\boldsymbol{\alpha}^s, X^{s-1})$ for any $\boldsymbol{\alpha} \in \triangle_K$. For any $\boldsymbol{\alpha} \in \triangle_K$, $\widehat{\mathcal{R}}(\boldsymbol{\alpha}, t) \leq \max_{\boldsymbol{\alpha} \in \triangle_K} \sum_{s=1}^t (\boldsymbol{\alpha} - \boldsymbol{\alpha}^s)^\intercal \nabla_{\boldsymbol{\alpha}} G(\boldsymbol{\alpha}^s, X^{s-1})$ by using the gradient trick on the concave function $G$. Finally, we derive a high probability upper bound on $|G_\star(\boldsymbol{\alpha}, X') - G(\boldsymbol{\alpha}, X')|$ for any $\boldsymbol{\alpha} \in \triangle_K$ and $X' \in \mathbb{R}^{N \times F}$. We show that it suffices to find an upper bound $H^{\text{miss}}$ with high probability $1 - 1/N$ on $\max_{i \leq N} \|(\boldsymbol{x}^0)_i - (\boldsymbol{x}^\star)_i\|_2^2$, which is the norm of a random vector with independent, zero-mean subgaussian coordinates, which allows us to use Bernstein's inequality (Corollary G.6 with $\delta = 1/N$).

Subsequently, we show that for all $\boldsymbol{x} \in \mathbb{R}^d$, $|\log(D_0(\boldsymbol{x})/D_\star(\boldsymbol{x}))| \leq H^{\text{miss}}/(4h)$ with probability $1 - 1/N$. Finally, the definitions of $G_0$ and $G_\star$ allow us to derive the second term of the sum in the upper bound. $\square$

The term in $\mathcal{O}(t)$ comes from the approximation in $\mathcal{O}(1)$ made between $G$ and $G_\star$ (Corollary G.5) at each round of F3I. Removing that term would perhaps require supplementary steps, *e.g.*, considering $D_t$, the density computed on points $\{(\boldsymbol{x}^t)_1, (\boldsymbol{x}^t)_2, \ldots, (\boldsymbol{x}^t)_N\}$ at iteration $t$, instead of $D_0$.

## 5 Jointly training the imputation model and a model for a downstream task

As noticed by several prior works (Le Morvan et al., 2021; Le Morvan & Varoquaux, 2024; Vo et al., 2024), a good imputation quality does not necessarily go hand in hand with an improved performance in a downstream task run on the imputed data set, *e.g.*, for classification (Le Morvan et al., 2021), regression (Ayme et al., 2023), or structure learning (Vo et al., 2024). That might explain why, in some cases, data sets imputed with naive constant imputations that are known to distort the initial data distribution might yield better performance metrics than those with more sophisticated approaches (Le Morvan & Varoquaux, 2024). In this section, we propose a generic approach that optimizes both for an imputation task and a specific downstream task, by learning the optimal (convex) imputation pattern for some model parameters.

Assuming that there is a convex, differentiable pointwise loss function $\ell$ for the downstream task, we now consider the maximization problem $\max_{\boldsymbol{\alpha} \in \triangle_K} \mathcal{G}(\boldsymbol{\alpha}, X; \beta)$ on $X \in \mathbb{R}^{N \times F}$ on $\boldsymbol{\alpha}$, where

$$\mathcal{G}(\boldsymbol{\alpha}, X; \beta) \triangleq (1 - \beta)G(\boldsymbol{\alpha}, X) - \beta/N \sum_{i \leq N} \ell(\boldsymbol{x}_i(\boldsymbol{\alpha})) , \tag{2}$$

---
[1] However, that loss function is not necessarily non-negative.

where $\beta \in [0,1]$ is a positive regularization parameter related to the importance of the downstream task. As reported in many papers on multi-task learning (Chen et al., 2018; Yu et al., 2020; Liu et al., 2021), simply replacing the gradient of G in the loss of the AdaHedge learner in F3I by the weighted sum of the gradient of G and $\ell$ might lead to optimization issues, for instance, stalling update due to orthogonal gradients.

A recent method named PCGrad (Yu et al., 2020) performs gradient surgery for multi-task learning. In particular, PCGrad allows us to obtain theoretical guarantees on the performance of the training if the weighted sum $-\mathcal{G}$ is convex and $L$-Lipschitz continuous with $L > 0$ and if both $\ell$ and $-G$ are convex and differentiable (Yu et al., 2020, Theorems 1-2). $-G$ is convex by Proposition C.2 and differentiable by Proposition C.1. Naturally, if $\nabla \ell$ is itself Lipschitz continuous with a positive Lipschitz constant, Proposition C.5 implies that this condition is verified for the objective function in equation 2. A simple example of such a loss function is the pointwise log loss $\ell(\boldsymbol{x}) = -y \log C_{\boldsymbol{\omega}}(\boldsymbol{x})$ for the binary classification task, where $y$ is the true class in $\{0,1\}$ for sample $\boldsymbol{x}$ and $C_{\boldsymbol{\omega}} : \boldsymbol{x} \mapsto 1/(1 + \exp(-\boldsymbol{\omega}^\mathsf{T}\boldsymbol{x}))$ is the sigmoid function of parameter $\boldsymbol{\omega}$. Proofs are in Appendix (Section F).

Then, we modify F3I by changing the loss fed to the AdaHedge learner $\mathcal{L}$ in Line 10 in Algorithm 1. At iteration $s$, instead of using the loss $g_s(\boldsymbol{\alpha}) \triangleq -\langle \boldsymbol{\alpha}, \nabla_{\boldsymbol{\alpha}} G(\boldsymbol{\alpha}^s, X^{s-1}) \rangle$, we consider $\overline{g}_s(\boldsymbol{\alpha}) \triangleq -\langle \boldsymbol{\alpha}, \mathcal{L}(\boldsymbol{\alpha}, X^{s-1}) \rangle$, where $\mathcal{L}(\boldsymbol{\alpha}, X^{s-1})$ is equal to

$$(1-\beta)\nabla_{\boldsymbol{\alpha}} G^{\mathrm{PC}}(\boldsymbol{\alpha}^s, X^{s-1}) - \beta/N \sum_{i \leq N} \nabla_{\boldsymbol{\alpha}} \ell^{\mathrm{PC}}((\boldsymbol{x}^{s-1})_i(\boldsymbol{\alpha}^s)) \rangle \,, \tag{3}$$

and $\nabla_{\boldsymbol{\alpha}} G^{\mathrm{PC}}$ and $\nabla_{\boldsymbol{\alpha}} \ell^{\mathrm{PC}}$ are the gradient function of $G$ and $\ell$ with respect to their first argument corrected by the PCGrad procedure (Yu et al., 2020, Algorithm 1). We call PCGrad-F3I this joint training version of F3I. Under the conditions laid in the statement of Theorem 2 in (Yu et al., 2020), at any iteration $s \leq t$, if $(\boldsymbol{\alpha}^s)^{\mathrm{PC}}$ and $\boldsymbol{\alpha}^s$ are respectively the parameters obtained after applying one PCGrad or a regular AdaHedge update to $\boldsymbol{\alpha}^{s-1}$, then $\mathcal{G}((\boldsymbol{\alpha}^s)^{\mathrm{PC}}, X^{s-1}; \beta) \geq \mathcal{G}(\boldsymbol{\alpha}^s, X^{s-1}; \beta)$. That is,

**Theorem 5.1.** High-probability upper bound on the joint imputation-downstream task performance. *Under Assumptions B.1-B.6, for any $X \in (\mathbb{R} \cup \{N/A\})^{N \times F}$, convex pointwise loss $\ell$ such that $\nabla \ell$ is Lipschitz-continuous, and $\beta \in [0,1]$, under the conditions in Theorem 2 from (Yu et al., 2020), $\max_{\boldsymbol{\alpha} \in \triangle_K} \sum_{s=1}^{t} \mathcal{G}(\boldsymbol{\alpha}, X^{s-1}; \beta) - \mathcal{G}(\boldsymbol{\alpha}^s, X^{s-1}; \beta) \leq C_{(G,\ell)}^{AH}\sqrt{t} + (1 - \beta)H^{miss}h^{-1}t$ w.h.p. $1 - 1/N$, where $H^{miss} = \mathcal{O}(F + \ln N)$ depends on the missingness mechanism, $h$ is chosen to guarantee that $G$ is concave, and $C_{(G,\ell)}^{AH}$ is the constant related to AdaHedge being applied with gains $\overline{g}_s(\cdot)$.*

For $\beta = 0$, this bound matches Theorem 4.4, and for $\beta = 1$, this is the classical AdaHedge regret bound (Theorem 8 in (De Rooij et al., 2014)) with loss $\ell$.

# 6 Experimental study

This section is restricted to the comparison of our algorithmic contributions F3I and PCGrad-F3I to baselines for imputation-only and joint imputation-binary classification tasks on real-world data sets, due to space constraints. In Appendix (Section H), we also empirically validate our theoretical results (Theorems 4.2, 4.4 and 5.1) and test the imputation and classification performance on additional real-world data sets for drug repurposing (with up to $9,000$ features), compared to other (older) baselines, and on synthetic data sets (with up to $20,000$ features) that comply with Assumptions B.1-B.6, for all missingness mechanisms. Further information about hyperparameter tuning and computing infrastructure is also available in Section H. We also report complementary experiments in Appendix I (e.g., on other missingness mechanisms).

## 6.1 Imputation-only task

First, we study the imputation quality–without any downstream task. We resorted to the framework HyperImpute (Jarrett et al., 2022) to implement and run the benchmark for an imputation task across different performance metrics (including RMSE) on four standard data sets BreastCancer (Wolberg et al., 1993), Diabetes (from scikit-learn (Pedregosa et al., 2011)), HeartDisease (Janosi et al., 1989), Ionosphere (selva86, 2024) and a data set for drug repurposing, Gottlieb (Luo et al., 2016). The missing rate is at 30%.

Table 1: Average and standard deviation values of imputation quality metrics and runtime across 10 seeds.

| Data set | Alg. | RMSE ↓ | MAE ↓ | WD ↓ | Runtime ↓ |
|---|---|---|---|---|---|
| BreastCancer | **F3I (ours)** | **0.08** ±**0.03** | **0.03** ±**0.01** | **0.06** ±**0.02** | 0.14 ±0.04 |
| | kNN (uniform) | 0.31 ±0.02 | 0.12 ±0.01 | 0.29 ±0.02 | **0.01** ±**0.00** |
| | kNN (distance) | 0.29 ±0.02 | 0.11 ±0.01 | 0.27 ±0.02 | **0.01** ±**0.00** |
| | GAIN | 0.28 ±0.03 | 0.11 ±0.02 | 0.25 ±0.05 | 34 ±6 |
| | GRAPE | 0.37 ±0.03 | 0.19 ±0.02 | 0.28 ±0.03 | 5,091 ±1,707 |
| | HyperImpute | (+231%) 0.26 ±0.03 | 0.09 ±0.02 | 0.22 ±0.04 | 7 ±2 |
| | MIRACLE | 4.32 ±0.35 | 4.22 ±0.33 | 10.00 ±1.04 | 77 ±4 |
| | NewImp | 415 ±189 | 294 ±179 | 695 ±416 | 3,282 ±819 |
| | Remasker | 0.27 ±0.03 | 0.11 ±0.02 | 0.25 ±0.04 | 393 ±32 |
| | TDM | 0.35 ±0.04 | 0.21 ±0.03 | 0.40 ±0.05 | 532 ±22 |
| Diabetes | **F3I (ours)** | (+27%) 0.26 ±0.00 | 0.18 ±0.00 | 0.18 ±0.01 | 0.17 ±0.03 |
| | kNN (uniform) | **0.23** ±**0.01** | 0.16 ±0.00 | 0.21 ±0.01 | **0.02** ±**0.01** |
| | kNN (distance) | 0.24 ±0.01 | 0.16 ±0.00 | 0.20 ±0.01 | **0.02** ±**0.00** |
| | GAIN | 0.24 ±0.01 | 0.17 ±0.01 | 0.27 ±0.01 | 4.72 ±1.36 |
| | GRAPE | 0.34 ±0.01 | 0.27 ±0.01 | 0.37 ±0.02 | 120 ±3 |
| | HyperImpute | 0.24 ±0.01 | **0.13** ±**0.00** | **0.14** ±**0.02** | 35.4 ±10.1 |
| | MIRACLE | 4.37 ±0.16 | 4.29 ±0.16 | 10.33 ±0.38 | 8.18 ±1.16 |
| | NewImp | 0.39 ±0.03 | 0.30 ±0.03 | 0.72 ±0.07 | 12,508 ±152 |
| | Remasker | **0.23** ±**0.01** | 0.14 ±0.00 | 0.15 ±0.00 | 246 ±13 |
| | TDM | 0.29 ±0.01 | 0.23 ±0.01 | 0.21 ±0.01 | 190 ±28 |
| Gottlieb | **F3I (ours)** | (+111%) 0.04 ±0.01 | 0.02 ±0.00 | 0.03 ±0.00 | 2 ±1 |
| | kNN (uniform) | 0.04 ±0.00 | 0.02 ±0.00 | 0.03 ±0.00 | **0.22** ±**0.04** |
| | kNN (distance) | 0.04 ±0.00 | 0.02 ±0.00 | 0.02 ±0.00 | **0.22** ±**0.03** |
| | GAIN | 0.04 ±0.01 | 0.02 ±0.00 | 0.02 ±0.00 | 103 ±9 |
| | GRAPE | 0.12 ±0.01 | 0.09 ±0.01 | 0.12 ±0.02 | 6,670 ±217 |
| | HyperImpute | **0.03** ±**0.00** | **0.01** ±**0.00** | **0.01** ±**0.00** | 44 ±18 |
| | MIRACLE | 4.44 ±0.32 | 4.38 ±0.31 | 10.50 ±0.75 | 212 ±20 |
| | NewImp | 131 ±72.4 | 71.3 ±47.2 | 170 ±112 | 12,933 ±621 |
| | Remasker | 0.18 ±0.04 | 0.14 ±0.03 | 0.30 ±0.08 | 3,016 ±42 |
| | TDM | - | - | - | - |
| HeartDisease | **F3I (ours)** | **0.14** ±**0.05** | **0.07** ±**0.03** | - | 0.10 ±0.02 |
| | kNN (uniform) | 0.40 ±0.03 | 0.25 ±0.03 | - | **0.00** ±**0.0** |
| | kNN (distance) | 0.36 ±0.03 | 0.22 ±0.02 | - | 0.01 ±0.00 |
| | GAIN | 0.30 ±0.07 | 0.18 ±0.05 | - | 34 ±7 |
| | GRAPE | 0.54 ±0.03 | 0.38 ±0.04 | - | 536 ±439 |
| | HyperImpute | (+210%) 0.24 ±0.07 | 0.13 ±0.05 | - | 17 ±5 |
| | MIRACLE | 4.91 ±0.44 | 4.72 ±0.41 | - | 82 ±2 |
| | NewImp | 308 ±176 | 197 ±127 | - | 2,404 ±270 |
| | Remasker | 0.25 ±0.05 | 0.14 ±0.03 | - | 32 ±2 |
| | TDM | 0.48 ±0.04 | 0.36 ±0.03 | - | 340 ±219 |
| Ionosphere | **F3I (ours)** | (+11%) 0.21 ±0.06 | 0.15 ±0.05 | 0.29 ±0.12 | 0.19 ±0.03 |
| | kNN (uniform) | 0.22 ±0.04 | 0.15 ±0.03 | 0.31 ±0.07 | **0.00** ±**0.00** |
| | kNN (distance) | 0.22 ±0.04 | 0.15 ±0.03 | 0.30 ±0.07 | **0.00** ±**0.00** |
| | GAIN | 0.47 ±0.05 | 0.34 ±0.05 | 0.57 ±0.07 | 50 ±11 |
| | GRAPE | 0.48 ±0.03 | 0.39 ±0.04 | 0.49 ±0.12 | 765 ±110 |
| | HyperImpute | **0.20** ±**0.07** | **0.13** ±**0.05** | **0.26** ±**0.10** | 99 ±73 |
| | MIRACLE | 5.22 ±0.54 | 5.14 ±0.53 | 12.4 ±1.35 | 96 ±2 |
| | NewImp | 0.60 ±0.18 | 0.47 ±0.13 | 1.04 ±0.35 | 4,174 ±1,169 |
| | Remasker | 0.33 ±0.03 | 0.26 ±0.04 | 0.51 ±0.09 | 77 ±3 |
| | TDM | 0.39 ±0.04 | 0.33 ±0.04 | 0.57 ±0.10 | 697 ±13 |

Table 2: Average of RMSE score over 10 seeds (rounded to the closest third decimal place) on the Ionosphere dataset for varying missing rates and missingness mechanisms. KNN is the kNN algorithm with distance-dependent weights. Results for KNN with uniform weights are available in Table 22 in Appendix.

| | $p^{\mathrm{miss}}$ | **F3I (ours)** | KNN | HyperImpute | MIRACLE | GAIN | Remasker | GRAPE |
|---|---|---|---|---|---|---|---|---|
| M | 10% | 0.21 ±0.04 | 0.22 ±0.04 | **0.20 ±0.04** | 5.33 ±0.18 | 0.44 ±0.04 | 0.35 ±0.04 | 0.47 ±0.05 |
| C | 25% | 0.21 ±0.03 | 0.21 ±0.04 | **0.19 ±0.04** | 5.51 ±0.33 | 0.44 ±0.03 | 0.36 ±0.03 | 0.46 ±0.03 |
| A | 50% | **0.22 ±0.04** | 0.23 ±0.03 | **0.22 ±0.04** | 5.08 ±0.53 | 0.46 ±0.02 | 0.35 ±0.03 | 0.46 ±0.02 |
| R | 75% | **0.28 ±0.03** | 0.29 ±0.02 | **0.28 ±0.03** | 5.33 ±0.72 | 0.48 ±0.03 | 0.34 ±0.02 | 0.48 ±0.01 |
| | 90% | **0.35 ±0.02** | 0.37 ±0.02 | 0.36 ±0.03 | 4.89 ±0.75 | 0.48 ±0.027 | 0.36 ±0.04 | 0.44 ±0.02 |
| M | 10% | 0.20 ±0.06 | 0.22 ±0.05 | **0.17 ±0.05** | 5.69 ±0.16 | 0.40 ±0.06 | 0.33 ±0.05 | 0.45 ±0.06 |
| A | 25% | 0.19 ±0.04 | 0.20 ±0.04 | **0.18 ±0.04** | 5.46 ±0.19 | 0.47 ±0.05 | 0.31 ±0.03 | 0.43 ±0.03 |
| R | 50% | 0.23 ±0.03 | 0.25 ±0.03 | **0.22 ±0.03** | 5.35 ±0.24 | 0.46 ±0.04 | 0.32 ±0.04 | 0.43 ±0.04 |
| | 75% | **0.28 ±0.02** | 0.29 ±0.03 | **0.28 ±0.05** | 5.22 ±0.33 | 0.47 ±0.04 | 0.34 ±0.04 | 0.44 ±0.03 |
| | 90% | 0.37 ±0.03 | 0.37 ±0.02 | **0.35 ±0.03** | 5.26 ±0.41 | 0.45 ±0.03 | 0.37 ±0.03 | 0.44 ±0.02 |
| M | 10% | **0.19 ±0.05** | 0.21 ±0.05 | **0.19 ±0.05** | 5.48 ±0.17 | 0.42 ±0.04 | 0.34 ±0.05 | 0.44 ±0.04 |
| N | 25% | 0.20 ±0.04 | 0.21 ±0.04 | **0.19 ±0.04** | 5.30 ±0.24 | 0.45 ±0.04 | 0.32 ±0.04 | 0.45 ±0.03 |
| A | 50% | **0.23 ±0.03** | 0.24 ±0.03 | 0.22 ±0.03 | 5.11 ±0.46 | 0.46 ±0.02 | 0.32 ±0.03 | 0.45 ±0.02 |
| R | 75% | **0.28 ±0.02** | 0.30 ±0.02 | **0.28 ±0.05** | 4.94 ±0.67 | 0.48 ±0.03 | 0.33 ±0.03 | 0.44 ±0.01 |
| | 90% | 0.37 ±0.03 | 0.39 ±0.03 | **0.38 ±0.04** | 4.92 ±0.75 | 0.47 ±0.03 | 0.39 ±0.03 | 0.49 ±0.01 |

Table 3: Average of RMSE score over 10 seeds (rounded to the closest third decimal place) on the Breast Cancer dataset for varying missing rates and missingness mechanisms. KNN is the kNN algorithm with distance-dependent weights. Results for KNN with uniform weights are available in Table 23 in Appendix.

| | $p^{\mathrm{miss}}$ | **F3I (ours)** | KNN | HyperImpute | MIRACLE | GAIN | Remasker | GRAPE |
|---|---|---|---|---|---|---|---|---|
| M | 10% | **0.03 ±0.01** | 0.11 ±0.03 | 0.05 ±0.03 | 4.19 ±0.11 | 0.08 ±0.04 | 0.06 ±0.03 | 0.28 ±0.04 |
| C | 25% | **0.04 ±0.01** | 0.14 ±0.03 | 0.09 ±0.04 | 4.10 ±0.14 | 0.12 ±0.03 | 0.11 ±0.04 | 0.28 ±0.02 |
| A | 50% | **0.07 ±0.02** | 0.18 ±0.03 | 0.14 ±0.03 | 4.01 ±0.23 | 0.18 ±0.02 | 0.15 ±0.03 | 0.31 ±0.01 |
| R | 75% | **0.08 ±0.01** | 0.21 ±0.01 | 0.20 ±0.01 | 3.89 ±0.35 | 0.21 ±0.01 | 0.20 ±0.01 | 0.31 ±0.01 |
| | 90% | **0.13 ±0.07** | 0.22 ±0.01 | 0.21 ±0.01 | 3.84 ±0.39 | 0.23 ±0.03 | 0.21 ±0.01 | 0.31 ±0.01 |
| M | 10% | **0.13 ±0.02** | 0.55 ±0.06 | 0.48 ±0.06 | 5.48 ±0.32 | 0.49 ±0.06 | 0.48 ±0.06 | 0.65 ±0.07 |
| A | 25% | **0.11 ±0.02** | 0.41 ±0.02 | 0.38 ±0.02 | 4.80 ±0.21 | 0.38 ±0.02 | 0.38 ±0.02 | 0.47 ±0.03 |
| R | 50% | **0.10 ±0.02** | 0.31 ±0.01 | 0.29 ±0.01 | 4.40 ±0.23 | 0.29 ±0.01 | 0.29 ±0.01 | 0.37 ±0.02 |
| | 75% | **0.10 ±0.02** | 0.26 ±0.01 | 0.25 ±0.01 | 4.23 ±0.31 | 0.25 ±0.01 | 0.25 ±0.01 | 0.33 ±0.01 |
| | 90% | **0.10 ±0.01** | 0.25 ±0.00 | 0.24 ±0.01 | 4.28 ±0.40 | 0.24 ±0.01 | 0.24 ±0.01 | 0.32 ±0.01 |
| M | 10% | **0.10 ±0.02** | 0.39 ±0.04 | 0.34 ±0.04 | 4.83 ±0.21 | 0.35 ±0.04 | 0.34 ±0.04 | 0.50 ±0.04 |
| N | 25% | **0.08 ±0.02** | 0.31 ±0.02 | 0.27 ±0.02 | 4.37 ±0.21 | 0.29 ±0.02 | 0.28 ±0.02 | 0.38 ±0.02 |
| A | 50% | **0.08 ±0.01** | 0.25 ±0.01 | 0.23 ±0.01 | 4.47 ±0.43 | 0.24 ±0.01 | 0.23 ±0.01 | 0.34 ±0.02 |
| R | 75% | **0.08 ±0.01** | 0.24 ±0.01 | 0.22 ±0.01 | 3.94 ±0.29 | 0.23 ±0.01 | 0.22 ±0.01 | 0.30 ±0.01 |
| | 90% | **0.10 ±0.24** | 0.24 ±0.01 | 0.23 ±0.01 | 3.83 ±0.44 | 0.24 ±0.01 | 0.23 ±0.01 | 0.32 ±0.01 |

In this benchmark, we included recent methods from the literature which benefited from open-source, modular `scikit-learn` (Pedregosa et al., 2011)-like implementations: GAIN (Yoon et al., 2018a), GRAPE (You et al., 2020), HyperImpute (Jarrett et al., 2022), MIRACLE (Kyono et al., 2021), NewImp (Chen et al., 2024), Remasker (Du et al., 2023) and TDM (Zhao et al., 2023). We considered the scenario MNAR in the framework HyperImpute to add missing values. We report in Table 1 the corresponding numerical results across 10 runs with different random seeds. The root mean square error (RMSE), the mean average error (MAE), the Wasserstein distance (WD), along with the runtime in seconds are shown. We also add the results for the kNN algorithms respectively with uniform and distance-dependent weights (*i.e.*, inversely

proportional to the distance to the neighbor). Note that HeartDisease has native missing values, which is why the Wasserstein distance cannot be computed, and TDM failed on the Gottlieb data set. We also perform additional experiments on the Ionosphere and the Breast Cancer data sets for varying missing rates (0.1, 0.25, 0.5, 0.75, 0.9) and missingness mechanisms (MCAR, MAR, MNAR). The results are reported in Table 2 (Ionosphere) and 3 (Breast Cancer). Bold type is the top performer, underline denotes the second best (and average percentage of deterioration of performance across metrics compared to the top performer).

Those results show that, on the imputation task alone, F3I offers a good tradeoff between imputation quality (regardless of the performance metric) and computational efficacy (runtime). Indeed, F3I performs on par or better than the state-of-the-art, while remaining computationally efficient by several orders of magnitude. We also report numerical results across 100 runs restricted to the best baselines in Table 20 in Appendix (Section H), which confirm these observations.

## 6.2   Joint imputation-binary classification task

Second, we implement the joint imputation-classification training with the log-loss function and sigmoid classifier $\ell(\boldsymbol{x}) \triangleq -y \log C_{\boldsymbol{\omega}}(\boldsymbol{x})$ mentioned in Section 5, where $y \in \{0, 1\}$ is the binary class associated with sample $\boldsymbol{x} \in \mathbb{R}^F$. To implement PCGrad-F3I, we chain the imputation phase by F3I with an MLP classifier, which returns logits. At time $t$, the imputation part applies at a fixed set of parameters $\boldsymbol{\omega}^t$ with the learner losses defined in equation 3.

Table 4: Average and standard deviation of Area Under the Curve (AUC) values (rounded to the closest second decimal place) across several runs on the joint imputation-classification task (MCAR scenario, $p^{\text{miss}} = 50\%$). Bold type is the top performer, underline denotes the second best (and average percentage of change in performance across data sets compared to the top performer RF-GAP). * Remasker and HyperImpute are extremely slow when combined with a MLP, especially on the largest data sets MNIST and PREDICT, which is why the values are shown for 20 iterations for HyperImpute on the MNIST data set, and otherwise missing.

| Task | Data | BreastCancer | Ionosphere | MNIST | PREDICT |
|---|---|---|---|---|---|
| Joint Imputation-Classification | GRAPE | 0.55 ±0.13 | 0.71 ±0.12 | **1.00 ±0.00** | 0.49 ±0.07 |
| | NeuMiss | 0.62 ±0.18 | 0.70 ±0.16 | 0.99 ±0.07 | 0.50 ±0.01 |
| | **PCGradF3I (ours)** | (+7%) **0.70 ±0.14** | 0.77 ±0.14 | 0.99 ±0.09 | 0.51 ±0.01 |
| Separate Imputation-Classification | HyperImpute | 0.56 ±0.14 | 0.74 ±0.18 | 0.82 ±0.24* | –* |
| | K-NN | 0.54 ±0.13 | 0.74 ±0.15 | 0.93 ±0.17 | 0.47 ±0.07 |
| | Mean | 0.52 ±0.07 | 0.72 ±0.13 | 0.64 ±0.18 | 0.48 ±0.00 |
| | Remasker | 0.56 ±0.13 | 0.80 ±0.15 | –* | –* |
| | RF-GAP | 0.50 ±0.00 | **0.83 ±0.17** | **1.00 ±0.00** | **0.53 ±0.13** |

We compare the classification performance of PCGrad-F3I with *imputation methods with separate classifier training*: imputing by the mean (Mean), a K nearest-neighbor algorithm with weights inversely proportional to the distance to neighbors (K-NN (Troyanskaya et al., 2001)), a random forest classifier (RF-GAP (Rhodes et al., 2023)), or one of the top baselines for imputation (HyperImpute (Jarrett et al., 2022) and Remasker (Du et al., 2023)) prior to applying the MLP classifier; or with *imputation methods with* joint *classifier training* like PCGradF3I: adding a NeuMiss block (Le Morvan et al., 2020) to the MLP classifier, or training simultaneously GRAPE (You et al., 2020) and the MLP classifier. We use the same MLP architecture across all imputation techniques. The criterion for training the models is the log loss, and we split the samples into training (70%), validation (20%), and testing (10%) sets, where the former two sets are used for training the MLP and hyperparameter finetuning (see Appendix in Section H.2), and the performance metric–Area Under the Curve (AUC)–is computed on the latter set. Further experimental details can be found in Appendix (Section H). We consider the MNIST dataset (LeCun et al., 1998), which comprises grayscale images of handwritten digits, and another drug repurposing data set, PREDICT (Réda, 2023a). In MNIST, we restrict our study to images annotated with class 0 or 1 to get a binary classification problem. Moreover, we also include again the BreastCancer (Wolberg et al., 1993) and Ionosphere (selva86, 2024) data sets with their native classification

labels. In all data sets, we remove pixels at random with probability 50% using a MCAR mechanism. Table 4 displays the numerical results across 100 iterations with different random seeds.

The empirical performance of PCGrad-F3I on classification tasks is on par with the state-of-the-art, with a very small average deterioration of performance compared to the top baseline RF-GAP (Rhodes et al., 2023) ($-2.1\%$ in AUC). However, we argue that the fairest baselines to benchmark PCGradF3I for classification might be the imputation approaches with joint classifier training, in which case PCGradF3I comes on top. We also show in Appendix (Figures 9-10 in Section H) that PCGradF3I preserves the correct shapes in MNIST across missingness proportions and mechanisms.

## 7 Limitations

We list here three limitations of our contribution. First, F3I is based on iterative improvements of a K-NN imputer. Yet, K-NN imputation is costly when the number of samples is very large. A solution is to use approximate neighbor-finding algorithms such as FAISS (Johnson et al., 2019), LSH (Zhao et al., 2014; Tsai & Yang, 2014) or Annoy (Bernhardsson, 2018) and leverage the use of GPUs to accelerate F3I. Moreover, F3I might also be prone to the curse of dimensionality, due to its similarity to KNN and the use of a kernel density estimator, which might require the application of (invertible) dimensionality reduction techniques prior to the imputation by F3I, for instance, using a Principal Component Analysis (PCA). Under Assumptions B.1-B.6, our theoretical results still hold in that case. Second, F3I can only be used for continuous variables. Third, the theoretical guarantees derived in Theorems 4.2-5.1 require strong assumptions on the data distribution (*i.e.*, Gaussian distributed data), which may not accurately reflect the statistical properties of data sets in practical applications. However, as shown in Section 6 and H in Appendix, F3I can still be applied to real-world data and be competitive.

## 8 Discussion

We introduce an algorithm named F3I which iteratively improves a K-nearest neighbor imputation, by tuning the neighbor-associated weights. F3I is versatile as it can be jointly trained with any classification task to meaningfully impute values depending on the end goal. Moreover, F3I features theoretical guarantees on the imputation quality and the preservation of the data distribution across missingness mechanisms, including not-missing-at-random. Empirically, the performance of F3I is similar or better than the state-of-the-art across data sets, while being more computationally tractable. The experimental code and implementation of F3I are provided as supplementary material.

Combining online learning and density ratio estimation is a simple and flexible idea that could be improved further, notably to perhaps remove the linear term in the number of iterations in Theorem 4.4. For instance, the density ratio estimation step might benefit from the classifier-based approach developed in BORE (Tiao et al., 2021), in particular in a version of F3I where a k-d tree would be rebuilt at every iteration to consider density $D_t$ on points $\{(\boldsymbol{x}^t)_1, \ldots, (\boldsymbol{x}^t)_N\}$ instead of the density estimated on the naively imputed initial points $\{(\boldsymbol{x}^0)_1, \ldots, (\boldsymbol{x}^0)_N\}$. Another avenue of research would be to adapt F3I to heterogenous data with both continuous or categorical features. A naive approach would use Gower's distance (Gower, 1971) instead of Chebyshev distance to find the K nearest neighbors for any point, and then use a consensus procedure to obtain the imputed value depending on the type of variable (categorical or continuous). This approach achieves a significant improvement over HyperImpute, which natively handles heterogeneous data. Further details can be found in Appendix H.3. One might also wonder whether imputation could be improved by considering sample-specific weights, as, currently, $\boldsymbol{\alpha}$ is shared by all samples. This creates a new computational challenge as the number of samples grows. Finally, for a new sample $\boldsymbol{x} \in (\mathbb{R} \cup \{\texttt{N/A}\})^F$, is there a way not to re-run the full F3I procedure? If we assume that the new sample comes from the data set, the simplest idea is to apply on $\boldsymbol{x}$ the initial imputer and successively the imputation improvement model in Algorithm 2 with the weight vector $\boldsymbol{\alpha}^t$, where $t$ is the final step of F3I. However, this out-of-sample imputation loses the theoretical guarantees in Section 4. Finding a theoretically-backed approach for out-of-sample imputation would also be an interesting question to investigate.

**Broader Impact Statement**

The contributions of this paper are essentially theoretical. However, by the definition of our imputation algorithm, possibly sensitive information can leak to imputed values without specific steps taken to avoid it, potentially undermining fairness in downstream analyses. This might be mitigated by modifying the imputation model in Algorithm 2 for instance.

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

## A Other related works and comments on F3I

**Related works** Relevant methods developed for multivariate time-series data might also be adapted to single-timepoint data sets: Conditional Score-based Diffusion Models (CDSI) (Tashiro et al., 2021a), Generative Adversarial Networks (Luo et al., 2018; Yoon et al., 2018a) or Last Observation Carried Forward (LOCF), where the last non-missing value is duplicated until the next non-missing time point. However, those methods are best-suited in the case where temporal connections can be made, and we restrict our study to single-timepoint data.

Due to diffusion models' impressive generative modeling capacity, many imputation approaches based on them have been developed in recent years. One of the first approaches in this direction was Conditional Score-Based Diffusion models (Tashiro et al., 2021b), which attempts to impute values at missing time points in multivariate time-series data by conditioning the diffusion model on the observed time points and then denoising from white noise using the conditional diffusion model. Another similar approach (Chen et al., 2023) constructs and solves a Schrödinger Bridge problem with conditional constraints to impute missing values in time-series data, where the conditional constraints are derived from the observed values. However, both these approaches are only available for imputation in multivariate time-series data, which is out of the scope of our paper.

An interesting approach for tabular data (Zheng & Charoenphakdee, 2022) consists of removing the temporal transformer layers from CSDI (Tashiro et al., 2021b) and constructing embeddings for categorical variables, after which the same diffusion procedure as CSDI is applied. Another diffusion-based approach (Jolicoeur-Martineau et al., 2024) constructs multiple copies of the dataset with different noise samples added, on which gradient-boosted trees are trained to perform the reverse diffusion process to reconstruct the missing data values. The NewImp (Chen et al., 2024) model, on the other hand, attempts to learn the missing value distribution by learning the score function of the joint distribution using a Denoising Score Matching approach and then by stepwise reconstructing the missing values using an ODE simulation of the joint distribution based on these learned scores. Another similar approach, MissDiff (Ouyang et al., 2025), attempts to learn a score-based model by modifying the loss in a conventional Variance Preserving SDE for diffusion models to only consider the observed values.

A variational autoencoder-based approach (Peis et al., 2022) generates the missing data by training multiple separate variational autoencoders to encode the latent space of each feature and the dependencies among them. This training is helpful in the presence of diverse feature types. New approaches also explore the use of transformer models in missing data imputation. For instance, ReMasker (Du et al., 2024) is an extension of the Masked AutoEncoder (He et al., 2021) where the encoder and decoder models consist of a sequence of transformer layers, and the model learns by masking additionally available values and learning to predict them, similar to CSDI (Tashiro et al., 2021b). Another approach based on a modification of OTImputer (Muzellec et al., 2020b) attempts to minimize the Wasserstein distance between latent representations of the original and imputed data set generated via a neural network, where the latent representation generating neural network is fit to maximize the Mutual Information to prevent model collapse.

Other generative approaches involve using Generative Adversarial Networks. The simplest GAN-based approach (Li et al., 2019) jointly trains two pairs of generator-discriminator networks, one to predict the data and the other to predict the missing masks. Another GAN-based approach masks certain values in the data set besides the missing ones. It uses the generator to generate all masked values, after which the discriminator is trained to predict the mask matrix from the imputed data set (Yoon et al., 2018b).

---

**Algorithm 2** Imputation improvement model $\texttt{Impute}(\cdot; \boldsymbol{\alpha}, Z)$

---

**Input:** Guess for a $F$-dimensional $\boldsymbol{x} \in \mathbb{R}^F$, missing indicator for that sample $\boldsymbol{m} \in \{0,1\}^F$
**Parameters:** Number of neighbors $K$, weights $\boldsymbol{\alpha} \in \triangle_K$, reference set $Z = \{\boldsymbol{z}^i\}_{0 \leq i \leq N} \subset \mathbb{R}^F$
**Output:** Improved guess $\widetilde{\boldsymbol{x}} \in \mathbb{R}^F$

\# Neighbor indices by increasing Chebychev distance: $\arg\min_{j \leq N}^{1,2,\ldots,K}$ selects the $K$ elements in $1, 2, \ldots, N$ with smallest values (with a k-d tree for instance)

$(n_1, n_2, \ldots, n_K) \leftarrow \arg\min_{j \leq N}^{1,2,\ldots,K} \max_{f \leq F} |x_f - z_f^j|$

$\widetilde{x}_f \leftarrow \sum_{k \leq K} \alpha_k z_f^{n_k}$ for any $f \leq F$, $m^f = 1$

---

Finally, GRAPE is a somewhat more novel approach (You et al., 2020), which consists of representing the data set as a labeled bipartite graph, with the two node types representing the samples and the features and each edge label representing the value of the particular feature for that sample. One then uses a GraphSAGE (Hamilton et al., 2017) inspired Graph Neural Network to predict the missing edge labels, which are the imputed values.

**Comparison of F3I to optimal transport-based imputation**     Authors of OTImputer in (Muzellec et al., 2020a) leverage optimal transport (OT) to define a loss function based on Sinkhorn divergences for imputation. This loss function, like $G$, aims at quantifying the gap in data distribution between any two random batches of samples from the data matrix and can also be iteratively minimized through a gradient descent approach. The imputation is performed feature-wise. However, this approach requires the input of several parameters, among which $t_{\max}$, the budget for the number of improvements, which is always fully exhausted (contrary to F3I where an early stopping criterion exists); $m$ the size of the randomly sampled batches and $K$ the number of batches which are evaluated. Moreover, contrary to F3I, the OT imputer does not provide theoretical guarantees on the imputation quality.

**Estimation of the kernel density in F3I**     Given a kernel K with bandwidth $h$–here, we use a Gaussian kernel and the value of $h$ as provided in Appendix C–the probability of a point $\boldsymbol{x}$ with respect to a set of points $\mathcal{X}$ is defined as $D_0(\boldsymbol{x}) = \sum_{\boldsymbol{x}' \in \mathcal{X}} K(\boldsymbol{x} - \boldsymbol{x}'; h)$. To ensure the stability of the optimization procedure, we compute directly the logarithm of the kernel density $D_0$ using function $\texttt{kernel\_density}(\cdot, \texttt{h=h,kernel=`gaussian'})$ from the k-d tree class in the Python package $\texttt{scikit-learn}$ (Pedregosa et al., 2011).

## B   Theoretical assumptions

In this section, we state the formal expression of our assumptions about the data generation procedure to derive theoretical guarantees from our algorithm. We explicitly write the pseudo-code for the imputation improvement model in Algorithm 2 and the full data generation procedure in Algorithm 3.

First, we assume that each value in the full data matrix is drawn from independent fixed-variance Gaussian distributions.

**Assumption B.1.** Independent Gaussian distributions. There exist $\boldsymbol{\mu} \in \mathbb{R}^F$ and $\sigma > 0$ such that, for any sample $i \leq N$ and any feature $f \leq F$, $(x^\star)_i^f \sim_{\text{iid}} \mathcal{N}(\mu_f, \sigma^2)$.

The random indicator variables $m_i^f$ are then independently drawn according to the missingness mechanism with probability $p^{\text{miss}}$. If $m_i^f = 1$, then the covariate at position $(i, f)$ $x_i^f$ in $X$ is unavailable, otherwise, $x_i^f = (x^\star)_i^f$. We will provide an analysis of our algorithm for three types of missingness mechanisms:

**Assumption B.2.** MCAR mechanism: Bernouilli distribution. The random indicator variables for missing values $m_i^f$ are drawn *iid* from $\mathcal{B}(p^{\text{miss}}(\boldsymbol{x}))$, where $p^{\text{miss}}(\boldsymbol{x}) \in (0,1)$ is a constant value for any $\boldsymbol{x}$.

**Assumption B.3.** MAR mechanism. We assume a subset $\mathcal{F}_o$ of size $F_o < F$ features is always observed. We denote $(x^\star)_i^{|\text{ obs}} \triangleq (x_i^\star[f])_{f \in \mathcal{F}_o}$. Then there exist a function $p^{\text{miss}}$, $\forall \boldsymbol{x} \in \mathbb{R}^{F_o}$, $\mathbb{P}(m_i^f = 1 \mid (x^\star)_i^{|\text{ obs}} = \boldsymbol{x}) = p^{\text{miss}}(\boldsymbol{x})$ .

---

**Algorithm 3** Data generation procedure according to Assumptions B.1-B.6

---

**Input:** $N$ number of samples, $F$ number of features
**Output:** Initial data $X^{\text{miss}} \in (\mathbb{R} \cup \{\texttt{NaN}\})^{N \times F}$ and naively imputed $X^0 \in \mathbb{R}^{N \times F}$
# Generation of the complete data set
**for** $i = 1, 2, \ldots, N$ **do**
    **for** $f = 1, 2, \ldots, F$ **do**
        $(\boldsymbol{x}^\star)_i \sim \mathcal{N}_F(\mu_f, \sigma^2)$ (where $\Theta \triangleq (\boldsymbol{\mu} = (\mu_1, \ldots, \mu_F), \sigma^2 \boldsymbol{I}_{F \times F}) \in \mathbb{R}^F \times \mathbb{R}^{F \times F}$)
    **end for**
**end for**
# Missingness mechanism
**for** $i = 1, 2, \ldots, N$ **do**
    **for** $f = 1, 2, \ldots, F$ **do**
        $m_i^f \sim_{\text{iid}} p^{\text{miss}}((\boldsymbol{x}^\star)_i, f)$
        **if** $m_i^f = 1$ **then**
            **then** $(x^{\text{miss}})_i^f \leftarrow \texttt{NaN}$
            **else** $(x^{\text{miss}})_i^f \leftarrow (x^\star)_i^f$
        **end if**
    **end for**
**end for**
# Create the naively imputed data set
**for** $i = 1, 2, \ldots, N$ **do**
    **for** $f = 1, 2, \ldots, F$ **do**
        **if** $m_i^f = 1$ **then**
            # K-nearest neighbor imputation with uniform weights
            # $k^{\text{th}}$ closest neighbor for $(\boldsymbol{x}^{\text{miss}})_i^f$ is denoted $\mathcal{K}((\boldsymbol{x}^{\text{miss}})_i, f, k)$
            **then** $(x^0)_i^f \leftarrow \frac{1}{K} \sum_{k \leq K} x_{\mathcal{K}((\boldsymbol{x}^{\text{miss}})_i, f, k)}^f = \frac{1}{K} \sum_{k \leq K} (x^\star)_{\mathcal{K}((\boldsymbol{x}^{\text{miss}})_i, f, k)}^f$
            **else** $(x^0)_i^f \leftarrow (x^\star)_i^f$
        **end if**
    **end for**
**end for**

---

**Assumption B.4.** MNAR mechanism: Gaussian self-masking (Assumption 4 from Le Morvan et al. (2020)). The probability of event $\{m_i^f = 1\}$ depends on $(x^\star)_i^f$: $\exists K_f \in (0, 1)$, $\forall x \in \mathbb{R}$, $\mathbb{P}(m_i^f = 1 \mid (x^\star)_i^f = x) = K_f e^{-\frac{1}{\sigma^2}(x - \mu_f)^2} = p^{\text{miss}}(\boldsymbol{x})$ .

We also ensure that there are exactly $K$ neighbors for the initial simple guesses and that we know a (constant) upper bound on the norm of any feature vectors.

**Assumption B.5.** Number of neighbors $K$. In the remainder of the paper, if $\{i \leq N \mid m_i^f = 0\}$ is the set of data point indices for which the feature $f$ is not missing, then $K \leq \min_{f \leq F} |\{i \leq N \mid m_i^f = 0\}|$. Without a loss of generality, $\min_{f \leq F} |\{i \leq N \mid m_i^f = 0\}| \geq 2$ (otherwise, we can ignore the corresponding feature).

**Assumption B.6.** Upper bound on any of the $(\|\boldsymbol{x}_i\|_2^2)_{i \leq n}$. We assume a constant $S > 0$ exists, such that for any $i \leq N$, $\|\boldsymbol{x}_i\|_2^2 \leq S$ (ignoring potential missing values). Up to renormalization, we assume that $S = 1$. Moreover, the initial imputation step (the "simple guess") preserves that condition, meaning that for any $i \leq N$ and $t \geq 0$, $\|\boldsymbol{x}_i^t\|_2^2 \leq S$, where $X^0$ is the imputed data matrix with the initial imputation step, and $X^t$ for $t \geq 1$ is obtained through Algorithm 2.

*Remark* B.7. Assumption B.6 can hold. Indeed, Assumption B.6 is satisfied by the K-nearest neighbor imputation with uniform weights, where the imputed value equals the mean of all feature-wise values from the $K$ neighbors.

**Assumption B.8.** Assumptions on $G$ and $\ell$ for Theorem 5.1 Let $\ell$ be a convex pointwise loss such that $\nabla \ell$ is Lipschitz-continuous and $\beta \in [0, 1]$. Define $\mathcal{G}$ as in equation 2. Also define $\boldsymbol{H}(\mathcal{G}; \boldsymbol{\alpha}, \boldsymbol{\alpha}') = \int_0^1 \nabla \mathcal{G}(\boldsymbol{\alpha}, X; \beta)^\intercal \nabla^2 \mathcal{G}(\boldsymbol{\alpha} + a(\boldsymbol{\alpha}' - \boldsymbol{\alpha}), X; \beta) da$. Let $\boldsymbol{g_1} = \nabla_\alpha (1 - \beta) G(\boldsymbol{\alpha})$, $\boldsymbol{g_2} = -\nabla_\alpha \frac{\beta}{N} \sum_{i \leq N} \ell(\boldsymbol{x_i}(\boldsymbol{\alpha}))$, $\phi_{12}$

be the angle between $g_1$ and $g_2$, and $g = g_1 + g_2$. Let $\alpha^t$ be the updated value of $\alpha$ and let $\lambda$ be the step-size for this update. Lemma F.1 shows that $-\mathcal{G}$ is also Lipschitz-continuous w.r.t. $\alpha$, so let this Lipschitz constant be $L$. We assume that there exists $w \le L$ such that $\boldsymbol{H}(-\mathcal{G}, \alpha, \alpha^t) \ge w\|g\|_2^2$. We also assume that $\cos\phi_{12} \le \frac{2\|g_1\|_2\|g_2\|_2}{\|g_1\|_2^2\|g_2\|_2^2}$, $w \ge (1 - \cos^2(\phi_{12})\frac{\|g_1-g_2\|_2^2}{\|g_1+g_2\|_2^2}W$ and $\lambda \ge \frac{2}{w-(1-\cos^2(\phi_{12}))\frac{\|g_1-g_2\|_2^2}{\|g_1+g_2\|_2^2}W}$.

## C   Properties of the objective function G

**Proposition C.1.** *Continuity and derivability of $G$. $G$ is continuous and infinitely derivable with respect to $\alpha \in \triangle_K$.*

*Proof.* $G$ is a composition and sum of indefinitely derivable functions on their respective domains, which are compatible: log on $\mathbb{R}^{+*}$, exp on $\mathbb{R}$ of image domain $\mathbb{R}^{+*}$, $\|\cdot\|_2$ and the linear imputation model (Algorithm 2) on $\mathbb{R}$ with image domain $\mathbb{R}$. $\qquad\square$

**Proposition C.2.** *Strict concavity of $G$ in $\alpha$.  Assume that $\eta < 4KN$. Then there exists $h_0 > 0$ such that for all $h \ge h_0$, $G$ is strictly concave in $\alpha$.*

We aim to show that a value of $h_0$ always exists such that, for $h \ge 0$, the Hessian matrix of $G$ with respect to $\alpha$ is negative (semi-)definite. First, we compute the Hessian matrix of $G$.

**Lemma C.3.** *Gradient of $G$ with respect to $\alpha$. The gradient $\nabla_\alpha G(\alpha, X) \in \mathbb{R}^K$ at $\alpha \in \mathbb{R}^K$ and fixed $X \in \mathbb{R}^{N \times F}$ is*

$$\nabla_\alpha G(\alpha, X) = -\sum_{i \le N} \frac{D_0(\boldsymbol{x}_i(\alpha))^{-1}}{2hN^2(\sqrt{2\pi h})^F} \left( \sum_{j \le N} e^{-\frac{1}{4h}\|\boldsymbol{x}_i(\alpha) - (\boldsymbol{x}^0)_j\|_2^2} \left( \boldsymbol{x}_i(\alpha) - (\boldsymbol{x}^0)_j \right) \right)^\intercal \widetilde{Z}^{n_i} - 2\eta\alpha ,$$

*where $n^i \triangleq (n_1^i, n_2^i, \ldots, n_K^i)$ is the set of indices of the $K$-nearest neighbors of $\boldsymbol{x}_i$ among the reference set $Z = \{(\boldsymbol{x}^0)_1, (\boldsymbol{x}^0)_2, \ldots, (\boldsymbol{x}^0)_N\}$, $\widetilde{Z}^{n_i} \in \mathbb{R}^{F \times K}$ where the $k^{th}$ column of $\widetilde{Z}^{n_i}$ is defined as $(\widetilde{z}^{n_i})_k^f = 0$ if $m_i^f = 0$, $(\widetilde{z}^{n_i})_k^f = (x^0)_{n_k^i}^f$ otherwise. That is, $(\widetilde{z}^{n_i})_k$ is equal to the $k^{th}$ closest neighbor of $\boldsymbol{x}_i$ (by increasing order of distance) on missing coordinates of $\boldsymbol{x}_i$, and equal to zero otherwise.*

*Proof.* The gradient of $G$ at $\alpha \in \triangle_K$ for a fixed $X \in \mathbb{R}^{N \times F}$ is

$$\nabla_\alpha G(\alpha, X) = \sum_{i \le N} \frac{D_0(\boldsymbol{x}_i(\alpha))^{-1}}{N} \nabla_\alpha D_0(\boldsymbol{x}_i(\alpha)) - 0 - 2\eta\alpha$$

$$\nabla_\alpha D_0(\boldsymbol{x}_i(\alpha)) = -\frac{1}{4hN(\sqrt{2\pi h})^F} \sum_{j \le N} e^{-\frac{1}{4h}\|\boldsymbol{x}_i(\alpha) - (\boldsymbol{x}^0)_j\|_2^2} \nabla_\alpha \|\boldsymbol{x}_i(\alpha) - (\boldsymbol{x}^0)_j\|_2^2$$

$$\nabla_\alpha \|\boldsymbol{x}_i(\alpha) - (\boldsymbol{x}^0)_j\|_2^2 = 2 \left( \boldsymbol{x}_i(\alpha) - (\boldsymbol{x}^0)_j \right) \nabla_\alpha \boldsymbol{x}_i(\alpha) \text{ and } \nabla_\alpha \boldsymbol{x}_i(\alpha) = \widetilde{Z}^{n_i} .$$

$\qquad\square$

**Lemma C.4.** *Hessian matrix of $G$ with respect to $\alpha$. Let us denote for any $i, j \le N$ and $\alpha \in \triangle_K$*

- $u_\alpha^{ij} \triangleq e^{-\frac{1}{4h}\|\boldsymbol{x}_i(\alpha) - (\boldsymbol{x}^0)_j\|_2^2}$ and $U_\alpha^i \triangleq \sum_{j \le N} u_\alpha^{ij} = N(\sqrt{2\pi h})^F D_0(\boldsymbol{x}_i(\alpha))$,

- $S_\alpha^i \triangleq \sum_{j \le N} u_\alpha^{ij}(\boldsymbol{x}_i(\alpha) - (\boldsymbol{x}^0)_j)$ and $T_\alpha^i \triangleq \sum_{j \le N} u_\alpha^{ij}(\|\boldsymbol{x}_i(\alpha) - (\boldsymbol{x}^0)_j\|_2^2 - 2h)$.

*Then the covariate at position $(k, q)$ of Hessian matrix $\nabla_\alpha^2 G(\alpha, X) \in \mathbb{R}^{K \times K}$ at $\alpha$ and fixed $X$ is*

$$\frac{\partial^2 G(\alpha, X)}{\partial\alpha_k \partial\alpha_q} = \sum_{i \le N} \left( \frac{T_\alpha^i}{NU_\alpha^i} - \frac{(S_\alpha^i)^\intercal S_\alpha^i}{4h^2(U_\alpha^i)^2} \right) (\widetilde{z}^{n_i})_q^\intercal (\widetilde{z}^{n_i})_k - \eta\mathbb{1}(q = k) .$$

*Proof.* According to Lemma C.3, for any $k \leq K$

$$\frac{\partial G(\boldsymbol{\alpha}, X)}{\partial \alpha_k} = -\sum_{i \leq N} \frac{D_0(\boldsymbol{x}_i(\boldsymbol{\alpha}))^{-1}}{2hN^2(\sqrt{2\pi}h)^F}(S_{\boldsymbol{\alpha}}^i)^\intercal (\widetilde{\boldsymbol{z}}^{n_i})_k - 2\eta\alpha_k .$$

This implies that

$$\frac{\partial^2 G(\boldsymbol{\alpha}, X)}{\partial \alpha_k \partial \alpha_q} + 2\eta \mathbb{1}(q = k) = \sum_{i \leq N} \frac{-D_0(\boldsymbol{x}_i(\boldsymbol{\alpha}))^{-1}}{2hN^2(\sqrt{2\pi}h)^F}\left(\left(\frac{\partial S_{\boldsymbol{\alpha}}^i}{\partial \alpha_q}\right)^\intercal (\widetilde{\boldsymbol{z}}^{n_i})_k\right.$$
$$\left. - \frac{(S_{\boldsymbol{\alpha}}^i)^\intercal (\widetilde{\boldsymbol{z}}^{n_i})_k}{D_0(\boldsymbol{x}_i(\boldsymbol{\alpha}))}\frac{\partial D_0(\boldsymbol{x}_i(\boldsymbol{\alpha}))}{\partial \alpha_q}\right)$$

And then

$$\frac{\partial S_{\boldsymbol{\alpha}}^i}{\partial \alpha_q} = \frac{-1}{4h}\sum_{j \leq N} e^{-\frac{1}{4h}\|\boldsymbol{x}_i(\boldsymbol{\alpha})-(\boldsymbol{x}^0)_j\|_2^2}\left(\frac{\partial \|\boldsymbol{x}_i(\boldsymbol{\alpha})-(\boldsymbol{x}^0)_j\|_2^2}{\partial \alpha_q}\right)^\intercal (\boldsymbol{x}_i(\boldsymbol{\alpha})-(\boldsymbol{x}^0)_j)$$
$$+ \sum_{j \leq N} e^{-\frac{1}{4h}\|\boldsymbol{x}_i(\boldsymbol{\alpha})-(\boldsymbol{x}^0)_j\|_2^2}\frac{\partial \boldsymbol{x}_i(\boldsymbol{\alpha})}{\partial \alpha_q}$$
$$= \frac{-1}{2h}\sum_{j \leq N} e^{-\frac{1}{4h}\|\boldsymbol{x}_i(\boldsymbol{\alpha})-(\boldsymbol{x}^0)_j\|_2^2}(\widetilde{\boldsymbol{z}}^{n_i})_q^\intercal (\boldsymbol{x}_i(\boldsymbol{\alpha})-(\boldsymbol{x}^0)_j)^\intercal (\boldsymbol{x}_i(\boldsymbol{\alpha})-(\boldsymbol{x}^0)_j)$$
$$+ \sum_{j \leq N} e^{-\frac{1}{4h}\|\boldsymbol{x}_i(\boldsymbol{\alpha})-(\boldsymbol{x}^0)_j\|_2^2}(\widetilde{\boldsymbol{z}}^{n_i})_q^\intercal$$

That is,

$$\frac{\partial S_{\boldsymbol{\alpha}}^i}{\partial \alpha_q} = (\widetilde{\boldsymbol{z}}^{n_i})_q^\intercal \underbrace{\left(\sum_{j \leq N} e^{-\frac{1}{4h}\|\boldsymbol{x}_i(\boldsymbol{\alpha})-(\boldsymbol{x}^0)_j\|_2^2}\left(1 - (2h)^{-1}\|\boldsymbol{x}_i(\boldsymbol{\alpha})-(\boldsymbol{x}^0)_j\|_2^2\right)\right)}_{=-2hT_{\boldsymbol{\alpha}}^i}$$

$$\frac{\partial D_0(\boldsymbol{x}_i(\boldsymbol{\alpha}))}{\partial \alpha_q} = -\frac{(S_{\boldsymbol{\alpha}}^i)^\intercal (\widetilde{\boldsymbol{z}}^{n_i})_q}{2hN(\sqrt{2\pi}h)^F} \text{ according to Lemma C.3 .}$$

Moreover, since $S_{\boldsymbol{\alpha}}^i, (\widetilde{\boldsymbol{z}}^{n_i})_k \in \mathbb{R}^F$ for any $k \leq K$

$$(S_{\boldsymbol{\alpha}}^i)^\intercal (\widetilde{\boldsymbol{z}}^{n_i})_k (S_{\boldsymbol{\alpha}}^i)^\intercal (\widetilde{\boldsymbol{z}}^{n_i})_q = \underbrace{(S_{\boldsymbol{\alpha}}^i)^\intercal (\widetilde{\boldsymbol{z}}^{n_i})_q}_{=(\widetilde{\boldsymbol{z}}^{n_i})_q^\intercal S_{\boldsymbol{\alpha}}^i}(S_{\boldsymbol{\alpha}}^i)^\intercal (\widetilde{z}^{n^i})_k = (\widetilde{\boldsymbol{z}}^{n_i})_q^\intercal (S_{\boldsymbol{\alpha}}^i)^\intercal S_{\boldsymbol{\alpha}}^i (\widetilde{\boldsymbol{z}}^{n_i})_k .$$

$\square$

Then, to show that $G$ is (strictly) concave, it is enough to show that the Hessian matrix $\nabla_{\boldsymbol{\alpha}}^2 G(\boldsymbol{\alpha}, X)$ is negative semi-definite (or definite). We assume that $\eta < 4S^2 K = 4K$ (using Assumption B.6), which is the case for most realistic settings.

*Proof.* Let us denote $\boldsymbol{x}_{\boldsymbol{\alpha}}^{ij} \triangleq x_{\boldsymbol{\alpha}}^i - (\boldsymbol{x}^0)_j$ for any $i, j \leq N$. Then

$$\frac{T_{\boldsymbol{\alpha}}^i}{NU_{\boldsymbol{\alpha}}^i} - \frac{(S_{\boldsymbol{\alpha}}^i)^\intercal S_{\boldsymbol{\alpha}}^i}{4h^2(U_{\boldsymbol{\alpha}}^i)^2} = \frac{T_{\boldsymbol{\alpha}}^i}{NU_{\boldsymbol{\alpha}}^i} - \frac{1}{4h^2}\left(\sum_{j,j' \leq N}\frac{u_{\boldsymbol{\alpha}}^{ij}}{U_{\boldsymbol{\alpha}}^i}\frac{u_{\boldsymbol{\alpha}}^{ij'}}{U_{\boldsymbol{\alpha}}^i}(\boldsymbol{x}_{\boldsymbol{\alpha}}^{ij})^\intercal \boldsymbol{x}_{\boldsymbol{\alpha}}^{ij'}\right) . \tag{4}$$

Now consider $(\boldsymbol{x}_{\boldsymbol{\alpha}}^{ij})^\intercal \boldsymbol{x}_{\boldsymbol{\alpha}}^{ij'} = \langle \boldsymbol{x}_i(\boldsymbol{\alpha})-(\boldsymbol{x}^0)_j, \ \boldsymbol{x}_i(\boldsymbol{\alpha})-(\boldsymbol{x}^0)_{j'}\rangle$. From the triangle equality, we have, for all $j, j' \leq N$

$$\langle \boldsymbol{x}_i(\boldsymbol{\alpha})-(\boldsymbol{x}^0)_j, \ \boldsymbol{x}_i(\boldsymbol{\alpha})-(\boldsymbol{x}^0)_{j'}\rangle = \frac{1}{2}(\|\boldsymbol{x}_i(\boldsymbol{\alpha})-(\boldsymbol{x}^0)_j\|_2^2 + \|\boldsymbol{x}_i(\boldsymbol{\alpha})-(\boldsymbol{x}^0)_{j'}\|_2^2 - \|(\boldsymbol{x}^0)_j-(\boldsymbol{x}^0)_{j'}\|_2^2) .$$

We plug this inequality into equation 4

$$\frac{T_{\boldsymbol{\alpha}}^i}{NU_{\boldsymbol{\alpha}}^i} - \frac{(S_{\boldsymbol{\alpha}}^i)^\intercal S_{\boldsymbol{\alpha}}^i}{4h^2(U_{\boldsymbol{\alpha}}^i)^2} = \frac{T_{\boldsymbol{\alpha}}^i}{NU_{\boldsymbol{\alpha}}^i} + \frac{1}{8h^2} \sum_{j,j' \leq N} \frac{u_{\boldsymbol{\alpha}}^{ij}}{U_{\boldsymbol{\alpha}}^i} \frac{u_{\boldsymbol{\alpha}}^{ij'}}{U_{\boldsymbol{\alpha}}^i} \|(\boldsymbol{x}^0)_j - (\boldsymbol{x}^0)_{j'}\|_2^2$$

$$\underbrace{- \frac{1}{8h^2} \sum_{j,j' \leq N} \frac{u_{\boldsymbol{\alpha}}^{ij}}{U_{\boldsymbol{\alpha}}^i} \frac{u_{\boldsymbol{\alpha}}^{ij'}}{U_{\boldsymbol{\alpha}}^i} (\|\boldsymbol{x}_i(\boldsymbol{\alpha}) - (\boldsymbol{x}^0)_j\|_2^2 + \|\boldsymbol{x}_i(\boldsymbol{\alpha}) - (\boldsymbol{x}^0)_{j'}\|_2^2)}_{\geq 0}$$

$$\leq \frac{T_{\boldsymbol{\alpha}}^i}{NU_{\boldsymbol{\alpha}}^i} + \frac{1}{8h^2} \sum_{j,j' \leq N} \frac{u_{\boldsymbol{\alpha}}^{ij}}{U_{\boldsymbol{\alpha}}^i} \frac{u_{\boldsymbol{\alpha}}^{ij'}}{U_{\boldsymbol{\alpha}}^i} \|(\boldsymbol{x}^0)_j - (\boldsymbol{x}^0)_{j'}\|_2^2 .$$

Obviously $\frac{u_{\boldsymbol{\alpha}}^{ij}}{U_{\boldsymbol{\alpha}}^i} \leq 1$. Moreover, since Assumption B.6 gives $\|(\boldsymbol{x}^0)_j\|_2^2 \leq S$ and $\|\boldsymbol{x}_j(\boldsymbol{\alpha})\|_2^2 \leq S$ (using Jensen's inequality) for any $j \leq N$,

$$\frac{T_{\boldsymbol{\alpha}}^i}{NU_{\boldsymbol{\alpha}}^i} - \frac{(S_{\boldsymbol{\alpha}}^i)^\intercal S_{\boldsymbol{\alpha}}^i}{4h^2(U_{\boldsymbol{\alpha}}^i)^2} \leq \frac{T_{\boldsymbol{\alpha}}^i}{NU_{\boldsymbol{\alpha}}^i} + \frac{2N^2S}{8h^2} = \frac{T_{\boldsymbol{\alpha}}^i}{NU_{\boldsymbol{\alpha}}^i} + \frac{N^2S}{4h^2}$$

$$= \sum_{j \leq N} \frac{u_{\boldsymbol{\alpha}}^{ij}}{NU_{\boldsymbol{\alpha}}^i} (\|\boldsymbol{x}^i(\boldsymbol{\alpha}) - (\boldsymbol{x}^0)_j\|_2^2 - 2h) + \frac{NS}{4h^2}$$

$$\leq \frac{1}{N} \sum_{j \leq N} \|\boldsymbol{x}^i(\boldsymbol{\alpha}) - (\boldsymbol{x}^0)_j\|_2^2 - 2h + \frac{N^2S}{4h^2}$$

$$\leq \frac{1}{N} \sum_{j \leq N} (\|\boldsymbol{x}^i(\boldsymbol{\alpha})\|_2^2 + \|(\boldsymbol{x}^0)_j\|_2^2) - 2h + \frac{N^2S}{4h^2}$$

$$\leq 2S - 2h + N^2S(4h^2)^{-1} .$$

We set $C(h) \triangleq h^{-2}(-2h^3 + 2Sh^2 + N^2S/4)$, and fix $\boldsymbol{v} \in \mathbb{R}^K$. Then

$$\boldsymbol{v}^\intercal \nabla_{\boldsymbol{\alpha}}^2 G(\boldsymbol{\alpha}, X) \boldsymbol{v} = -\eta \|\boldsymbol{v}\|_2^2 + \sum_{i \leq N} \left( \frac{T_{\boldsymbol{\alpha}}^i}{NU_{\boldsymbol{\alpha}}^i} - \frac{(S_{\boldsymbol{\alpha}}^i)^\intercal S_{\boldsymbol{\alpha}}^i}{4h^2(U_{\boldsymbol{\alpha}}^i)^2} \right) (\boldsymbol{v}^\intercal (\widetilde{Z}^{n_i})^\intercal \widetilde{Z}^{n_i} \boldsymbol{v})$$

$$\leq -\eta \|\boldsymbol{v}\|_2^2 + C(h) \sum_{i \leq N} \|\widetilde{Z}^{n_i} \boldsymbol{v}\|_2^2$$

so, using Technical lemma 1 (proven below),

$$\boldsymbol{v}^\intercal \nabla_{\boldsymbol{\alpha}}^2 G(\boldsymbol{\alpha}, X) \boldsymbol{v} \leq \|\boldsymbol{v}\|_2^2 (2KSC(h) - \eta) = \|\boldsymbol{v}\|_2^2 \times h^{-2} \left( -4KSh^3 + (4S^2K - \eta)h^2 + \frac{KN^2S^2}{2} \right) .$$

Then we choose $h > 0$ such that $\boldsymbol{v}^\intercal \nabla_{\boldsymbol{\alpha}}^2 G(\boldsymbol{\alpha}, X) \boldsymbol{v} < 0$. That is

$$-4KSh^3 + (4S^2K - \eta)h^2 + \frac{KN^2S^2}{2} < 0 \Leftrightarrow -2h^3 + \frac{4S^2K - \eta}{2KS}h^2 + \frac{N^2S}{4} < 0 . \tag{5}$$

Under the assumption of $\eta < 4S^2K$, this is equivalent to analyzing the following cubic equation

$$-2h^3 + bh^2 + c = 0 \text{ where } b, c > 0 .$$

The cubic equation above admits three roots and at least one real root. We show that at least one real root is positive (thus, corresponds to a valid bandwidth). To show that there exists $h > 0$ such that $-2h^3 + bh^2 + c < 0$, it is enough to show that there exists $h' \in \mathbb{R}$ such that on $[h', +\inf)$, continuous and infinitely derivable function $x \mapsto -2x^3 + bx^2 + c$ is strictly decreasing. We have $\frac{\mathrm{d}}{\mathrm{d}h}(-2h^3 + bh^2 + c) = -6h^2 + 2bh$ with roots 0

and $b/3$, and $\frac{\mathrm{d}^2}{\mathrm{d}^2 h}(-2h^3 + bh^2 + c) = -12h + 2b$, and then $-12 \times 0 + 2b = 2b > 0$ and $-12 \times \frac{b}{3} + 2b = -2b < 0$. The analysis of the behavior of $x \mapsto -2x^3 + bx^2 + c$ then shows that the condition is fulfilled for $h' = b/3 > 0$.

Finally, the value of $h$ can be found through the known closed-form expressions of roots of the rightmost cubic polynomial in $h$ in equation 5. $\qquad\square$

**Proposition C.5.** The gradient of $G(\cdot, X)$ for any $X \in \mathbb{R}^{N \times F}$ is Lipschitz-continuous. *There exists a positive constant $H$ such that*

$$\|\nabla_{\boldsymbol{\alpha}} G(\boldsymbol{\alpha}, X) - \nabla_{\boldsymbol{\alpha}} G(\boldsymbol{\alpha}', X)\|_2 \leq H \|\boldsymbol{\alpha} - \boldsymbol{\alpha}'\|_2 \,.$$

*Proof.* According to Lemma C.3, and using notation from Lemma C.4, for any $X \in \mathbb{R}^{N \times F}$ and $\boldsymbol{\alpha} \in \triangle_K$

$$\nabla_{\boldsymbol{\alpha}} G(\boldsymbol{\alpha}, X) = -\frac{1}{2hN} \sum_{i,j \leq N} \frac{u_{\boldsymbol{\alpha}}^{ij}}{U_{\boldsymbol{\alpha}}^i} \left( \boldsymbol{x}_i(\boldsymbol{\alpha}) - (\boldsymbol{x}^0)_j \right)^{\mathsf{T}} \widetilde{Z}^{n_i} - 2\eta \boldsymbol{\alpha} \,.$$

Then for any $\boldsymbol{\alpha}, \boldsymbol{\alpha}' \in \triangle_K$

$$\|\nabla_{\boldsymbol{\alpha}} G(\boldsymbol{\alpha}, X) - \nabla_{\boldsymbol{\alpha}} G(\boldsymbol{\alpha}', X)\|_2$$

$$= \| - \frac{1}{2hN} \sum_{i,j \leq N} \left( \frac{u_{\boldsymbol{\alpha}}^{ij}}{U_{\boldsymbol{\alpha}}^i} \left( \boldsymbol{x}_i(\boldsymbol{\alpha}) - (\boldsymbol{x}^0)_j \right) - \frac{u_{\boldsymbol{\alpha}'}^{ij}}{U_{\boldsymbol{\alpha}'}^i} \left( \boldsymbol{x}_i(\boldsymbol{\alpha}') - (\boldsymbol{x}^0)_j \right) \right)^{\mathsf{T}} \widetilde{Z}^{n_i} - 2\eta(\boldsymbol{\alpha} - \boldsymbol{\alpha}') \|_2$$

$$\leq \frac{1}{2hN} \sum_{i \leq N} \| \sum_{j \leq N} \left( \frac{u_{\boldsymbol{\alpha}}^{ij}}{U_{\boldsymbol{\alpha}}^i} \left( \boldsymbol{x}_i(\boldsymbol{\alpha}) - (\boldsymbol{x}^0)_j \right) - \frac{u_{\boldsymbol{\alpha}'}^{ij}}{U_{\boldsymbol{\alpha}'}^i} \left( \boldsymbol{x}_i(\boldsymbol{\alpha}') - (\boldsymbol{x}^0)_j \right) \right)^{\mathsf{T}} \widetilde{Z}^{n_i} \|_2 + 2\eta \|\boldsymbol{\alpha} - \boldsymbol{\alpha}'\|_2$$

$$\leq \frac{1}{2hN} \sum_{i \leq N} \| \sum_{j \leq N} \frac{u_{\boldsymbol{\alpha}}^{ij}}{U_{\boldsymbol{\alpha}}^i} \left( \boldsymbol{x}_i(\boldsymbol{\alpha}) - (\boldsymbol{x}^0)_j \right) - \frac{u_{\boldsymbol{\alpha}'}^{ij}}{U_{\boldsymbol{\alpha}'}^i} \left( \boldsymbol{x}_i(\boldsymbol{\alpha}') - (\boldsymbol{x}^0)_j \right) \|_2 \sqrt{2KS} + 2\eta \|\boldsymbol{\alpha} - \boldsymbol{\alpha}'\|_2$$

(using the Cauchy-Schwartz inequality and Technical lemma 1)

$$= \frac{\sqrt{KS}}{\sqrt{2}hN} \sum_{i \leq N} \| \sum_{j \leq N} \frac{u_{\boldsymbol{\alpha}}^{ij}}{U_{\boldsymbol{\alpha}}^i} \boldsymbol{x}_i(\boldsymbol{\alpha}) - \frac{u_{\boldsymbol{\alpha}'}^{ij}}{U_{\boldsymbol{\alpha}'}^i} \boldsymbol{x}_i(\boldsymbol{\alpha}') + \left( \frac{u_{\boldsymbol{\alpha}'}^{ij}}{U_{\boldsymbol{\alpha}'}^i} - \frac{u_{\boldsymbol{\alpha}}^{ij}}{U_{\boldsymbol{\alpha}}^i} \right)(\boldsymbol{x}^0)_j \|_2 + 2\eta \|\boldsymbol{\alpha} - \boldsymbol{\alpha}'\|_2$$

$$\stackrel{U_{\boldsymbol{\alpha}}^i = \sum_j u_{\boldsymbol{\alpha}}^{ij}}{=} \frac{\sqrt{KS}}{\sqrt{2}hN} \sum_{i \leq N} \| \boldsymbol{x}_i(\boldsymbol{\alpha}) - \boldsymbol{x}_i(\boldsymbol{\alpha}') + \sum_{j \leq N} \left( \frac{u_{\boldsymbol{\alpha}'}^{ij}}{U_{\boldsymbol{\alpha}'}^i} - \frac{u_{\boldsymbol{\alpha}}^{ij}}{U_{\boldsymbol{\alpha}}^i} \right)(\boldsymbol{x}^0)_j \|_2 + 2\eta \|\boldsymbol{\alpha} - \boldsymbol{\alpha}'\|_2$$

$$\leq \frac{\sqrt{KS}}{\sqrt{2}hN} \sum_{i \leq N} \left( \sqrt{S} \|\boldsymbol{\alpha} - \boldsymbol{\alpha}'\|_2 + \| \sum_{j \leq N} \left( \frac{u_{\boldsymbol{\alpha}'}^{ij}}{U_{\boldsymbol{\alpha}'}^i} - \frac{u_{\boldsymbol{\alpha}}^{ij}}{U_{\boldsymbol{\alpha}}^i} \right)(\boldsymbol{x}^0)_j \|_2 \right) + 2\eta \|\boldsymbol{\alpha} - \boldsymbol{\alpha}'\|_2$$

$$\leq \frac{S\sqrt{K}}{\sqrt{2}hN} \sum_{i \leq N} \left( \|\boldsymbol{\alpha} - \boldsymbol{\alpha}'\|_2 + \sqrt{2F} \sqrt{\sum_{j \leq N} \left( \frac{u_{\boldsymbol{\alpha}'}^{ij}}{U_{\boldsymbol{\alpha}'}^i} - \frac{u_{\boldsymbol{\alpha}}^{ij}}{U_{\boldsymbol{\alpha}}^i} \right)^2} \right) + 2\eta \|\boldsymbol{\alpha} - \boldsymbol{\alpha}'\|_2 \,,$$

(using Cauchy-Schwartz, Assumption B.6, and $\|X^0\|_F \leq \sqrt{2FS}$)

All that remains is to show that $f_i : \boldsymbol{\alpha} \in \triangle_K \mapsto (\frac{u_{\boldsymbol{\alpha}}^{ij}}{U_{\boldsymbol{\alpha}}^i})_{j \leq N} \in \triangle_N$ (which is a bounded space) is Lipschitz-continuous in $\boldsymbol{\alpha}$. For starters, if all coordinates of $f_i$ $f_{i,j} : \boldsymbol{\alpha} \in \triangle_K \mapsto u_{\boldsymbol{\alpha}}^{ij}/U_{\boldsymbol{\alpha}}^i \in [0, 1]$ are Lipschitz-continuous, each with constant $L_j$, then it is easy to show that $f_i$ is Lipschitz-continuous (always with respect to the $\ell_2$-norm) with constant $L = \sqrt{\sum_{j \leq N} L_j^2}$. Let's consider now any $j \leq N$. For any pair of points $\boldsymbol{\alpha}_1, \boldsymbol{\alpha}_2 \in \triangle_K$, let's introduce the linear path $\gamma : t \in [0, 1] \mapsto t\boldsymbol{\alpha}_1 + (1 - t)\boldsymbol{\alpha}_2$. $f_{i,j} \circ \gamma$ is well-defined, continuous on the closed space $[0, 1]$, differentiable on $(0, 1)$, then by the mean-value theorem

$$f_{i,j}(\boldsymbol{\alpha}_1) - f_{i,j}(\boldsymbol{\alpha}_2) = f_{i,j} \circ \gamma(1) - f_{i,j} \circ \gamma(0) \leq \sup_{t' \in [0,1]} \underbrace{\nabla_t f_{i,j} \circ \gamma(t')}_{=(\nabla_{\boldsymbol{\alpha}} f_{i,j})(\gamma(t'))^{\mathsf{T}}(\boldsymbol{\alpha}_1 - \boldsymbol{\alpha}_2)} (1 - 0) \,. \qquad$$

Meaning that, using the Cauchy-Schwartz inequality '

$$|f_{i,j}(\boldsymbol{\alpha}_1) - f_{i,j}(\boldsymbol{\alpha}_2)| \leq \| \sup_{t \in [0,1]} (\nabla_{\boldsymbol{\alpha}} f_{i,j})(\gamma(t)) \|_2 \|\boldsymbol{\alpha}_1 - \boldsymbol{\alpha}_2\|_2 \leq \underbrace{\| \sup_{\boldsymbol{\alpha} \in \triangle_K} \nabla_{\boldsymbol{\alpha}} f_{i,j}(\boldsymbol{\alpha}) \|_2}_{=L_j} \|\boldsymbol{\alpha}_1 - \boldsymbol{\alpha}_2\|_2 \ .$$

Let's show that the value of $L_j$ is bounded (we are not interested in finding the tightest value of $L_j$, simply that $L_j < \infty$). Then for any $\boldsymbol{\alpha} \in \triangle_K$

$$\nabla_{\boldsymbol{\alpha}} f_{i,j}(\boldsymbol{\alpha}) = \left( -\frac{u_{\boldsymbol{\alpha}}^{ij}}{2hU_{\boldsymbol{\alpha}}^i}(\boldsymbol{x}_i(\boldsymbol{\alpha}) - (\boldsymbol{x}^0)_j) - \frac{u_{\boldsymbol{\alpha}}^{ij}}{(U_{\boldsymbol{\alpha}}^i)^2} \sum_{\ell \leq N} \frac{-1}{2h} u_{\boldsymbol{\alpha}}^{i\ell}(\boldsymbol{x}_i(\boldsymbol{\alpha}) - (\boldsymbol{x}^0)_\ell) \right)^{\top} \widetilde{Z}^{n_i}$$

$$= -\frac{u_{\boldsymbol{\alpha}}^{ij}}{2hU_{\boldsymbol{\alpha}}^i} \left( (\boldsymbol{x}_i(\boldsymbol{\alpha}) - (\boldsymbol{x}^0)_j) - \sum_{\ell \leq N} \frac{u_{\boldsymbol{\alpha}}^{i\ell}}{U_{\boldsymbol{\alpha}}^i}(\boldsymbol{x}_i(\boldsymbol{\alpha}) - (\boldsymbol{x}^0)_\ell) \right)^{\top} \widetilde{Z}^{n_i} \ ,$$

Using the Cauchy-Schwartz inequality and Technical lemma 1

$$\|\nabla_{\boldsymbol{\alpha}} f_{i,j}(\boldsymbol{\alpha})\|_2 \leq \frac{1}{2h} \underbrace{\left| \frac{u_{\boldsymbol{\alpha}}^{ij}}{U_{\boldsymbol{\alpha}}^i} \right|}_{\leq 1} \left\| (\boldsymbol{x}_i(\boldsymbol{\alpha}) - (\boldsymbol{x}^0)_j) - \sum_{\ell \leq N} \frac{u_{\boldsymbol{\alpha}}^{i\ell}}{U_{\boldsymbol{\alpha}}^i}(\boldsymbol{x}_i(\boldsymbol{\alpha}) - (\boldsymbol{x}^0)_\ell) \right\|_2 \sqrt{2KS}$$

Using the triangular inequality on the $\ell_2$-norm

$$\leq \frac{1}{2h} \left( \underbrace{\|\boldsymbol{x}_i(\boldsymbol{\alpha}) - (\boldsymbol{x}^0)_j\|_2}_{\leq \sqrt{2S}} + \sum_{\ell \leq N} \underbrace{\left| \frac{u_{\boldsymbol{\alpha}}^{i\ell}}{U_{\boldsymbol{\alpha}}^i} \right|}_{\leq 1} \underbrace{\|\boldsymbol{x}_i(\boldsymbol{\alpha}) - (\boldsymbol{x}^0)_\ell\|_2}_{\leq \sqrt{2S}} \right) \sqrt{2KS}$$

$$\leq \frac{1}{2h}(N+1)\sqrt{2S}\sqrt{2KS} = \frac{(N+1)}{h}S\sqrt{K} \ .$$

All in all, $f_{i,j}$ is $\frac{(N+1)}{h}S\sqrt{K}$-Lipschitz continuous in $\boldsymbol{\alpha}$ and then $f_i$ is $\frac{(N+1)}{h}S\sqrt{NK}$-Lipschitz continuous in $\boldsymbol{\alpha}$. Finally

$$\|\nabla_{\boldsymbol{\alpha}} G(\boldsymbol{\alpha}, X) - \nabla_{\boldsymbol{\alpha}} G(\boldsymbol{\alpha}', X)\|_2 \leq \frac{S\sqrt{K}}{\sqrt{2}hN} \sum_{i \leq N} \left( \|\boldsymbol{\alpha} - \boldsymbol{\alpha}'\|_2 + \sqrt{2F}\|f_i(\boldsymbol{\alpha}') - f_i(\boldsymbol{\alpha})\|_2 \right)$$

$$+ 2\eta\|\boldsymbol{\alpha} - \boldsymbol{\alpha}'\|_2$$

$$\leq \frac{S\sqrt{K}}{\sqrt{2}hN} \sum_{i \leq N} \left( 1 + \sqrt{2F}\frac{(N+1)}{h}S\sqrt{NK} \right) \|\boldsymbol{\alpha} - \boldsymbol{\alpha}'\|_2$$

$$+ 2\eta\|\boldsymbol{\alpha} - \boldsymbol{\alpha}'\|_2$$

$$\leq \underbrace{\left( \frac{S\sqrt{K}}{\sqrt{2}h} \left( 1 + \sqrt{2F}\frac{(N+1)}{h}S\sqrt{K} \right) + 2\eta \right)}_{\triangleq H} \|\boldsymbol{\alpha} - \boldsymbol{\alpha}'\|_2$$

Then $\nabla_{\boldsymbol{\alpha}} G(\boldsymbol{\alpha}, X)$ is $H$-Lipschitz continuous in $\boldsymbol{\alpha}$ with respect to the $\ell_2$-norm. $\qquad \square$

## D  Bounds on the mean squared error

We recall that the loss function associated with the mean squared error (MSE) is defined as the average of the MSE between each true sample and its corresponding imputed sample at iteration $t$ (Definition 4.1)

$$\mathcal{L}^{\mathrm{MSE}}(X^t, X^\star) \triangleq \frac{1}{N} \sum_{i \leq N} \mathrm{MSE}((\boldsymbol{x}^t)_i, (\boldsymbol{x}^\star)_i) = \frac{1}{N} \sum_{i \leq N} \frac{1}{F} \sum_{f \leq F} ((x^t)_i^f - (x^\star)_i^f)^2 \ .$$

**Theorem D.1.** Bounds in high probability and in expectation on the MSE for F3I (Theorem 4.2). *Under Assumptions B.1-B.6, if $X^t$ is any imputed matrix at iteration $t \geq 1$ (after the initial imputation step), $X^\star$ is the corresponding full (unavailable in practice) matrix*

$$\mathcal{L}^{MSE}(X^t, X^\star) \leq C^{miss}/F \quad \text{with high probability } 1 - 1/N, \text{ where } C^{miss} = \mathcal{O}((\sigma^{miss})^2 F + \ln N) \ ,$$

*and $\sigma^{miss}$ is linked to the variance of the data distribution and depends on the missingness mechanism.*

*Proof.* First, we denote $\mathcal{K}(\boldsymbol{x}, X^0, k)$ the index of the $k^{\mathrm{th}}$ nearest neighbor to vector $\boldsymbol{x}$ among the rows of $X^0$, that is, $\{(\boldsymbol{x}^0)_1, (\boldsymbol{x}^0)_2, \ldots, (\boldsymbol{x}^0)_N\}$. The selection of neighbors does not depend on $f$ after the initial imputation step at $t = 0$. We recall that for any step $t \geq 1$, $(x^t)_i^f \triangleq (x^\star)_i^f$ if $m_i^f = 0$, $\sum_{k \leq K} \alpha_k^t (x^0)_{\mathcal{K}((\boldsymbol{x}^{t-1})_i, X^0, k)}^f$ otherwise. Then, for any $i \leq N$

$$
\begin{aligned}
\mathrm{MSE}(\boldsymbol{x}_i^t, \boldsymbol{x}_i^\star) &= \frac{1}{F} \sum_{f \leq F} \Big( \sum_{k \leq K} \alpha_k^t (x^0)_{\mathcal{K}(x_i^{t-1}, X^0, k)}^f - (x^\star)_i^f \Big)^2 \\
&\leq \frac{1}{F} \sum_{k \leq K} \alpha_k^t \sum_{f \leq F} \Big( (x^0)_{\mathcal{K}(x_i^{t-1}, X^0, k)}^f - (x^\star)_i^f \Big)^2 \\
&\quad \text{(using Jensen's inequality on convex function } x \mapsto x^2 \text{ and } \sum_{k \leq K} \alpha_k^t = 1) \\
&\leq \frac{1}{F} \sum_{k \leq K} \alpha_k^t \| (\boldsymbol{x}^0)_{\mathcal{K}(x_i^{t-1}, X^0, k)} - (\boldsymbol{x}^\star)_i \|_2^2 \ .
\end{aligned}
$$

Applying Corollary G.6 (proven below) with $\delta = 1/N$ and using $\sum_{k \leq K} \alpha_k^t = 1$ yields

$$\forall i \leq N, \quad \mathrm{MSE}(\boldsymbol{x}_i^t, \boldsymbol{x}_i^\star) \leq \frac{1}{F} \times 1 \times C_{1/N^3}^{\mathrm{miss}} \quad \text{w.p.} \quad 1 - 1/N \ .$$

Then we conclude by noticing that $\mathcal{L}^{\mathrm{MSE}}(X^t, X^\star) \leq \frac{N}{N} \frac{1}{F} C_{1/N^3}^{\mathrm{miss}} \quad \text{w.p.} \quad 1 - 1/N.$ $\qquad \square$

## E  Regret analysis of F3I

**Theorem E.1.** High-probability upper bound on the imputation quality for F3I (Theorem 4.4). *Under Assumptions B.1-B.6, for any initial matrix $X \in (\mathbb{R} \cup \{N/A\})^{N \times F}$,*

$$\max_{\boldsymbol{\alpha} \in \triangle_K} \sum_{s=1}^t G_\star(\boldsymbol{\alpha}, X^{s-1}) - G_\star(\boldsymbol{\alpha}^s, X^{s-1}) \quad \leq \quad C_G^{AH} \sqrt{t} + H^{miss} h^{-1} t \ ,$$

*with probability $1 - 1/N$, where $H^{miss} = \mathcal{O}(F + \ln N)$ is another value which depends on the missingness mechanism and $h$ is chosen to guarantee that $G$ is concave in its first argument (Proposition C.2). $C_G^{AH} = \mathcal{O}(\sqrt{\log(K)})$ is the constant associated with the regret bound on the gain in F3I incurred by AdaHedge.*

*Proof.* We set the gain in F3I to $g^s(\boldsymbol{\alpha}) \triangleq \sum_{k \leq K} \alpha_k \frac{\partial G}{\partial \alpha_k}(\boldsymbol{\alpha}^s, X^{s-1})$ for $s \leq t$ and $\boldsymbol{\alpha} \in \triangle_K$. We recall the gradient trick for any convex function $\ell$ of gradient $g_t := \nabla \ell(\boldsymbol{x}_t)$ at a point $\boldsymbol{x}_t$; the following inequality

holds thanks to the convexity of $\ell$: for any point $\boldsymbol{x}$, $\ell(\boldsymbol{x}_t) - \ell(\boldsymbol{x}) \leq \nabla g_t^\mathsf{T}(\boldsymbol{x}_t - \boldsymbol{x})$. We use the gradient trick to transfer the regret bound from a linear loss function (the one used in AdaHedge) to the convex loss $-G(\cdot, X^{t-1})$ (the target loss function) for any $t \geq 1$, for all $\boldsymbol{\alpha}, \boldsymbol{\alpha}^s \in \triangle_K$, $s \leq t$,

$$\sum_{s=1}^{t} G(\boldsymbol{\alpha}, X^{s-1}) - G(\boldsymbol{\alpha}^s, X^{s-1}) \leq \sum_{s=1}^{t} (\boldsymbol{\alpha} - \boldsymbol{\alpha}^s)^\mathsf{T} \nabla_{\boldsymbol{\alpha}} G(\boldsymbol{\alpha}^s, X^{s-1}) \,.$$

and note that for any $\boldsymbol{\alpha} \in \triangle_K$,

$$\sum_{s=1}^{t} g^s(\boldsymbol{\alpha}) - g^s(\boldsymbol{\alpha}^s) = \sum_{s=1}^{t} \boldsymbol{\alpha}^\mathsf{T} \nabla_{\boldsymbol{\alpha}} G(\boldsymbol{\alpha}^s, X^{s-1}) - (\boldsymbol{\alpha}^s)^\mathsf{T} \nabla_{\boldsymbol{\alpha}} G(\boldsymbol{\alpha}^s, X^{s-1})$$

$$= \sum_{s=1}^{t} (\boldsymbol{\alpha} - \boldsymbol{\alpha}^s)^\mathsf{T} \nabla_{\boldsymbol{\alpha}} G(\boldsymbol{\alpha}^s, X^{s-1}) \,.$$

Then applying the regret bound of AdaHedge (Technical lemma 2, proven below) to that gain yields at time $t > 1$ (rightmost term) and the gradient trick on the function $G$ which is concave in its first argument (leftmost term) with Proposition C.2

$$\forall \boldsymbol{\alpha} \in \triangle_K, \ \sum_{s=1}^{t} G(\boldsymbol{\alpha}, X^{s-1}) - G(\boldsymbol{\alpha}^s, X^{s-1}) \leq 2\delta_t \sqrt{t \log(K)} + 16\delta_t \left(2 + \frac{\log K}{3}\right) \,, \tag{6}$$

where $\delta_t \triangleq \max_{s \leq t} \left(\max_{k \leq K} \frac{\partial G}{\partial \alpha_k}(\boldsymbol{\alpha}^s, X^{s-1}) - \min_{q \leq K} \frac{\partial G}{\partial \alpha_q}(\boldsymbol{\alpha}^s, X^{s-1})\right)$. Now we go from $G$ to $G_\star$ point-wise. Corollary G.6 with $\delta = 1/N$ states that under Assumptions B.1-B.6, there exists $C_{1/N^3}^{\mathrm{miss}} = \mathcal{O}(F + \ln N)$ such that for any $i \leq N$, $\|(\boldsymbol{x}^0)_i - (\boldsymbol{x}^\star)_i\|_2^2 \leq C_{1/N^3}^{\mathrm{miss}}$ with probability $1 - 1/N$. By triangle inequality,

$$\forall \boldsymbol{x} \in \mathbb{R}^F \ \forall i \leq N, \ \|\boldsymbol{x} - (\boldsymbol{x}^\star)_i\|_2^2 - \|\boldsymbol{x} - (\boldsymbol{x}^0)_i\|_2^2 \leq \|(\boldsymbol{x}^0)_i - (\boldsymbol{x}^\star)_i\|_2^2 \leq C_{1/N^3}^{\mathrm{miss}} \text{ w.p. } 1 - 1/N \,.$$

Then, with probability $1 - 1/N$,

$$\forall \boldsymbol{x} \in \mathbb{R}^F, \qquad \|\boldsymbol{x} - (\boldsymbol{x}^\star)_i\|_2^2 \qquad \leq \|\boldsymbol{x} - (\boldsymbol{x}^0)_i\|_2^2 + C_{1/N^3}^{\mathrm{miss}}$$

$$\implies \qquad -\tfrac{1}{4h}\|\boldsymbol{x} - (\boldsymbol{x}^\star)_i\|_2^2 \qquad \geq -\frac{1}{4h}\|\boldsymbol{x} - (\boldsymbol{x}^0)_i\|_2^2 - \frac{C_{1/N^3}^{\mathrm{miss}}}{4h}$$

$$\implies \qquad e^{-\frac{1}{4h}\|\boldsymbol{x}-(\boldsymbol{x}^\star)_i\|_2^2} \qquad \geq e^{-\frac{C_{1/N^3}^{\mathrm{miss}}}{4h}} e^{-\frac{1}{4h}\|\boldsymbol{x}-(\boldsymbol{x}^0)_i\|_2^2}$$

$$\implies \qquad \sum_{i \leq N} e^{-\frac{1}{4h}\|\boldsymbol{x}-(\boldsymbol{x}^\star)_i\|_2^2} \qquad \geq e^{-\frac{C_{1/N^3}^{\mathrm{miss}}}{4h}} \sum_{i \leq N} e^{-\frac{1}{4h}\|\boldsymbol{x}-(\boldsymbol{x}^0)_i\|_2^2}$$

$$\implies \quad \log\left(\frac{\sum_{i \leq N} e^{-\frac{1}{4h}\|\boldsymbol{x}-(\boldsymbol{x}^0)_i\|_2^2}}{\sum_{i \leq N} e^{-\frac{1}{4h}\|\boldsymbol{x}-(\boldsymbol{x}^\star)_i\|_2^2}}\right) = \log\left(\frac{D_0(\boldsymbol{x})}{D_\star(\boldsymbol{x})}\right) \leq \frac{C_{1/N^3}^{\mathrm{miss}}}{4h} \,.$$

Symmetrically (by switching the roles of $(\boldsymbol{x}^\star)_i$ and $(\boldsymbol{x}^0)_i$ in the previous inequalities), we obtain with probability $1 - 1/N$

$$\forall \boldsymbol{x} \in \mathbb{R}^F, \ \log\left(\frac{D_\star(\boldsymbol{x})}{D_0(\boldsymbol{x})}\right) \ = \ -\log\left(\frac{D_0(\boldsymbol{x})}{D_\star(\boldsymbol{x})}\right) \leq C_{1/N^3}^{\mathrm{miss}}/(4h)$$

$$\implies \ \left|\log\left(\frac{D_0(\boldsymbol{x})}{D_\star(\boldsymbol{x})}\right)\right| \ \leq \ C_{1/N^3}^{\mathrm{miss}}/(4h) \,.$$

That is, for any $\boldsymbol{\alpha} \in \triangle_K$ and $X \in \mathbb{R}^{N \times F}$, with probability $1 - 1/N$

$$|(G - G_\star)(\boldsymbol{\alpha}, X)| = \frac{1}{N} \sum_{i \leq N} \log\left(\frac{D_0(\texttt{Impute}(\boldsymbol{x}_i; \boldsymbol{\alpha}))}{D_\star(\texttt{Impute}(\boldsymbol{x}_i; \boldsymbol{\alpha}))}\right) - \log\left(\frac{D_0(\boldsymbol{x}_i)}{D_\star(\boldsymbol{x}_i)}\right) \leq \frac{C_{1/N^3}^{\mathrm{miss}}}{2h} \,. \tag{7}$$

Finally, we combine Equations equation 6-equation 7 to obtain for any $\boldsymbol{\alpha} \in \triangle_K$, with probability $1 - 1/N$

$$\sum_{s=1}^{t} G_\star(\boldsymbol{\alpha}, X^{s-1}) - G_\star(\boldsymbol{\alpha}^s, X^{s-1}) \leq \frac{C_{1/N^3}^{\text{miss}}}{h} t + \underbrace{2\delta_t \sqrt{t \log(K)} + 16\delta_t \left(2 + \frac{\log K}{3}\right)}_{=C_G^{\text{AH}}\sqrt{t}} .$$

$\square$

## F Joint training on a downstream task

**Lemma F.1.** Any loss $\ell$ with a Lipschitz continuous gradient allows the use of PCGrad (Yu et al., 2020) combined with F3I. *If $\nabla\ell$ is $L$-Lipschitz continuous with a finite $L > 0$ with respect to its single argument, then for any matrix $X \in \mathbb{R}^{N \times F}$, $\boldsymbol{\alpha} \mapsto \nabla_{\boldsymbol{\alpha}}\left((1-\beta)G(\boldsymbol{\alpha}, X) - \frac{\beta}{N}\sum_{i \leq N} \ell(\texttt{Impute}(\boldsymbol{x}^i, \boldsymbol{\alpha}))\right)$ is also Lipschitz continuous with a positive finite constant.*

*Proof.* Note that Proposition C.5 establishes that the gradient of $G$ with respect to $\boldsymbol{\alpha}$ is $H$-Lipschitz continuous with $H > 0$. Then for all $\boldsymbol{\alpha}, \boldsymbol{\alpha}' \in \triangle_K$

$$\left\| \nabla_{\boldsymbol{\alpha}}\left((1-\beta)G(\boldsymbol{\alpha}, X) + \frac{\beta}{N}\sum_{i \leq N}\ell(\boldsymbol{x}_i(\boldsymbol{\alpha})) - \left((1-\beta)G(\boldsymbol{\alpha}', X) + \frac{\beta}{N}\sum_{i \leq N}\ell(\boldsymbol{x}_i(\boldsymbol{\alpha}'))\right)\right) \right\|_2$$

$$\leq (1-\beta)\|\nabla_{\boldsymbol{\alpha}}G(\boldsymbol{\alpha}, X) - \nabla_{\boldsymbol{\alpha}}G(\boldsymbol{\alpha}', X)\|_2 + \frac{\beta}{N}\sum_{i \leq N}\|\nabla_{\boldsymbol{\alpha}}\ell(\boldsymbol{x}_i(\boldsymbol{\alpha})) - \nabla_{\boldsymbol{\alpha}}\ell(\boldsymbol{x}_i(\boldsymbol{\alpha}'))\|_2$$

$$\leq H(1-\beta)\|\boldsymbol{\alpha} - \boldsymbol{\alpha}'\|_2 + \frac{L\beta}{N}\sum_{i \leq N}\|\boldsymbol{x}_i(\boldsymbol{\alpha}) - \boldsymbol{x}_i(\boldsymbol{\alpha}')\|_2$$

$$\leq (H(1-\beta) - L\beta\sqrt{KS})\|\boldsymbol{\alpha} - \boldsymbol{\alpha}'\|_2 .$$

The last step holds because of the fact that, for any $i \leq N$, if $\mathcal{K}(\boldsymbol{x}_i, X^0, k)$ is the index of the $k^{\text{th}}$ nearest neighbor of $\boldsymbol{x}_i$ among $\{(\boldsymbol{x}^0)_1, \ldots, (\boldsymbol{x}^0)_N\} \subseteq \mathbb{R}^F$ and $\boldsymbol{x}_i^{\mathcal{M}_i}$ is the vector restricted to columns $f$ such that $x_i^f$ is missing

$$\|\boldsymbol{x}_i(\boldsymbol{\alpha}) - \boldsymbol{x}_i(\boldsymbol{\alpha}')\|_2^2 = \left\| \sum_{k \leq K}(\alpha_k - \alpha_k')(x^0)_{\mathcal{K}(\boldsymbol{x}_i, X^0, k)}^{\mathcal{M}_i} \right\|_2^2$$

$$= \sum_{f \in \mathcal{M}_i}\left(\sum_{k \leq K}(\alpha_k - \alpha_k')(x^0)_{\mathcal{K}(x_i, X^0, k)}^f\right)^2$$

$$= \sum_{f \in \mathcal{M}_i}\langle \boldsymbol{\alpha} - \boldsymbol{\alpha}', [(x^0)_{\mathcal{K}(\boldsymbol{x}_i, X^0, 1)}^f, ..., (x^0)_{\mathcal{K}(\boldsymbol{x}_i, X^0, K)}^f]^{\mathsf{T}}\rangle^2$$

$$\leq \|\boldsymbol{\alpha} - \boldsymbol{\alpha}'\|_2^2 \sum_{f \in \mathcal{M}_i}\|[(x^0)_{\mathcal{K}(\boldsymbol{x}_i, X^0, 1)}^f, ..., (x^0)_{\mathcal{K}(\boldsymbol{x}_i, X^0, K)}^f]^{\mathsf{T}}\|_2^2$$

$$= \|\boldsymbol{\alpha} - \boldsymbol{\alpha}'\|_2^2 \sum_{f \in \mathcal{M}_i}\sum_{k \leq K}((x^0)_{\mathcal{K}(\boldsymbol{x}_i, X^0, k)}^f)^2$$

$$\leq \|\boldsymbol{\alpha} - \boldsymbol{\alpha}'\|_2^2 \sum_{k \leq K}\sum_{f \leq F}((x^0)_{\mathcal{K}(\boldsymbol{x}_i, X^0, k)}^f)^2$$

$$= \|\boldsymbol{\alpha} - \boldsymbol{\alpha}'\|_2^2 \sum_{k \leq K}\|(\boldsymbol{x}^0)_{\mathcal{K}(\boldsymbol{x}_i, X^0, k)}\|_2^2$$

$$\leq \|\boldsymbol{\alpha} - \boldsymbol{\alpha}'\|_2^2 KS \text{ using Assumption B.6} .$$

The first inequality is obtained by applying the Cauchy-Schwartz inequality $|\mathcal{M}_i|$ times, since the selection of neighbors does not depend on $\boldsymbol{\alpha}$. Note that $(x_i(\boldsymbol{\alpha}))^f = (x_i(\boldsymbol{\alpha}'))^f$ for any $f \notin \mathcal{M}_i$. $\square$

**Example 1.** A simple example of a convex loss function $\ell$ with a Lipschitz-continuous gradient function. *The pointwise log loss $\ell(\boldsymbol{x}) = -y \log C_{\boldsymbol{\omega}}(\boldsymbol{x})$ for the binary classification task is convex and such that $\nabla_{\boldsymbol{x}}\ell$ is Lipschitz continuous, where $y$ is the true class in $\{0, 1\}$ for sample $\boldsymbol{x}$ and $C_{\boldsymbol{\omega}} : \boldsymbol{x} \mapsto 1/(1 + \exp(-\boldsymbol{\omega}^{\mathsf{T}}\boldsymbol{x}))$ is the sigmoid function of parameter $\boldsymbol{\omega}$.*

*Proof.* $\ell$ is continuous and twice differentiable on $\mathbb{R}^F$. Knowing that $\nabla_{\boldsymbol{x}} C_{\boldsymbol{\omega}}(\boldsymbol{x}) = C_{\boldsymbol{\omega}}(\boldsymbol{x})(1 - C_{\boldsymbol{\omega}}(\boldsymbol{x}))\boldsymbol{\omega}^{\mathsf{T}}$, the Hessian matrix of $\ell$ in its single argument is

$$\forall \boldsymbol{x} \in \mathbb{R}^F \ \forall y \in \{0, 1\}, \ \nabla_{\boldsymbol{x}}^2 \ell(\boldsymbol{x}) = y C_{\boldsymbol{\omega}}(\boldsymbol{x})(1 - C_{\boldsymbol{\omega}}(\boldsymbol{x}))\boldsymbol{\omega}\boldsymbol{\omega}^{\mathsf{T}} \ .$$

In particular, it is easy to see that $\ell$ is convex, because for any $\boldsymbol{v} \in \mathbb{R}^F$,

$$\boldsymbol{v}^{\mathsf{T}} \nabla_{\boldsymbol{x}}^2 \ell(\boldsymbol{x}) \boldsymbol{v} = \underbrace{y C_{\boldsymbol{\omega}}(\boldsymbol{x})(1 - C_{\boldsymbol{\omega}}(\boldsymbol{x}))}_{\geq 0}(\boldsymbol{v}^{\mathsf{T}}\boldsymbol{\omega})^2 \geq 0 \ .$$

Then for any $\boldsymbol{x} \in \mathbb{R}^F$ and $y \in \{0, 1\}$,

$$\|\nabla_{\boldsymbol{x}}^2 \ell(\boldsymbol{x})\|_F^2 \ = \ \underbrace{y C_{\boldsymbol{\omega}}(\boldsymbol{x})(1 - C_{\boldsymbol{\omega}}(\boldsymbol{x}))}_{\leq 1 \times 1/4} \sum_{f, f' \leq F} (\omega^f)^2 \leq \frac{1}{4}\|\boldsymbol{\omega}\|_F^2 \ .$$

Similarly to the proof of Proposition C.5, proving that $\nabla_{\boldsymbol{x}}\ell$ is Lipschitz continuous in each of its $F$ coordinates will be enough to prove that $\nabla_{\boldsymbol{x}}\ell$ is Lipschitz continuous as well. For any pair of points $\boldsymbol{x}_1, \boldsymbol{x}_2 \in \mathbb{R}^F$, we introduce the linear path $\gamma' : t \in [0, 1] \mapsto t\boldsymbol{x}_1 + (1 - t)\boldsymbol{x}_2$. For any $f \leq F$, $t \in [0, 1] \mapsto \left(\nabla_{\boldsymbol{x}}\ell(\gamma'(t))\right)^f \in \mathbb{R}$ is well-defined, continuous on the closed space $[0, 1]$, differentiable on $(0, 1)$. Then applying the mean value theorem to this function yields

$$\left|\left(\nabla_{\boldsymbol{x}}\ell(\boldsymbol{x}_1)\right)^f - \left(\nabla_{\boldsymbol{x}}\ell(\boldsymbol{x}_2)\right)^f\right| \leq \|\sup_{\boldsymbol{x} \in \mathbb{R}^F} \left(\nabla_{\boldsymbol{x}}^2 \ell(\boldsymbol{x})\right)^f\|_2 \|\boldsymbol{x}_1 - \boldsymbol{x}_2\|_2 \leq \frac{\|\boldsymbol{\omega}\|_2}{2}\|\boldsymbol{x}_1 - \boldsymbol{x}_2\|_2 \ .$$

Then $\nabla_{\boldsymbol{x}}\ell$ is Lipschitz-continuous with constant $\sqrt{\sum_{f \leq F} \frac{1}{4}\|\boldsymbol{\omega}\|_2^2} = \frac{\|\boldsymbol{\omega}\|_F^2}{2} > 0$. $\qquad\square$

**Theorem F.2.** High-probability upper bound on the joint imputation-downstream task performance (Theorem 5.1). *Under Assumptions B.1-B.6, for any initial matrix $X \in (\mathbb{R} \cup \{\textit{N/A}\})^{N \times F}$, convex pointwise loss $\ell$ such that $\nabla\ell$ is Lipschitz-continuous, and $\beta \in [0, 1]$, under the conditions mentioned in Theorem 2 from Yu et al. (2020)*

$$\max_{\boldsymbol{\alpha} \in \triangle_K} \sum_{s=1}^{t} (1 - \beta)\Big(G_{\star}(\boldsymbol{\alpha}, X^{s-1}) - G_{\star}(\boldsymbol{\alpha}^s, X^{s-1})\Big) - \frac{\beta}{N}\sum_{i \leq N}\Big(\ell((\boldsymbol{x}^{s-1})_i(\boldsymbol{\alpha})) - \ell((\boldsymbol{x}^{s-1})_i(\boldsymbol{\alpha}^s))\Big)$$
$$\leq C_{(G,\ell)}^{AH}\sqrt{t} + (1 - \beta)H^{miss}h^{-1}t \ ,$$

*with probability $1 - 1/N \in (0, 1)$, where $H^{miss} = \mathcal{O}(F + \ln N)$ depends on the missingness mechanism and $C_{(G,\ell)}^{AH}$ is the constant related to AdaHedge being applied with gains $\overline{g}_s(\cdot)$.*

*Proof.* Similarly to the proof of Theorem 4.4, the application of the AdaHedge regret bound (Technical lemma 2), and the gradient trick on the concave function $(1 - \beta)G(\cdot, X) + \beta\ell(\cdot)$

$$\sum_{s=1}^{t} (1 - \beta)\Big(G(\boldsymbol{\alpha}^{\mathrm{PC}}, X^{s-1}) - G((\boldsymbol{\alpha}^s)^{\mathrm{PC}}, X^{s-1})\Big) + \frac{\beta}{N}\sum_{i \leq N}\Big(\ell((\boldsymbol{x}^{s-1})_i(\boldsymbol{\alpha}^s)^{\mathrm{PC}}) - \ell((\boldsymbol{x}^{s-1})_i(\boldsymbol{\alpha}^{\mathrm{PC}}))\Big)$$
$$\leq C_{(G,\ell)}^{\mathrm{AH}}\sqrt{t} \ ,$$

where $\boldsymbol{\alpha}^{\mathrm{PC}}$ and $(\boldsymbol{\alpha}^s)^{\mathrm{PC}}$ are the parameters updated with PCGrad (Yu et al., 2020). Assuming the three conditions in Theorem 2 from Yu et al. (2020) are all satisfied, which only depend on functions $G$ and $\ell$, then for $\boldsymbol{\theta} \in \{\boldsymbol{\alpha}, \boldsymbol{\alpha}^s\}$

$$(1 - \beta)G(\boldsymbol{\theta}, X^{s-1}) - \frac{\beta}{N}\sum_{i \leq N}\ell((\boldsymbol{x}^{s-1})_i(\boldsymbol{\theta})) \leq (1 - \beta)G(\boldsymbol{\theta}^{\mathrm{PC}}, X^{s-1}) - \frac{\beta}{N}\sum_{i \leq N}\ell((\boldsymbol{x}^{s-1})_i(\boldsymbol{\theta}^{\mathrm{PC}})) .$$

Finally, we apply the pointwise approximation in high probability of $G_\star$ by $G$ (Corollary G.6 for $\delta = 1/N$) that yields for any $\boldsymbol{\alpha} \in \triangle_K$

$$\sum_{s=1}^{t}(1 - \beta)\Big(G_\star(\boldsymbol{\alpha}, X^{s-1}) - G_\star(\boldsymbol{\alpha}^s, X^{s-1})\Big) + \frac{\beta}{N}\sum_{i \leq N}\Big(\ell((\boldsymbol{x}^{s-1})_i(\boldsymbol{\alpha}^s)) - \ell((\boldsymbol{x}^{s-1})_i(\boldsymbol{\alpha}))\Big)$$
$$\leq C_{(G,\ell)}^{\mathrm{AH}}\sqrt{t} + (1 - \beta)C_{1/N^3}^{\mathrm{miss}}h^{-1}t .$$

$\square$

To implement PCGrad-F3I, we also need to compute $\nabla_{\boldsymbol{\alpha}}\ell((\boldsymbol{x}^{s-1})_i(\boldsymbol{\alpha}^s))$ at each iteration $s$ for each point $i$. By the chain rule,

$$\nabla_{\boldsymbol{\alpha}}\ell((\boldsymbol{x}^{s-1})_i(\boldsymbol{\alpha}^s)) = \nabla_{\boldsymbol{x}}\ell(\boldsymbol{x})_{|\boldsymbol{x}=(\boldsymbol{x}^{s-1})_i(\boldsymbol{\alpha}^s)}\nabla_{\boldsymbol{\alpha}}(\boldsymbol{x}^{s-1})_i(\boldsymbol{\alpha})_{|\boldsymbol{\alpha}=\boldsymbol{\alpha}^s} ,$$

In particular, we give the gradient at any $\boldsymbol{\alpha}$ and $i \leq N$ for the log-loss with sigmoid classifier below

**Lemma F.3.** Gradient $\nabla_{\boldsymbol{\alpha}}\ell((\boldsymbol{x}^{s-1})_i(\boldsymbol{\alpha}))$ for Example 1. *The gradient at any $\boldsymbol{\alpha}$ for the log-loss $\ell$ with sigmoid classifier $C_{\boldsymbol{\omega}}$ where the true class for sample $\boldsymbol{x} \in \mathbb{R}^F$ is $y \in \{0, 1\}$ is*

$$\nabla_{\boldsymbol{\alpha}}\ell(\boldsymbol{x}(\boldsymbol{\alpha})) = -y(1 - C_{\boldsymbol{\omega}}(\boldsymbol{x}(\boldsymbol{\alpha})))\boldsymbol{\omega}^{\mathsf{T}}\widetilde{Z}_s^{n_i} ,$$

*where $\widetilde{Z}_s^{n_i} \in \mathbb{R}^{F \times K}$ is the matrix which $k^{th}$ column is defined as $(\widetilde{z}_s^{n_i})_f^k = 0$ if $m_i^f = 0$, and otherwise, $(\widetilde{z}_s^{n_i})_f^k$ is the value of the feature $f$ for the $k^{th}$ closest neighbor to $(\boldsymbol{x}^{s-1})_i$ among rows $(\boldsymbol{x}^0)_2, \ldots, (\boldsymbol{x}^0)_N\}$ of $X^0$ $\{(\boldsymbol{x}^0)_1, (\boldsymbol{x}^0)_2, \ldots, (\boldsymbol{x}^0)_N\}$ (see Lemma C.3).*

# G    Technical lemmas

We consider below for any $i \leq N$ the matrix $\widetilde{Z}^{n_i} \in \mathbb{R}^{F \times K}$ where the $k^{\mathrm{th}}$ column of $\widetilde{Z}^{n_i}$ is defined as $(\widetilde{z}^{n_i})_f^k = 0$ if $m_i^f = 0$, otherwise $(\widetilde{z}^{n_i})_f^k$ is the value of the feature $f$ for the $k^{\mathrm{th}}$ closest neighbor to $(\boldsymbol{x}^{s-1})_i$ among rows of $X^0$ $\{(\boldsymbol{x}^0)_1, (\boldsymbol{x}^0)_2, \ldots, (\boldsymbol{x}^0)_N\}$. That is, $(\widetilde{z}^{n_i})^k$ is equal to the $k^{\mathrm{th}}$ closest neighbor of $\boldsymbol{x}_i$ (by increasing order of distance) on missing coordinates of $\boldsymbol{x}_i$, and equal to zero otherwise. To upper-bound norms involving matrix $\widetilde{Z}^{n_i}$, we use the following lemma

**Technical lemma 1.** Upper bound on $\ell_2$ norms on $\widetilde{Z}^{n_i}$. *For any $i \leq N$ and any vectors $\boldsymbol{v} \in \mathbb{R}^K$ and $\boldsymbol{u} \in \mathbb{R}^F$,*

$$\|\widetilde{Z}^{n_i}\boldsymbol{v}\|_2 \leq \sqrt{2KS}\|\boldsymbol{v}\|_2 \text{ and } \|\boldsymbol{u}^{\mathsf{T}}\widetilde{Z}^{n_i}\|_2 \leq \sqrt{2KS}\|\boldsymbol{u}\|_2 .$$

*Proof.* Using the Cauchy-Schwartz inequality applied respectively $F$ and $K$ times and if $\|M\|_F = \sqrt{\sum_i\sum_j|m_i^j|^2} = \sqrt{\sum_i\|\boldsymbol{m}_i\|_2^2} = \sqrt{\sum_j\|\boldsymbol{m}^j\|_2^2}$ is the Frobenius matrix norm of matrix $M$, then $\|\widetilde{Z}^{n_i}\|_F^2 \leq 2KS$ and

$$\|\widetilde{Z}^{n_i}\boldsymbol{v}\|_2^2 = \sum_{f \leq F}\langle(\widetilde{Z}^{n_i})_f^{\mathsf{T}}, \boldsymbol{v}\rangle^2 \leq \underbrace{\sum_{f \leq F}\|(\widetilde{Z}^{n_i})_f^{\mathsf{T}}\|_2^2}_{=_{\mathrm{def}}\|\widetilde{Z}^{n_i}\|_F^2}\|\boldsymbol{v}\|_2^2 \leq 2KS\|\boldsymbol{v}\|_2^2 .$$

$$\|\boldsymbol{u}^{\mathsf{T}}\widetilde{Z}^{n_i}\|_2^2 = \sum_{k \leq K}\langle\boldsymbol{u}, (\widetilde{Z}^{n_i})^k\rangle^2 \leq \|\boldsymbol{u}\|_2^2\underbrace{\sum_{k \leq K}\|(\widetilde{Z}^{n_i})^k\|_2^2}_{=_{\mathrm{def}}\|\widetilde{Z}^{n_i}\|_F^2} \leq 2KS\|\boldsymbol{u}\|_2^2 .$$

$\square$

**Technical lemma 2.** Regret of AdaHedge. *On the online learning problem with $K$ elements, using gains $\boldsymbol{\alpha} \mapsto g^s(\boldsymbol{\alpha}) \triangleq \sum_{k \leq K} \alpha_k U_k$ for $s \leq t$, and denoting $\delta_t \triangleq \max_{s \leq t} (\max_{k \leq K} U_k - \min_{q \leq K} U_q)$, the regret at time $t > 1$ incurred by AdaHedge with predictions $(\boldsymbol{\alpha}^s)_{s \leq t}$ is*

$$\max_{\boldsymbol{\alpha} \in \triangle_K} \sum_{s=1}^{t} g^s(\boldsymbol{\alpha}) - g^s(\boldsymbol{\alpha}^s) \leq 2\delta_t \sqrt{t \log(K)} + 16\delta_t(2 + \log(K)/3) \, .$$

*Proof.* This statement stems directly from Theorem 8 and Corollary 17 in De Rooij et al. (2014) applied to the loss $\ell^s = -g^s$, and using the fact that $\alpha_k \leq 1$ for any $k \leq K$. □

In several proofs, we need an upper bound on $\|(\boldsymbol{x}^0)_j - (\boldsymbol{x}^\star)_i\|_2^2$ for any pair $i, j \leq N$, where $X^0$ is the initially $K$-nearest neighbor-imputed matrix and $X^\star$ is the corresponding full matrix. This is our most important lemma for analyzing F3I. This result still holds for any missingness mechanism such that the random variable $(x^0)_i^f - (x^\star)_i^f$ is a zero-mean subgaussian for any $i \leq N$ and $f \leq F$, independent across *features*. We show below that this statement includes all three mechanisms mentioned in Assumptions B.2-B.4 as described in Algorithm 3.

**Technical lemma 3.** Concentration bound on the norm of the difference between $(\boldsymbol{x}^0)_j$ and $(\boldsymbol{x}^\star)_i$. *Under any assumption in Assumptions B.2-B.4, if we consider a subset of features $\mathcal{F} \subseteq \{1, 2, \ldots, F\}$ such that $\boldsymbol{x}^{|\mathcal{F}}$ is the restriction of $\boldsymbol{x} \in \mathbb{R}^F$ to features in $\mathcal{F}$, then*

$$\forall c \geq \frac{4 \ln N}{(\sigma^{miss})^2} \left(1 + \sqrt{1 + \frac{4(\sigma^{miss})^2 |\mathcal{F}|}{\ln N}}\right) \forall i, j \leq N, \ \|(\boldsymbol{x}^0)_j^{|\mathcal{F}} - (\boldsymbol{x}^\star)_i^{|\mathcal{F}}\|_2^2 \leq (\sigma^{miss})^2(|\mathcal{F}| + c) \, ,$$

*with probability $1 - \exp\left(-\frac{(\sigma^{miss} c)^2}{4(8|\mathcal{F}|+c)} + 2 \ln N\right) \in [0, 1]$, where $\sigma^{miss} \triangleq \max(\sigma_2, \sigma^{GSM})$, where for an initial $K$-nearest imputation with uniform weights,*

$$\sigma_2 \triangleq \sigma\sqrt{1 + 1/K} \quad \text{(Assumptions B.2-B.3)} \quad \text{and} \quad \sigma^{GSM} \triangleq \sigma\sqrt{(K+3)/3K} \quad \text{(Assumption B.4)} \, .$$

*Proof.* We summarized the procedure according to which the data matrices $X^\star$ and $X^0$ are generated in Algorithm 3. In particular, we assumed that $(x^\star)_i^f \sim_{\text{iid}} \mathcal{N}(\mu_f, \sigma^2)$ for any $i \leq N, f \leq F$ and fixed $\sigma > 0$ (Assumption B.1), $\mu = (\mu_1, \ldots, \mu_F) \in \mathbb{R}^F$, and that $K \leq \min_{f \leq F} |\{i \leq N \mid m_i^f = 0\}|$ (Assumption B.5), where the last term is the number of samples which do not miss the value of feature $f$ in the data set. Based on this, we can assume the following independence relationships for any $i, j \leq F$, where $i \neq j$, and $f, f' \leq F$, where $f \neq f'$

$$(x^\star)_i^f \perp\!\!\!\perp (x^\star)_j^f \tag{8}$$

$$(x^0)_i^f \perp\!\!\!\perp (x^\star)_i^f \mid m_i^f = 1 \tag{9}$$

$$(x^0)_i^f \not\perp\!\!\!\perp (x^\star)_i^f \mid m_i^f = 0 \quad \left(\text{since } \left((x^0)_i^f \mid m_i^f = 0\right) = (x^\star)_i^f\right) \tag{10}$$

$$(x^0)_i^f \not\perp\!\!\!\perp (x^0)_j^f \mid m_i^f = 1, m_j^f = 1 \quad \text{(the two points can share a neighbor)} \tag{11}$$

$$(x^0)_i^f \not\perp\!\!\!\perp (x^0)_j^f \mid m_i^f = 1, m_j^f = 0 \quad ((\boldsymbol{x}^\star)_j \text{ can be a neighbor of } \boldsymbol{x}_i \text{ for } f) \tag{12}$$

$$(x^0)_i^f \perp\!\!\!\perp (x^0)_j^f \mid m_i^f = 0, m_j^f = 0 \tag{13}$$

$$(x^\star)_i^f \perp\!\!\!\perp (x^\star)_j^{f'} \tag{14}$$

$$(x^\star)_i^f \perp\!\!\!\perp (x^\star)_j^{f'} \tag{15}$$

$$(x^\star)_i^f \perp\!\!\!\perp (x^\star)_i^{f'} \text{ and } (x^0)_i^f \perp\!\!\!\perp (x^0)_i^{f'} \, . \tag{16}$$

What is the distribution of random variable $\left((x^0)_i^f \mid m_i^f = m\right)$ for $m \in \{0, 1\}$? If $m_i^f = 0$, that is, if the value for the feature $f$ and sample $i$ is not missing in the input matrix $X$, then $\left((x^0)_i^f \mid m_i^f = 0\right)$ follows the same law as $(x^\star)_i^f$. Otherwise, if $m_i^f = 1$, then at the initial imputation step, $(x^0)_i^f \mid m_i^f = 1$ is the arithmetic

mean of *exactly* $K$ independent random variables of distribution $\mathcal{N}(\mu_f, \sigma^2)$ (by Independence equation 8). [2] All in all,

$$\left((x^0)_i^f \mid m_i^f = 0\right) = (x^\star)_i^f \sim \mathcal{N}(\mu_f, \sigma^2) \quad \text{and} \quad \left((x^0)_i^f \mid m_i^f = 1\right) \sim \mathcal{N}(\mu_f, \sigma^2/K) . \tag{17}$$

Let us denote $\sigma_0 \triangleq \sigma$ and $\sigma_1 \triangleq \sigma/\sqrt{K}$ and $\sigma_2 \triangleq \sqrt{\sigma_0^2 + \sigma_1^2}$ and $\sigma_3 \triangleq \sqrt{\sigma_0^2 - \sigma_1^2}$ and $p_{if}^{\text{miss}} \triangleq \mathbb{P}(m_i^f = 1)$. Let us now consider the distribution of the random variable $\left((x^0)_i^f - (x^\star)_i^f\right)$ for any $i, f$

$$\forall i \leq N, \forall f \leq F, \quad \left((x^0)_i^f - (x^\star)_i^f \mid m_i^f = 0\right) = 0$$
$$\left((x^0)_i^f - (x^\star)_i^f \mid m_i^f = 1\right) \sim \mathcal{N}(0, \sigma_2^2) \quad \text{(by Independence 9) .}$$

Similarly, for any $i, j \leq N$

$$\forall j \neq i, \forall f \leq F, \ \left((x^0)_j^f - (x^\star)_i^f \mid m_j^f = 0\right) \sim \mathcal{N}(0, 2\sigma^2) \quad \text{(by Independence 8)}$$

$$\left((x^0)_j^f - (x^\star)_i^f \mid m_j^f = 1\right) \sim \begin{cases} \mathcal{N}(0, \sigma_2^2) & \text{if } \forall k \leq K, i \neq \mathcal{K}(x_j^f, X^0, k) \\ \mathcal{N}(0, \sigma_3^2) & \text{otherwise} \end{cases},$$

because in the last case, $(x^0)_j^f - (x^\star)_i^f = \frac{1}{K}\sum_{q \neq k}(x^\star)_{\mathcal{K}(x_j^f, X^0, q)}^f + (\frac{1}{K} - 1)(x^\star)_i^f$. Let us denote now $p_{ij} \triangleq \mathbb{P}\left(\forall k \leq K, i \neq \mathcal{K}(x_j^f, X^0, k) \mid m_j^f = 1\right)$. The law of total probability gives

$$\forall i \leq N, \ \forall f \leq F, \ \forall x \neq 0, \quad \mathbb{P}\left(\left((x^0)_i^f - (x^\star)_i^f\right) = x\right) = p_{if}^{\text{miss}} \mathcal{N}(x; 0, \sigma_2^2) \tag{18}$$
$$\mathbb{P}\left(\left((x^0)_i^f - (x^\star)_i^f\right) = 0\right) = 1 + p_{if}^{\text{miss}} \underbrace{\left(\mathcal{N}(0; 0, \sigma_2^2) - 1\right)}_{=1/\sqrt{2\pi\sigma_2^2}-1}$$

$$\forall i \neq j, \ \forall f \leq F, \ \forall x \in \mathbb{R}, \quad \mathbb{P}\left(\left((x^0)_j^f - (x^\star)_i^f\right) = x\right) = (1 - p_{if}^{\text{miss}})\mathcal{N}(0, 2\sigma_0^2) \tag{19}$$
$$+ \ p_{if}^{\text{miss}}\left(p_{ij}\mathcal{N}(x; 0, \sigma_2^2) + (1 - p_{ij})\mathcal{N}(x; 0, \sigma_3^2)\right) .$$

Then, we show that the random variable $(x^0)_j^f - (x^\star)_i^f$ is a zero-mean $\sigma^{\text{miss}}$-subgaussian variable under Assumptions B.2-B.4, where $\sigma^{\text{miss}}$ depends on the missingness mechanism and the initial imputation algorithm. We recall that a zero-mean $\sigma$-subgaussian variable $X$ satisfies $\mathbb{E}[e^{\lambda X}] \leq e^{\sigma^2\lambda^2/2}$ for all $\lambda \in \mathbb{R}$, with equality for any zero-mean Gaussian random variable of variance $\sigma^2$.

**Lemma G.1.** $(x^0)_j^f - (x^\star)_i^f$ *is a zero-mean $\sigma_2$-subgaussian random variable under Assumption B.2. For all $i \neq j \leq N$, $f \leq F$, under the MCAR assumption, $p_{jf}^{miss} = p \in (0, 1)$ is a constant and then $(x^0)_j^f - (x^\star)_i^f$ is a zero-mean $\sigma_2$-subgaussian random variable.*

*Proof.* First, let us denote $X_{ij}^f \triangleq (x^0)_j^f - (x^\star)_i^f$ for $i, j \leq N$ and $f \leq F$. Then using equation 18, it is clear that $X_{ii}^f$ is centered for any $i \leq N$. Similarly, due to equation 19 for any $i \neq j \leq N$

$$\mathbb{E}[X_{ij}^f] = (1 - p)\underbrace{\mathbb{E}[X_{ij}^f \mid m_j^f = 0]}_{=0} + p\underbrace{\mathbb{E}[X_{ij}^f \mid m_j^f = 1]}_{=0} = 0 .$$

Moreover,

$$\forall \lambda \in \mathbb{R}, \ \mathbb{E}[e^{\lambda X_{ii}^f}] = 1 \times \left(1 + p\left(1/\sqrt{2\pi\sigma_2^2} - 1\right)\right) + p\mathbb{E}_{Y \sim \mathcal{N}(0,\sigma_2^2)}[e^{\lambda Y}] - p \times 1 \times \left(1/\sqrt{2\pi\sigma_2^2}\right) ,$$

and then

$$\forall \lambda \in \mathbb{R}, \ \exp(\sigma_2^2\lambda^2/2) - \mathbb{E}[e^{\lambda X_{ii}^f}] = p - 1 + (1 - p)\exp(\sigma_2^2\lambda^2/2) \geq 0 .$$

---
[2]Due to the upper bound on $K$ (Assumption B.5).

Second, we notice that $\sigma_3 \leq \sigma_1 \leq \sigma_0 \leq \sigma_0\sqrt{2} \leq \sigma_2$ (since $K > 1$). It is easy to see that any $\sigma'$-subgaussian variable is also a $\sigma''$-subgaussian variable, where $\sigma' \leq \sigma''$. Then $X_{ij}^f \mid m_j^f = 1$ and $X_{ij}^f \mid m_j^f = 0$ are both $\sigma_2$-subgaussian. Then $X_{ij}^f$ is $\sigma_2$-subgaussian for any $i, j \leq N$. $\qquad\square$

**Lemma G.2.** $(x^0)_j^f - (x^\star)_i^f$ *is a zero-mean $\sigma_2$-subgaussian random variable under Assumption B.3. For all* $i \neq j \leq N$, $f \leq F$, *under the MAR assumption, the missingness depends on a fixed subset of* always observed *values $F^O \subset \{1, 2, \ldots, F\}$: $\mathbb{P}(m_j^f = 1 \mid x_j^\star) = h\big((x^\star)_j^{F^O}, f\big)$ where $(x^\star)_i^{F^O}$ is the restriction of $(x^\star)_i$ to rows in $F^O$ and $h$ some deterministic function.* [3] *Then $(x^0)_j^f - (x^\star)_i^f$ is a zero-mean $\sigma_2$-subgaussian random variable.*

*Proof.* Under Assumption B.3, for all $j \leq N$ and for all $f \in F^O$, $p_{jf}^{\mathrm{miss}} = 0$ and for all $f \notin F^O$,

$$p_{jf}^{\mathrm{miss}} = \int_{x_{f'}, f' \in F^O} h\big([(x^\star)_j^{f'} = x^{f'}, \ f' \in F^O], f\big)\Pi_{f' \in F^O}\mathcal{N}(x^{f'}; \mu_{f'}, \sigma^2)dx \ .$$

By Independence equation 15 and similarly to the proof of Lemma G.1, $X_{ji}^f$ is then a zero-mean $\sigma_2$-subgaussian random variable for any $f \leq F$ and $i, j \leq N$. $\qquad\square$

**Lemma G.3.** $(x^0)_j^f - (x^\star)_i^f$ *is a zero-mean $\sigma^{\mathrm{GSM}}$-subgaussian random variable under Assumption B.4. For all $i \neq j \leq N$, $f \leq F$, under the Gaussian self-masking mechanism from Assumption 4 in Le Morvan et al. (2020), the probability of $x_i^f$ missing is given by*

$$\forall x \in \mathbb{R}, \ \mathbb{P}(m_i^f = 1 \mid (x^\star)_i^f = x) = p_i^{miss}(x, f) = K_f \exp\Big(-\frac{(x - \mu_f)^2}{\sigma^2}\Big) \ with \ K_f \in (0, 1) \ .$$

*Then $(x^0)_j^f - (x^\star)_i^f$ is a zero-mean $\sigma^{GSM}$-subgaussian random variable, where $\sigma^{GSM} \triangleq \sigma\sqrt{\frac{K+3}{3K}}$.*

*Proof.* For all $i \leq N$, $f \leq F$, and for any $x \neq 0$, by the law of total probability

$$\mathbb{P}(X_{ii}^f = x) = \mathbb{P}(X_{ii}^f = x|m_i^f = 1)\mathbb{P}(m_i^f = 1) + \underbrace{\mathbb{P}(X_{ii}^f = x|m_i^f = 0)}_{=0 \text{ because } x \neq 0}\mathbb{P}(m_i^f = 0) \ .$$

Then since $\mathbb{P}(X_{ii}^f = x|m_i^f = 1, (x^\star)_i^f = y) = \mathbb{P}((x^0)_i^f = x + y|m_i^f = 1)$, using equation 17

$$\mathbb{P}(X_{ii}^f = x) = \int_{y \in \mathbb{R}} \mathbb{P}(X_{ii}^f = x|m_i^f = 1, (x^\star)_i^f = y)\mathbb{P}(m_i^f = 1|(x_i^\star)^f = y)\mathbb{P}((x^\star)_i^f = y)dy$$

$$= \int_{y \in \mathbb{R}} \mathcal{N}(x + y; \mu_f, \sigma^2/K)p_i^{\mathrm{miss}}(y, f)\mathcal{N}(y; \mu_f, \sigma^2)dy \triangleq I_{i,f}(x)$$

---

[3]With an abuse in notation as we denote $x_i^\star$ both the random variable and its realization.

$$\forall x \in \mathbb{R},\ I_{i,f}(x) = \int_{y \in \mathbb{R}} \frac{K_f \sqrt{K}}{2\pi\sigma^2} \exp\left(-\frac{(x+y-\mu_f)^2}{\frac{2\sigma^2}{K}}\right) \exp\left(-\frac{(y-\mu_f)^2}{\sigma^2}\right) \exp\left(-\frac{(y-\mu_f)^2}{2\sigma^2}\right) dy$$

$$= \frac{K_f \sqrt{K}}{2\pi\sigma^2} \int_{y \in \mathbb{R}} \exp\left(-\frac{Kx^2 + 2Kx(y-\mu_f) + K(y-\mu_f)^2 + 2(y-\mu_f)^2 + (y-\mu_f)^2}{2\sigma^2}\right) dy$$

$$= \frac{K_f \sqrt{K}}{2\pi\sigma^2} \int_{y \in \mathbb{R}} \exp\left(-\frac{1}{2\sigma^2}\left((K+3)(y-\mu_f)^2 + 2\sqrt{K+3}(y-\mu_f)\frac{Kx}{\sqrt{K+3}}\right.\right.$$
$$\left.\left. + \frac{K^2}{K+3}x^2 - \frac{K^2}{K+3}x^2 + Kx^2\right)\right) dy$$

$$= \frac{K_f \sqrt{K}}{2\pi\sigma^2} \int_{y \in \mathbb{R}} \exp\left(-\frac{1}{2\sigma^2}\left((y\sqrt{K+3} - \mu_f + \frac{Kx}{\sqrt{K+3}})^2 - \frac{3Kx^2}{K+3}\right)\right) dy$$

$$= \frac{K_f \sqrt{K}}{2\pi\sigma^2} \exp\left(-\frac{3Kx^2}{2\sigma^2(K+3)}\right) \int_{y \in \mathbb{R}} \exp\left(-\frac{K+3}{2\sigma^2}\left(y - \frac{\mu_f}{\sqrt{K+3}} + \frac{Kx}{K+3}\right)^2\right) dy$$

$$= \frac{K_f \sqrt{K}}{2\pi\sigma^2} \exp\left(-\frac{3Kx^2}{2\sigma^2(K+3)}\right) \sqrt{\frac{2\pi\sigma^2}{K+3}}$$

$$= K_f \sqrt{\frac{K}{2\pi\sigma^2(K+3)}} \exp\left(-\frac{3K}{2\sigma^2(K+3)}x^2\right) .$$

When $x = 0$, $X_{ii}^f$ follows the second law described at equation 18 and then

$$\mathbb{P}(X_{ii}^f = 0) = \mathbb{P}(X_{ii}^f = 0 | m_i^f = 1)\mathbb{P}(m_i^f = 1) + \underbrace{\mathbb{P}(X_{ii}^f = 0 | m_i^f = 0)}_{=1} \underbrace{\mathbb{P}(m_i^f = 0)}_{=1 - \mathbb{P}(m_i^f = 1)}$$

$$= \int_{y \in \mathbb{R}} \mathbb{P}(X_{ii}^f = 0 | m_i^f = 1, (x^\star)_i^f = y)\mathbb{P}(m_i^f = 1 \mid (x^\star)_i^f = y)\mathbb{P}((x^\star)_i^f = y)dy$$

$$+ 1 - \int_{y \in \mathbb{R}} \mathbb{P}(m_i^f = 1 \mid (x^\star)_i^f = y)\mathbb{P}((x^\star)_i^f = y)dy$$

$$= \int_{y \in \mathbb{R}} \mathcal{N}(y; \mu_f, \sigma^2/K)p_i^{\mathrm{miss}}(y, f)\mathcal{N}(y; \mu_f, \sigma^2)dy + 1 - \int_{y \in \mathbb{R}} p_i^{\mathrm{miss}}(y, f)\mathcal{N}(y; \mu_f, \sigma^2)dy$$

$$= I_{i,f}(0) + 1 - \frac{K_f}{\sqrt{2\pi\sigma^2}} \int_{y \in \mathbb{R}} e^{-\frac{3}{2\sigma^2}(y-\mu_f)^2} dy$$

$$= K_f \sqrt{\frac{K}{2\pi\sigma^2(K+3)}} + 1 - \frac{K_f}{\sqrt{2\pi\sigma^2}}\sqrt{\frac{2\pi\sigma^2}{3}} = 1 + K_f \left(\sqrt{\frac{K}{2\pi\sigma^2(K+3)}} - \frac{1}{\sqrt{3}}\right) .$$

That is

$$\forall x \neq 0,\ \mathbb{P}((x^0)_i^f - (x^\star)_i^f = x) = \frac{K_f}{\sqrt{3}} \times \mathcal{N}(x; 0, (\sigma^{\mathrm{GSM}})^2) \text{ where } \sigma^{\mathrm{GSM}} \triangleq \sigma\sqrt{\frac{K+3}{3K}} \tag{20}$$

$$\mathbb{P}((x^0)_i^f - (x^\star)_i^f = 0) = 1 + K_f \left(\sqrt{\frac{K}{2\pi\sigma^2(K+3)}} - \frac{1}{\sqrt{3}}\right) .$$

For the zero-mean variable X following the distribution described in equation 20, the moment-generating function (MGF) of $X$ is given by

$$\mathbb{E}[e^{tX}] = \int \mathbb{P}(X = x)e^{tx}dx$$

$$= \int_{x \neq 0} e^{tx} \frac{K_f}{\sqrt{6\pi(\sigma^{\mathrm{GSM}})^2}} \exp\left(-\frac{x^2}{2(\sigma^{\mathrm{GSM}})^2}\right) dx$$

$$= \frac{K_f}{\sqrt{3}} \mathbb{E}_{Y \sim \mathcal{N}(0,(\sigma^{\mathrm{GSM}})^2)}[e^{tY}] = \frac{K_f}{\sqrt{3}} \exp\left(\frac{(\sigma^{\mathrm{GSM}})^2 t^2}{2}\right) .$$

Choose $s = \sigma^{\mathrm{GSM}}$. Then,

$$\exp\left(\frac{s^2 t^2}{2}\right) - \mathbb{E}[e^{tX}] = \exp\left(\frac{(\sigma^{\mathrm{GSM}})^2 t^2}{2}\right) - \frac{K_f}{\sqrt{3}} \exp\left(\frac{(\sigma^{\mathrm{GSM}})^2 t^2}{2}\right)$$

$$= \exp\left(\frac{(\sigma^{\mathrm{GSM}})^2 t^2}{2}\right)\left(1 - \frac{K_f}{\sqrt{3}}\right) .$$

Clearly the minimum value, achieved at $t = 0$, is $1 - \frac{K_f}{\sqrt{3}} \geq 0$ since $K_f \in (0, 1)$ by definition. All in all, $X$ is a zero-mean $\sigma^{\mathrm{GSM}}$-subgaussian variable. □

**Lemma G.4.** If $X$ is $s$-subgaussian, then $\beta X$ is $\beta s$-gaussian when $\beta > 0$. *For any $\beta > 0$ and $X$ zero-mean $s$-subgaussian, $\beta X$ is zero-mean $\beta s$-subgaussian.*

*Proof.* If $X$ is a zero-mean $s$-subgaussian, then $\mathbb{E}[\beta X] = 0$ and

$$\forall t > 0, \ \mathbb{P}(|\beta X| \geq t) = \mathbb{P}(|X| \geq \beta^{-1}t) \leq 2\exp\left(-\frac{t^2}{2(\beta s)^2}\right) ,$$

and using Proposition 2.5.2 from Vershynin (2018), $\beta X$ is a (zero-mean) $\beta s$-subgaussian variable. □

Finally, we determine a concentration bound on $\|(\boldsymbol{x}^0)_j^{|\mathcal{F}} - (\boldsymbol{x}^\star)_i^{|\mathcal{F}}\|_2^2$ for any $i, j \leq N$ and $\mathcal{F} \subseteq \{1, 2, \ldots, F\}$ under any of the Assumptions B.2-B.4. Let us set $\sigma^{\mathrm{miss}} \triangleq \max(\sigma_2, \sigma^{\mathrm{GSM}})$ and introduce the $|\mathcal{F}|$-dimensional random vector $\widetilde{X}_{ji}^{|\mathcal{F}} \triangleq (\boldsymbol{x}^0)_j^{|\mathcal{F}} - (\boldsymbol{x}^\star)_i^{|\mathcal{F}}$ for any $i, j \leq N$. The $|\mathcal{F}|$ coefficients of $\widetilde{X}_{ji}^{|\mathcal{F}}$ follow the distribution described in Equations equation 18-equation 19. Then, the random vector $(\sigma^{\mathrm{miss}})^{-1} \widetilde{X}_{ji}^{|\mathcal{F}}$ has $|\mathcal{F}|$ independent 1-subgaussian zero-mean coefficients. The independence holds by Independence equation 16 and equation 8. The coefficients are 1-subgaussian due to Lemma G.4. Using Theorem 3.1.1 from (Vershynin, 2018), which relies on Bernstein's inequality applied to the random variables $(\sigma^{\mathrm{miss}})^{-1} \widetilde{X}_{ji}^f$, for any feature $f \in \mathcal{F}$ and samples $i, j \leq N$, for any constant $c > 0$

$$\mathbb{P}[(\sigma^{\mathrm{miss}})^{-2}\|\widetilde{X}_{ji}^{|\mathcal{F}}\|_2^2 \geq |\mathcal{F}| + c] \leq \exp\left(-\frac{c^2}{4(8|\mathcal{F}| + c)}\right)$$

$$\implies \quad \mathbb{P}[\|(\boldsymbol{x}^0)_j^{|\mathcal{F}} - (\boldsymbol{x}^\star)_i^{|\mathcal{F}}\|_2^2 \geq (\sigma^{\mathrm{miss}})^2(|\mathcal{F}| + c)] \leq \exp\left(-\frac{(\sigma^{\mathrm{miss}}c)^2}{4(8|\mathcal{F}| + c)}\right)$$

$$\implies \quad \mathbb{P}[\cup_{i,j \leq N}\{\|(\boldsymbol{x}^0)_j^{|\mathcal{F}} - (\boldsymbol{x}^\star)_i^{|\mathcal{F}}\|_2^2 \geq (\sigma^{\mathrm{miss}})^2(|\mathcal{F}| + c)\}] \leq \exp\left(-\frac{(\sigma^{\mathrm{miss}}c)^2}{4(8|\mathcal{F}| + c)} + 2\ln N\right) ,$$

by applying an union bound on $\{1, 2, \ldots, N\}^2$. And then for any positive constant $c$ such that $2\ln N - (\sigma^{\mathrm{miss}}c)^2/(4(8|\mathcal{F}| + c)) \leq 0$,

$$\mathbb{P}[\cap_{i,j \leq N} \|(\boldsymbol{x}^0)_j^{|\mathcal{F}} - (\boldsymbol{x}^\star)_i^{|\mathcal{F}}\|_2^2 \leq (\sigma^{\mathrm{miss}})^2(|\mathcal{F}| + c)]$$

$$= \quad 1 - \mathbb{P}[\cup_{i,j \leq N} \|(\boldsymbol{x}^0)_j^{|\mathcal{F}} - (\boldsymbol{x}^\star)_i^{|\mathcal{F}}\|_2^2 \geq (\sigma^{\mathrm{miss}})^2(|\mathcal{F}| + c)]$$

$$\geq \quad 1 - \exp\left(-\frac{(\sigma^{\mathrm{miss}}c)^2}{4(8|\mathcal{F}| + c)} + 2\ln N\right) .$$

A positive such $c$ always exists, which can be shown by choosing $c$ such that

$$c \geq \frac{4\ln N}{(\sigma^{\text{miss}})^2}\left(1 + \sqrt{1 + 4(\sigma^{\text{miss}})^2|\mathcal{F}|/\ln N}\right) > 0 \implies 2\ln N - (\sigma^{\text{miss}}c)^2/(4(8|\mathcal{F}| + c)) \leq 0 \, .$$

Note that, similarly, by union bound on $\{1, 2, \ldots, N\}$, for such a $c$,

$$\mathbb{P}[\cap_{i \leq N} \, \|(\boldsymbol{x}^0)_i^{|\mathcal{F}} - (\boldsymbol{x}^\star)_i^{|\mathcal{F}}\|_2^2 \leq (\sigma^{\text{miss}})^2(|\mathcal{F}| + c)] \geq 1 - \exp\left(-\frac{(\sigma^{\text{miss}}c)^2}{4(8|\mathcal{F}| + c)} + \ln N\right) \in [0, 1] \, .$$

$\square$

**Corollary G.5.** First concentration bound on $\|(\boldsymbol{x}^0)_j - (\boldsymbol{x}^\star)_i\|_2^2$. *Under any assumption in Assumptions B.2-B.4, then*

$$\forall c \geq \frac{4\ln N}{(\sigma^{miss})^2}\left(1 + \sqrt{1 + \frac{4(\sigma^{miss})^2 F}{\ln N}}\right) \quad \forall i, j \leq N, \quad \|(\boldsymbol{x}^0)_j - (\boldsymbol{x}^\star)_i\|_2^2 \leq (\sigma^{miss})^2(F + c) \, ,$$

*with probability* $1 - \exp\left(-\frac{(\sigma^{miss}c)^2}{4(8F+c)} + 2\ln N\right) \in [0, 1]$, *where* $\sigma^{miss} \triangleq \max(\sigma_2, \sigma^{GSM}) \propto \sigma$ *is defined in Technical lemma 3.*

*Proof.* This statement holds by application of Lemma 3 with $\mathcal{F} = \{1, 2, \ldots, F\}$. $\square$

**Corollary G.6.** Second concentration bound on $\|(\boldsymbol{x}^0)_j - (\boldsymbol{x}^\star)_i\|_2^2$ and $\|(\boldsymbol{x}^0)_i - (\boldsymbol{x}^\star)_i\|_2^2$. *Under any assumption in Assumptions B.2-B.4, for* $\sigma^{miss} \triangleq \max(\sigma_2, \sigma^{GSM}) \propto \sigma$ *(Technical lemma 3), let us denote*

$$C_\delta^{miss} \triangleq (\sigma^{miss})^2 F + 2\ln(1/\delta)\left(1 + \sqrt{1 + 8(\sigma^{miss})^2 F/\ln(1/\delta)}\right) \text{ for } \delta \leq 1/N \, .$$

*Then, with probability* $1 - \delta \in (0, 1)$, *for all* $i, j \leq N$, $\|(\boldsymbol{x}^0)_j - (\boldsymbol{x}^\star)_i\|_2^2 \leq C_{\delta/N^2}^{miss}$.

*Proof.* We solve the following equation in $c > 0$ from Corollary G.5,

$$\delta = \exp\left(-\frac{(\sigma^{\text{miss}}c)^2}{4(8F + c)} + 2\ln N\right) \Leftrightarrow -(\sigma^{\text{miss}})^2 c^2 + (4\ln(N^2/\delta))c + 32F\ln(N^2/\delta) = 0 \, .$$

This equation has two real roots, one positive root being

$$c_\delta \triangleq \frac{4\ln(N^2/\delta)}{(\sigma^{\text{miss}})^2}\left(1 + \sqrt{1 + 8(\sigma^{\text{miss}})^2 F/\ln(N^2/\delta)}\right) \, .$$

Applying Corollary G.5 with $c = c_\delta$ when $\delta \in (0, 1)$ yields for any $j, i \leq N$, with probability $1 - \frac{\delta}{N^2} \in (0, 1)$,

$$\|(\boldsymbol{x}^0)_j - (\boldsymbol{x}^\star)_i\|_2^2 \leq C_{\delta/N^2}^{\text{miss}} \, .$$

Applying an upper bound on $\{1, 2, \ldots, N\}^2$ yields the expected result. $\square$

**Proposition G.7.** Iterative improvement from $X^0$ until $X^t$. *For* $G_\circ \in \{G, G_\star\}$, *for any data matrix* $X \in (\mathbb{R} \cup \{\text{NaN}\})^{N \times F}$ *and* $(\boldsymbol{\alpha}^s)_{s \leq t} \in (\triangle_K)^t$

$$\sum_{s=1}^{t} G_\circ(\boldsymbol{\alpha}^s, X^{s-1}) = \frac{1}{N}\sum_{i \leq N} \log D_\circ((\boldsymbol{x}^t)_i(\boldsymbol{\alpha}^t))/\log D_\circ((\boldsymbol{x}^0)_i) - \eta\sum_{s=1}^{t}\|\boldsymbol{\alpha}^s\|_2^2 \, .$$

*Proof.* For any $G_\circ \in \{G, G_\star\}$ and $t \geq 1$, since $(\boldsymbol{x}^s)_i = (\boldsymbol{x}^{s-1})_i(\boldsymbol{\alpha}^s)$ for $s < t$ and $i \leq N$

$$\begin{aligned}
\sum_{s=1}^{t} G_\circ(\boldsymbol{\alpha}^s, X^{s-1}) &= \sum_{s=1}^{t}\frac{1}{N}\sum_{i \leq N}\log\frac{D_\circ((\boldsymbol{x}^{s-1})_i(\boldsymbol{\alpha}^s))}{D_\circ((\boldsymbol{x}^{s-1})_i)} - \eta\|\boldsymbol{\alpha}^s\|_2^2 \\
&= \frac{1}{N}\sum_{i \leq N}\log\frac{D_\circ((\boldsymbol{x}^t)_i(\boldsymbol{\alpha}^t))}{D_\circ((\boldsymbol{x}^0)_i)} - \eta\sum_{s=1}^{t}\|\boldsymbol{\alpha}^s\|_2^2 \, .
\end{aligned}$$

$\square$

# H    Experimental study

We compare our algorithmic contributions F3I and PCGrad-F3I to baselines for imputation and joint imputation-classification tasks. We considered as baselines the imputation by the mean value, the MissForest algorithm (Stekhoven & Bühlmann, 2012), K-nearest neighbor (KNN) imputation with uniform weights and distance-proportional weights, where the weight is inversely proportional to the distance to the neighbor (Troyanskaya et al., 2001), an Optimal Transport-based imputer (Muzellec et al., 2020a) and finally not-MIWAE (Ipsen et al., 2021).

We consider synthetic data sets produced by Algorithm 3, public drug repurposing data sets and the MNIST data set for handwritten-digit recognition (LeCun et al., 1998), along with the three missingness mechanisms corresponding to Assumptions B.2-B.4 for different missingness frequencies in $\{0.1, 0.25, 0.5, 0.75, 0.9\}$ across the full matrix. The missingness frequencies aim at approximating the actual expected probability of a missing value.

*Remark* H.1. Implementation of the missingness mechanisms. The implementations of the MCAR and MAR mechanisms come from (Muzellec et al., 2020a) (with `opt='logistic'`). For a MCAR mechanism or MAR mechanism implemented by (Muzellec et al., 2020a), it corresponds to the random probability of missing data. For the MNAR Gaussian self-masking and feature $f$, using the notation in Assumption B.4, we sample $K_f$ from $\mathcal{N}(\frac{3.5}{3} p^{\text{miss}}(1 - p^{\text{miss}}), 0.1)$ and we clip $K_f$ in $[0.01, 0.99]$ whenever necessary. Empirically, as long as $p^{\text{miss}}$, the expected missingness frequency is not too extreme (*i.e.*, far from the bounds of $[0, 1]$), the empirical probability of missingness is close to $p^{\text{miss}}$. However, controlling this probability more finely in the case of a MNAR mechanism might break the not-missing-at-random property.

*Remark* H.2. Computational resources. The experiments on synthetic data (Subsection H.1) were run on a personal laptop (processor 13th Gen Intel(R) Core(TM) i7-13700H, 20 cores @5GHz, RAM 32GB). The experiments on drug repurposing (Subsection H.2) were run on remote cluster servers (processor QEMU Virtual v2.5+, 48 cores @2.20GHz, RAM 500GB, and processor Intel Core i7-8750H, 20 cores @2.50GHz, RAM 7.7GB for the TRANSCRIPT drug repurposing data set (Réda, 2023b)). No GPU was used in our experiments.

*Remark* H.3. Time complexity for the imputation steps in F3I (Algorithm 1). The time complexity of running the KNN imputer (Troyanskaya et al., 2001) with uniform weights and building the k-d tree on $N$ $F$-dimensional points is $\mathcal{O}(FN \log N)$, both steps being performed once. For each input point $\boldsymbol{x}$, Algorithm 2 first queries $K$ nearest neighbors (each query has a time complexity of $\mathcal{O}(\log N)$) and then performs the imputation in at most $FK$ operations, for a total time complexity across all points of $\mathcal{O}(NK(\log N + F))$.

## H.1    Synthetic Gaussian data sets

The data matrices $X \in \mathbb{R}^{N \times F}$ with $N = 50$ samples and $F = 100$ features are generated according to Algorithm 3 (Lines 4-18), with the multivariate mean parameter $\boldsymbol{\mu} \sim \mathcal{N}(\boldsymbol{0}_F, \nu^2 \boldsymbol{I}_{F \times F})$ where $\nu = 0.1$ and covariance parameter $\Sigma = \sigma^2 \boldsymbol{I}_{F \times F}$. Hyperparameter values are reported in Table 5. We use $K = 5$ neighbors here for all algorithms for which it is relevant.

### H.1.1    Validation of theoretical results (single imputation task)

We first show on synthetic data that both Theorems 4.2 and 4.4 are experimentally validated, and look at the behavior of $\boldsymbol{\alpha}$ across imputation steps in F3I for all missingness mechanisms.

**Empirical validation of Theorem 4.2**    We fix the missingness frequency to 25% for all three missingness mechanisms. First, in Figure 1, for each missingness type in Assumptions B.2-B.4, we ran F3I on 100 randomly generated synthetic data matrices with each $\sigma \in \{0.01, 0.1, 0.15, 0.2, 0.25, 0.5\}$ instead of $\sigma = 0.1$ and reported the mean-squared error (MSE) loss $\mathcal{L}^{\text{MSE}}(X^t, X^\star)$ (where $X^t$ is the last imputed data set in F3I) along with the corresponding $\sigma$-dependent upper bound $C^{\text{miss}} = \mathcal{O}((\sigma^{\text{miss}})^2 F + \ln N)$. The exact definition of $C^{\text{miss}}$ is in the proof of Theorem 4.2, in Appendix D. The upper bound is largely above the MSE value for each iteration. This might be because concentration bounds derived from Bernstein's inequality are not very tight. From Table 6 which reports the numerical values shown on Figure 1, we notice that there is a correlation between $\sigma$ and $\mathcal{L}^{\text{MSE}}(X^t, X^\star)$. Moreover, $C^{\text{miss}}$ recovers interesting dependencies as an

Table 5: Hyperparameters for F3I (Algorithm 1) and its baselines, unless otherwise specified (as some of those hyperparameters might be finetuned in our experiments). K is the number of neighbors in F3I. The names of the hyperparameters match the corresponding argument names in their implementation in Python (official, in scikit-learn (Pedregosa et al., 2011) or in HyperImpute (Jarrett et al., 2022), if present). The $k$ in TDM is not a number of neighbors or anything equivalent.

| Imputer | Hyperparameters |
|---|---|
| F3I | `n_neighbors`$= K$, `max_iter`$= 500$, $\eta= 0.001$, `S`$= 1$, $\beta= 0$ |
| MissForest (Stekhoven & Bühlmann, 2012) | `n_estimators`$= K$, `max_depth`$= 10$, `max_size`$= 0.5$, `max_iters`$= 500$, $\beta= 0$ |
| KNN (Troyanskaya et al., 2001) (uniform) | `n_neighbors`$= K$, `distance`$=$'nan_euclidean' |
| KNN (Troyanskaya et al., 2001) (distance) | `n_neighbors`$= K$, `distance`$=$'nan_euclidean' |
| Optimal Transport (Muzellec et al., 2020a) | `eps`$= 0.01$, `lr`$= 0.01$, `max_iters`$= 500$, `batch_size`$= 128$, `n_pairs`$= 1$, `noise`$= 0.1$, `scaling`$= 0.9$ |
| not-MIWAE (Ipsen et al., 2021) | `n_latent`$= \lfloor F/2 \rfloor$, `n_hidden`$= 150$ |
| GAIN (Yoon et al., 2018a) | `batch_size`$= 128$, `n_epochs`$= 100$, `hint_rate`$= 0.8$, `loss_alpha`$= 10$ |
| GRAPE (You et al., 2020) | `node_dim`$= 64$, `edge_dim`$= 16$, `nepochs`$= 20,000$ |
| HyperImpute (Jarrett et al., 2022) | `imputation_order`$= 2$, `baseline_imputer`$= 0$, `optimizer`$=$'simple', `class_threshold`$= 5$, `optimize_thresh`$= 5,000$, `n_inner_iter`$= 40$, `select_patience`$= 5$ |
| MIRACLE (Kyono et al., 2021) | `lr`$= 0.001$, `batch_size`$= 1,024$, `num_outputs`$= 1$, `n_hidden`$= 32$, `reg_lambda`$= 1$, `reg_beta`$= 1$, `reg_m`$= 1.0$, `window`$= 10$, `max_steps`$= 400$, `seed_imputation`$=$'mean' |
| NewImp (Chen et al., 2024) | `entropy_reg`$= 10$, `eps`$= 0.01$, `lr`$= 0.01$, `opt`$=$'Adam', `niter`$= 50$, `kernel_func`$=$'xRBF', `mlp_hidden`$= [256, 256]$, `score_net_epoch`$= 2,000$, `score_net_lr`$= 0.001$, `score_loss_type`$=$'dsm', `bandwidth`$= 10$, `sampling_step`$= 500$, `batchsize`$= 128$, `n_pairs`$= 1$, `noise`$= 0.1$, `scaling`$= 0.9$ |
| Remasker (Du et al., 2023) | `batch_size`$= 64$, `max_epochs`$= 600$, `accum_iter`$= 1$, `mask_ratio`$= 0.5$, `embed_dim`$= 32$, `depth`$= 6$, `decoder_depth`$= 4$, `num_heads`$= 4$, `mlp_ratio`$= 4$, `encode_func`$=$'linear', `weight_decay`$= 0.05$, `lr`$=$None, `blr`$= 0.001$, `min_lr`$= 0.00001$, `warmup_epochs`$= 40$ |
| TDM (Zhao et al., 2023) | `k`$= 2$, `depth`$= 3$, `im_lr`$= 0.01$, `proj_lr`$= 0.01$, `opt`$=$'RMSprop', `niter`$= 2,000$, `batchsize`$= 128$, `n_pairs`$= 1$, `noise`$= 0.1$ |
| RF-GAP (Rhodes et al., 2023) | `prox_method`$=$'rfgap' |

upper bound of $\mathcal{L}^{\mathrm{MSE}}(X^t, X^\star)$. Indeed, we also observe empirically that $\mathcal{L}^{\mathrm{MSE}}(X^t, X^\star)$ is roughly linear in $\sigma^2 \approx (\sigma^{\mathrm{miss}})^2$ regardless of the missingness mechanism (see Figure 2), which matches the upper bound given by Theorem 4.2.

**Behavior of $\boldsymbol{\alpha}^t$ depending on imputation round $t$** Second, we look at the evolution of $\boldsymbol{\alpha}^t$ depending on the round $t$, knowing that at $t = 0$, $\boldsymbol{\alpha}^0 = \frac{1}{K}\mathbf{1}_K$ is a uniform weight vector. Figure 3 displays the evolution of weight $(\alpha^t)_k$ for each $k$-nearest neighbor in iteration $t$ in F3I. Surprisingly enough, the optimal weight vector is not proportional to the rank of the neighbor; that is, the closer the neighbor, the higher the weight, which often motivates some heuristics about k-nearest neighbor algorithms. Optimality (preserving the data distribution) puts higher weights on the first *and last* closest neighbors.

**Empirical validation of Theorem 4.4** Third, we look at the upper bound for the expected cumulative regret stated in Theorem 4.4. Using a missingness frequency of 25% again, we run 100 times F3I on synthetic

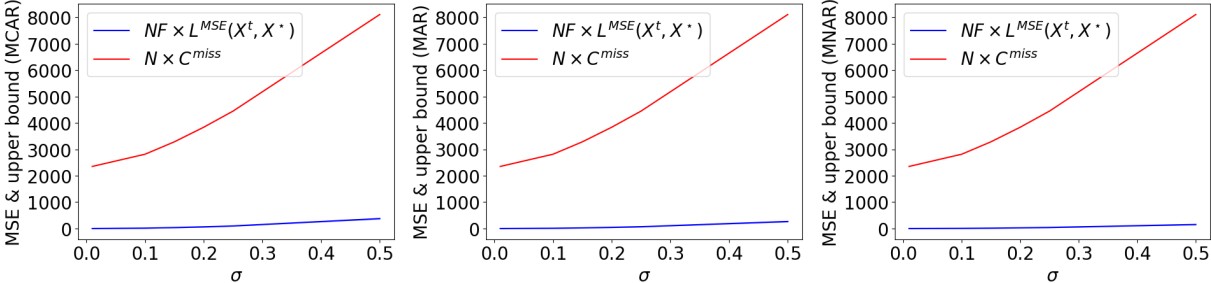

Figure 1: Emprical validation of Theorem 4.2 by comparing the value of the upper bound $N \times C^{\mathrm{miss}}$ and $NF \times \mathcal{L}^{\mathrm{MSE}}(X^t, X^\star)$ where $t$ is the final round for F3I. Left: MCAR setting. Center: MAR setting. Right: MNAR setting.

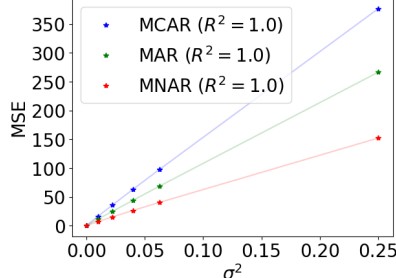

Figure 2: $NF \times \mathcal{L}^{\mathrm{MSE}}(X^t, X^\star)$ is linear in $\sigma^2$ regardless of the missingness mechanism (numerical values are reported in Table 6).

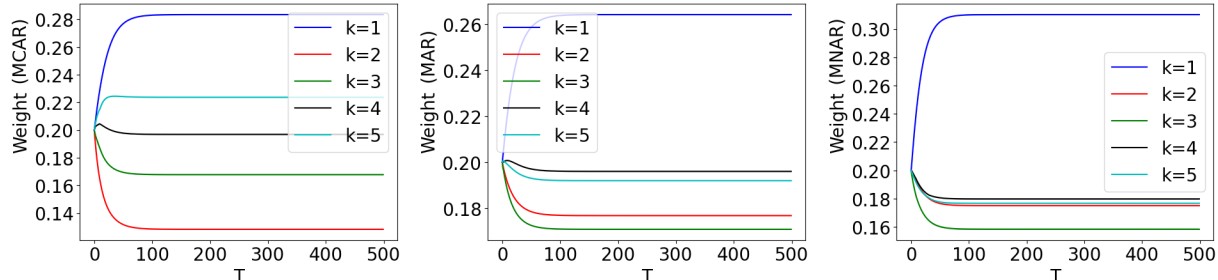

Figure 3: Evolution of the weight of each of the $K$-nearest neighbors for each sample as computed by F3I, where the $k$ neighbor is the $k^{\mathrm{th}}$-nearest point, depending on the round $T$. Left: MCAR setting. Center: MAR setting. Right: MNAR setting.

Table 6: Empirical validation of Theorem 4.2 by comparing the value of the upper bound $N \times C^{\mathrm{miss}}$ and the average and standard deviation value of $NF \times \mathcal{L}^{\mathrm{MSE}}(X^t, X^\star)$ across iterations where $t$ is the final round for F3I. All values are rounded to the closest second decimal place. Theorem 4.2 states that $\mathcal{L}^{\mathrm{MSE}}(X^t, X^\star) \leq C^{\mathrm{miss}}/F$ with probability $1 - 1/50$.

| Missingness type | $\sigma$ | $NF \times \mathcal{L}^{\mathrm{MSE}}(X^t, X^\star)$ | $N \times C^{\mathrm{miss}}$ |
|---|---|---|---|
| MCAR (Assumption B.2) | 0.01 | 0.63 $\pm$0.38 | 2,352.60 |
| | 0.10 | 16.36 $\pm$0.93 | 2,816.02 |
| | 0.15 | 36.12 $\pm$1.89 | 3,286.57 |
| | 0.20 | 63.26 $\pm$3.14 | 3,839.30 |
| | 0.25 | 97.44 $\pm$4.56 | 4,450.16 |
| | 0.50 | 376.29 $\pm$16.76 | 8,109.04 |
| MAR (Assumption B.3) | 0.01 | 0.42 $\pm$0.21 | 2,352.60 |
| | 0.10 | 11.53 $\pm$0.78 | 2,816.02 |
| | 0.15 | 25.19 $\pm$1.58 | 3,286.57 |
| | 0.20 | 44.07 $\pm$2.65 | 3,839.30 |
| | 0.25 | 68.50 $\pm$4.08 | 4,450.16 |
| | 0.50 | 266.19 $\pm$15.22 | 8,109.04 |
| MNAR (Assumption B.4) | 0.01 | 0.37 $\pm$0.23 | 2,352.60 |
| | 0.10 | 7.13 $\pm$0.64 | 2,816.02 |
| | 0.15 | 15.50 $\pm$1.22 | 3,286.57 |
| | 0.20 | 26.53 $\pm$1.86 | 3,839.30 |
| | 0.25 | 40.36 $\pm$2.66 | 4,450.16 |
| | 0.50 | 152.03 $\pm$9.56 | 8,109.04 |

data sets for all three missingness mechanisms and track the values of $\max_{\boldsymbol{\alpha} \in \triangle_K} \sum_{s=1}^{t} G_*(\boldsymbol{\alpha}, X^{s-1}) - G_*(\boldsymbol{\alpha}^s, X^{s-1})$ and its upper bound $C^{\mathrm{AH}}\sqrt{t} + H^{\mathrm{miss}}h^{-1}t$ across iterations, where $t$ is the final step of F3I (that can change across iterations). We compute the value of $\max_{\boldsymbol{\alpha} \in \triangle_K} \sum_{s=1}^{t} G_*(\boldsymbol{\alpha}, X^{s-1})$ by solving the related convex problem with function `minimize` in Python package `scipy.optimize` (Virtanen et al., 2020) after running F3I

$$\min_{\boldsymbol{\alpha} \in \mathbb{R}^K} -\sum_{s=1}^{t} G_\star(\boldsymbol{\alpha}, X^{s-1}) \quad \text{such that} \quad \forall k \leq K, \ \alpha_k \geq 0 \quad \text{and} \quad \sum_{k \leq K} \alpha_k = 1 \,,$$

where $G_\star$ is computed with respect to the true complete points $\{(\boldsymbol{x}^\star)_1, \ldots, (\boldsymbol{x}^\star)_N\}$ and $(\boldsymbol{x}^s)_i = $ `Impute`$((\boldsymbol{x}^{s-1})_i; \boldsymbol{\alpha}^s)$ if $s \geq 1$ and $X^0$ is the naively imputed matrix. Figure 4 and Table 7 show that the upper bound is always valid across those experiments. Some random data sets among the 100 might be harder than the others, incurring larger regret. However, Figure 4 shows that the upper bound adapts to these instances. The large gap between the empirical and theoretical peaks in hardness might be due, as for Theorem 4.2, to the conservative estimates given by Bernstein's inequality.

### H.1.2 Empirical performance (single imputation task)

The main reason why we did not test all the data sets present in Jarrett et al. (2022) is because some of them have categorical/discrete data as features: *e.g.*, Airfoil Self-Noise Dataset, Blood Transfusion Service Center Dataset, California Housing Dataset, Concrete Compressive Strength Dataset, Iris Dataset, and Letter Recognition Dataset. Our contribution F3I without Appendix H.3 is not suitable for such data, as highlighted in the limitations.

Now we compare the MSE of F3I to its baselines on synthetic data sets generated with Algorithm 3. Note that the definition of the MSE is slightly different from Definition 4.1 as we only compute the gaps in the

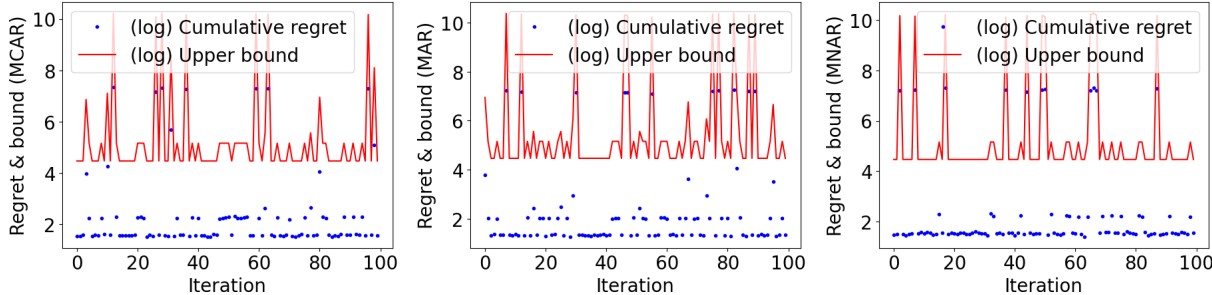

Figure 4: Cumulative regret for F3I and upper bound from Theorem 4.4 across 100 iterations in log-values. The blue points are always below the red lines. Left: MCAR setting. Center: MAR setting. Right: MNAR setting.

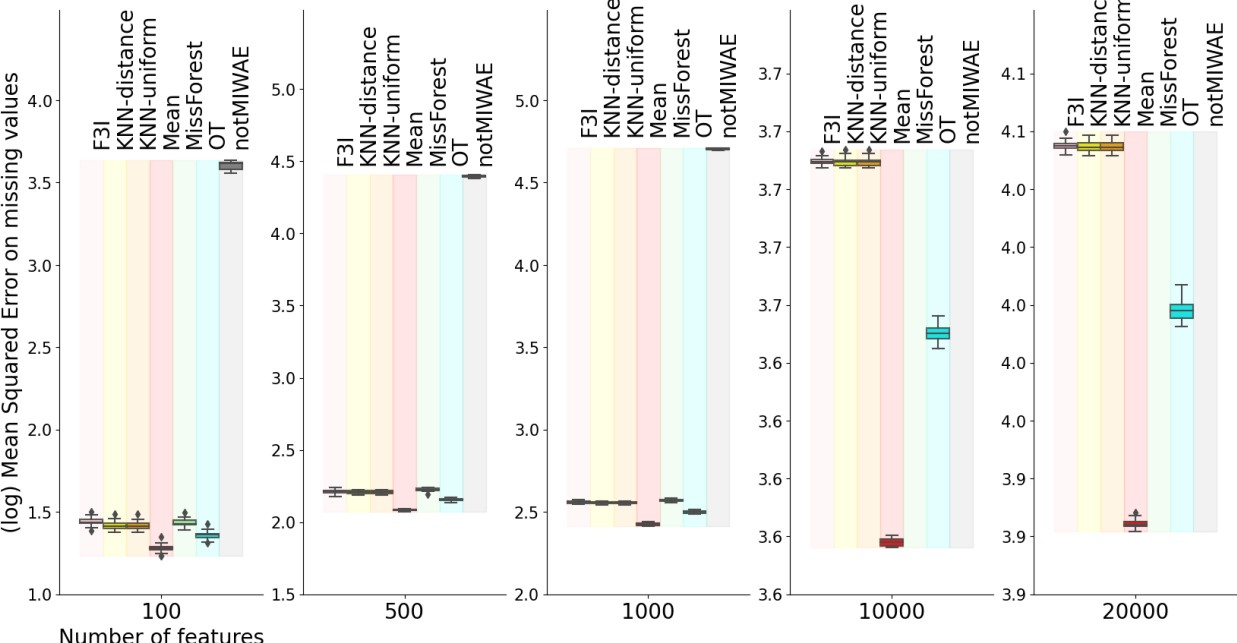

Figure 5: Imputation on 2 synthetic data sets × 10 different random seeds for generating missing values (MCAR, $p^{\mathrm{miss}} = 25\%$, K=3) for F3I, K-nearest neighbor imputers (Troyanskaya et al., 2001) (uniform or distance-based weights), mean imputation, MissForest (Stekhoven & Bühlmann, 2012), Optimal-Transport imputer (Muzellec et al., 2020a) and not-MIWAE (Ipsen et al., 2021).

Table 7: Empirical validation of Theorem 4.4 by comparing the value of the upper bound and the average and standard deviation value of the cumulative regret $\max_{\boldsymbol{\alpha} \in \triangle_K} \sum_{s=1}^{t} G_*(\boldsymbol{\alpha}, X^{s-1}) - G_*(\boldsymbol{\alpha}^s, X^{s-1})$ across iterations where $t$ is the final round for F3I (the maximum number of rounds is set to 500). All values are rounded to the closest second decimal place, except for the time round $t$, which is rounded to the closest integer. Theorem 4.4 states that $\max_{\boldsymbol{\alpha} \in \triangle_K} \sum_{s=1}^{t} G_*(\boldsymbol{\alpha}, X^{s-1}) - G_*(\boldsymbol{\alpha}^s, X^{s-1}) \leq C^{\mathrm{AH}}\sqrt{t} + H^{\mathrm{miss}}h^{-1}t$ with probability $1 - 1/2, 500 \approx 0.9996$.

| Missingness type | | # Iterations | Cumulative regret on $G$ | Upper bound from Theorem 4.4 |
|---|---|---|---|---|
| MCAR | (Assumption B.2) | 24 ±78 | 113.76 ±371.39 | 2,067.60 ±6,734.50 |
| MAR | (Assumption B.3) | 40 ±111 | 153.06 ±417.40 | 3,542.96 ±9,659.89 |
| MNAR | (Assumption B.4) | 35 ±96 | 158.61 ±438.45 | 3,015.86 ±8,336.24 |

positions of missing values

$$\overline{\mathcal{L}}^{\mathrm{MSE}}(X^t, X^\star) \triangleq \frac{1}{N} \sum_{i \leq N} \frac{1}{|\{f \mid m_i^f = 1\}|} \sum_{f, m_i^f = 1} ((x^t)_i^f - (x^\star)_i^f)^2 \leq F\mathcal{L}^{\mathrm{MSE}}(X^t, X^\star) \,.$$

We consider the following baselines: imputation by the mean value, random-forest-based imputation by MissForest (Stekhoven & Bühlmann, 2012), traditional KNN imputation (Troyanskaya et al., 2001) with uniform and distance-based weights, Optimal Transport-based imputation (Muzellec et al., 2020a) and the Bayesian network approach not-MIWAE (Ipsen et al., 2021). We consider a number of neighbors (for F3I and KNN) or of estimators (for MissForest) $K = 3$, a number of features $F \in \{100; 500; 1,000; 10,000; 20,000\}$ with $N = 50$ samples, and a missingness frequency $p^{\mathrm{miss}} \propto \{0.10, 0.25, 0.50, 0.75\}$. We generate 2 random data sets and perform 10 iterations of each algorithm on each data set, for a total of 20 values per combination of parameters ($F$, missingness mechanism), where the missingness mechanism is either MCAR, MAR, or MNAR.

We first note that not-MIWAE has a significantly worse imputation error than all other algorithms; see Figure 5. We then do not report the results for not-MIWAE in all other cases. Moreover, from $F = 1,000$, the runtime of MissForest is too long to be run. The remainder of the boxplots for the mean squared error and imputation runtime across iterations and data sets can be found in Figures 11-19.

Second, we note the superior performance of F3I, nearest neighbor, and mean imputers regarding the computational cost of imputation across missingness frequencies $p^{\mathrm{miss}}$, missingness mechanisms (MCAR, MAR, MNAR), and numbers $F$ of features. This makes F3I a competitive approach when the number of features is huge (for instance, $F \in \{10,000; 20,000\}$).

Third, as a general rule across missingness mechanisms and frequencies, the performance of F3I is close to the ones of other nearest-neighbor imputers and sometimes better when the number of features is large. It might be because F3I considers the same neighbors across features past the initial imputation step. This allows us to impute perhaps more reliably missing values, contrary to the other NN imputers where neighbor assignation is performed feature-wise. Moreover, we notice that the performance of F3I and all baselines are on par (that is, boxplots overlap) for data where the missing values are generated from a MNAR mechanism (Figures 17-19) regardless of the missing frequency.

Fourth, F3I performs worst for data generated by a MCAR or a MAR mechanism, which is the setting where mean imputation works best. This might make sense, as F3I tries to capture a specific missingness pattern that depends on neighbors in the data set. The (perfectly) random pattern might be the most difficult to infer for F3I and other nearest-neighbor imputers. Moreover, if we compare the performance in RMSE of F3I to the KNN algorithm with uniform weights, which is the naive imputer in the initialization of F3I, we see that the performance of F3I is on par with KNN on Uniform KNN on the Gottlieb dataset (Table 1), and slightly worse ($\approx$+20%) on the Spambase, Wine (White), and Wine (Red) datasets, as shown in Table 8. However, F3I can slightly outperform KNN on the Libras ($\approx$-10%) and Ionosphere ($\approx$-5%) datasets, and significantly. on the Diabetes ($\approx$-262%) and Heart Disease ($\approx$-185%) datasets. This shows that F3I is a novel

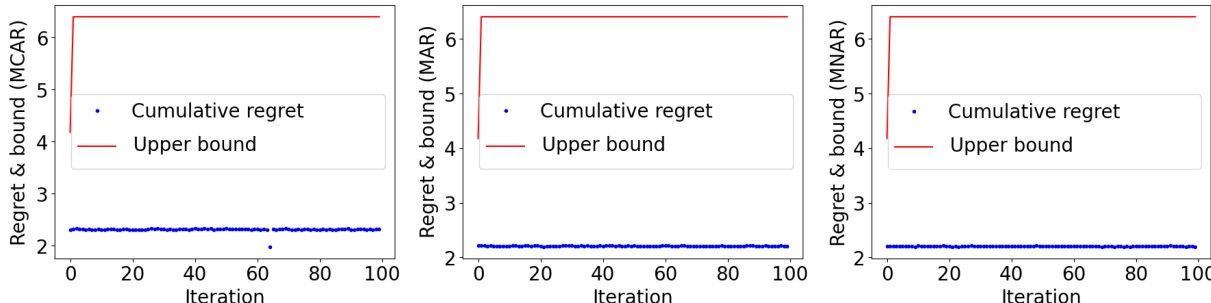

Figure 6: Cumulative regret for F3I and upper bound from Theorem 5.1 across 100 iterations for $\beta = 0.5$. The blue points are always below the red lines. Left: MCAR setting. Center: MAR setting. Right: MNAR setting.

approach to use KNN-based strategies for imputation which is at least as good as base KNN, and might offer superior performance on some datasets.

### H.1.3 Validation of theoretical results (joint imputation-classification task)

We implement the joint imputation-classification training with the log-loss function and sigmoid classifier $\ell(\boldsymbol{x}) \triangleq -y \log C_{\boldsymbol{\omega}}(\boldsymbol{x})$ mentioned in Example 1 (Appendix F), where $y \in \{0, 1\}$ is the binary class associated with sample $\boldsymbol{x} \in \mathbb{R}^F$. To implement PCGrad-F3I, we chain the imputation phase by F3I with an MLP, which returns logits. At time $t$, the imputation part applies at a fixed set of parameters $\boldsymbol{\omega}^t$ with the learner losses defined in equation 3. We construct the synthetic data sets for classification as follows. We draw two random matrices in the synthetic data set as in Algorithm 3 corresponding to the item and user feature matrices. We assign binary class labels to each item-user pair using a K-means++ algorithm (Arthur & Vassilvitskii, 2006) with $K = 2$ clusters on the item-user concatenated feature vectors.

We first validate the upper bound on the cumulative regret in Theorem 5.1 for all three missingness mechanisms we studied, similar to what was done for Theorem 4.4. As in our proofs (see Appendix F), we consider the log-loss $\ell$ with the sigmoid classifier $C_{\boldsymbol{\omega}}$ and $\beta = 0.5$. We consider a missingness frequency of 50%. To obtain the value of arg $\max_{\boldsymbol{\alpha} \in \triangle_K} \sum_{s=1}^{t} \overline{G}(\boldsymbol{\alpha}, X^{s-1})$, where $\overline{G}$ is defined as

$$\overline{G} : \boldsymbol{\alpha} \in \triangle_K, X \in \mathbb{R}^{N \times F} \mapsto (1 - \beta)G_{\star}(\boldsymbol{\alpha}, X) - \frac{\beta}{N} \sum_{i \leq N} \ell(\boldsymbol{x}_i(\boldsymbol{\alpha})) \,,$$

we solve the following optimization problem by solving the related convex problem with function `minimize` in Python package `scipy.optimize` (Virtanen et al., 2020), considering the $(X^{s-1})_{s \leq t}$ and parameter of the sigmoid classifier $C_{\omega}$ in the last epoch in PCGrad-F3I

$$\min_{\boldsymbol{\alpha} \in \mathbb{R}^K} - \sum_{s=1}^{t} \overline{G}(\boldsymbol{\alpha}, X^{s-1}) \quad \text{such that} \quad \forall k \leq K, \ \alpha_k \geq 0 \quad \text{and} \quad \sum_{k \leq K} \alpha_k = 1 \,.$$

The gradient and the Hessian matrix of the objective function $\overline{G}$ with respect to $\boldsymbol{\alpha}$ are obtained by combining the results from Lemmas C.3, Lemma C.4 and Section F. Figure 6 and Table 9 indeed show that the upper bound reliably holds on the cumulative regret for $\overline{G}$.

### H.1.4 Empirical performance (joint imputation-classification task)

Then, we compare the performance of PCGrad-F3I with adding a NeuMiss block (Le Morvan et al., 2020) and with performing an imputation by the mean ("Mean") before the classifier. Similarly to PCGrad-F3I, we chain an imputation part with an MLP classifier, which returns logits. In the baselines, at time $t$, the imputation part applies at a fixed set of parameters $\boldsymbol{\omega}^t$ an imputation by the mean, or a shared-weights

Table 8: Average and standard deviation values of imputation quality metrics (rounded to the closest second decimal place) and runtime across 10 different random seeds. RMSE: root mean square error. MAE: mean average error. WD: Wasserstein distance. Runtime is in seconds. Bold type is the top performer, underline denotes the second best (and average percentage of deterioration of performance across metrics compared to the top performer). We use the MNAR setting with 30% missing rate. KNN is the K-nearest neighbor algorithm with weights inversely proportional to the distance to the neighbor, KNN-Unif is the K-nearest neighbor algorithm with uniform weights.

| Data set | Alg. | RMSE ↓ | MAE ↓ | WD ↓ | Runtime ↓ |
|---|---|---|---|---|---|
| Spambase | **F3I (ours)** | 0.06 ±0.01 | 0.02 ±0.00 | 0.02 ±0.00 | 26.04 ±7.80 |
| | GAIN | 0.06 ±0.01 | 0.02 ±0.00 | 0.04 ±0.01 | 5.80 ±1.45 |
| | GRAPE | 0.07 ±0.01 | 0.03 ±0.00 | 0.04 ±0.01 | 4,090 ±691 |
| | KNN | **0.05 ±0.00** | **0.01 ±0.00** | **0.01 ±0.00** | 2.91 ±0.66 |
| | KNN-Unif | **0.05 ±0.00** | **0.01 ±0.00** | 0.02 ±0.00 | **1.99 ±0.25** |
| | HyperImpute | **0.05 ±0.00** | 0.02 ±0.00 | 0.03 ±0.01 | 22.01 ±4.51 |
| | MIRACLE | **0.05 ±0.00** | 0.03 ±0.00 | 0.05 ±0.00 | 51.76 ±4.37 |
| | NewImp | 0.17 ±0.07 | 0.11 ±0.03 | 0.27 ±0.07 | 54,190 ±9,704 |
| | Remasker | **0.05 ±0.00** | 0.02 ±0.00 | 0.03 ±0.00 | 5,507 ±676 |
| | TDM | 0.22 ±0.01 | 0.17 ±0.01 | 0.38 ±0.02 | 246.16 ±76.29 |
| Wine (White) | **F3I (ours)** | 0.11 ±0.01 | 0.07 ±0.01 | 0.06 ±0.01 | 19.20 ±4.51 |
| | GAIN | 0.10 ±0.01 | 0.07 ±0.00 | 0.12 ±0.01 | 5.84 ±1.30 |
| | GRAPE | 0.19 ±0.01 | 0.15 ±0.01 | 0.24 ±0.02 | 1,047 ±145 |
| | KNN | **0.08 ±0.01** | **0.05 ±0.00** | 0.05 ±0.00 | 2.12 ±0.34 |
| | KNN-Unif | 0.09 ±0.00 | 0.06 ±0.00 | 0.06 ±0.00 | **1.95 ±0.34** |
| | HyperImpute | **0.08 ±0.00** | **0.05 ±0.00** | **0.04 ±0.00** | 21.53 ±4.82 |
| | MIRACLE | 0.09 ±0.00 | 0.06 ±0.00 | 0.07 ±0.00 | 28.98 ±6.82 |
| | NewImp | 0.38 ±0.03 | 0.31 ±0.02 | 0.75 ±0.06 | 32,375 ±,3509 |
| | Remasker | **0.08 ±0.00** | **0.05 ±0.00** | 0.06 ±0.00 | 1,340 ±224 |
| | TDM | 0.16 ±0.01 | 0.13 ±0.01 | 0.12 ±0.02 | 170.92 ±29.28 |
| Wine (Red) | **F3I (ours)** | 0.13 ±0.01 | 0.09 ±0.01 | 0.09 ±0.00 | 2.29 ±0.43 |
| | GAIN | 0.13 ±0.01 | 0.09 ±0.01 | 0.14 ±0.01 | 5.90 ±2.45 |
| | GRAPE | 0.24 ±0.01 | 0.19 ±0.01 | 0.28 ±0.02 | 431 ±15 |
| | KNN | **0.10 ±0.00** | **0.06 ±0.00** | 0.07 ±0.01 | **0.17 ±0.03** |
| | KNN-Unif | 0.11 ±0.00 | 0.07 ±0.00 | 0.09 ±0.01 | **0.17 ±0.02** |
| | HyperImpute | **0.10 ±0.00** | 0.07 ±0.00 | **0.05 ±0.00** | 20.91 ±5.04 |
| | MIRACLE | 0.21 ±0.01 | 0.16 ±0.01 | 0.16 ±0.02 | 11.42 ±0.94 |
| | NewImp | 0.50 ±0.03 | 0.42 ±0.03 | 1.01 ±0.07 | 5,846 ±377 |
| | Remasker | **0.10 ±0.00** | 0.07 ±0.00 | 0.08 ±0.01 | 639.05 ±146.41 |
| | TDM | 0.19 ±0.01 | 0.14 ±0.01 | 0.10 ±0.01 | 157.67 ±15.86 |
| Libras | **F3I (ours)** | 0.08 ± 0.00 | 0.07 ± 0.00 | 0.06 ± 0.00 | 0.51 ± 0.15 |
| | GAIN | 0.05 ± 0.00 | 0.05 ± 0.00 | 0.03 ± 0.00 | 36.06 ± 6.72 |
| | GRAPE | 0.29 ± 0.01 | 0.14 ± 0.01 | 0.24 ± 0.01 | 7,039 ± 795 |
| | KNN | 0.09 ± 0.00 | 0.08 ± 0.01 | 0.07 ± 0.00 | **0.02 ± 0.00** |
| | KNN-Unif | 0.09 ± 0.00 | 0.08 ± 0.00 | 0.07 ± 0.00 | **0.02 ± 0.00** |
| | HyperImpute | **0.02 ± 0.00** | **0.02 ± 0.00** | **0.01 ± 0.00** | 11.06 ± 1.28 |
| | MIRACLE | 4.11 ± 0.27 | 9.80 ± 0.76 | 4.03 ± 0.26 | 22.01 ± 0.50 |
| | NewImp | 0.31 ± 0.09 | 0.52 ± 0.12 | 0.23 ± 0.05 | 579.65 ± 67.32 |
| | Remasker | 0.19 ± 0.01 | 0.24 ± 0.02 | 0.15 ±± 0.01 | 2,123 ± 271 |
| | TDM | 0.31 ± 0.01 | 0.14 ± 0.01 | 0.25 ± 0.00 | 665.01 ± 106.04 |

Table 9: Empirical validation of Theorem 5.1 by comparing the value of the upper bound and the average and standard deviation value of the cumulative regret $\max_{\boldsymbol{\alpha}\in\triangle_K}\sum_{s=1}^{t}\overline{G}(\boldsymbol{\alpha},X^{s-1})-\overline{G}(\boldsymbol{\alpha}^s,X^{s-1})$ across iterations where $t$ is the final round for F3I in the last epoch. All values are rounded to the closest second decimal place. The time round is fixed to $T=3$ in PCGrad-F3I and $\beta=0.5$. Theorem 4.4 states that $\max_{\boldsymbol{\alpha}\in\triangle_K}\sum_{s=1}^{t}\overline{G}(\boldsymbol{\alpha},X^{s-1})-\overline{G}(\boldsymbol{\alpha}^s,X^{s-1}) \leq C_{(G,\ell)}^{\mathrm{AH}}\sqrt{t}+(1-\beta)H^{\mathrm{miss}}h^{-1}t$ with probability $1-1/2,500\approx 0.9996$.

| Missingness type | | $t$ | Cumulative regret on $\overline{G}$ | $C_{(G,\ell)}^{\mathrm{AH}}\sqrt{t}+(1-\beta)H^{\mathrm{miss}}h^{-1}t$ |
|---|---|---|---|---|
| MCAR | (Assumption B.2) | 3 | 2.31 $\pm$0.03 | 6.38 $\pm$0.22 |
| MAR | (Assumption B.3) | 3 | 2.21 $\pm$0.01 | 6.37 $\pm$0.22 |
| MNAR | (Assumption B.4) | 3 | 2.21 $\pm$0.00 | 6.38 $\pm$0.22 |

Table 10: Hyperparameter values for the training of the MLP block for each imputation method (NeuMiss, mean imputation, F3I).

| **Hyperparameter** | # epochs | Weight decay in the optimizer | Learning rate | MLP depth |
|---|---|---|---|---|
| **Value** | 10 | 0 | 0.01 | 1 layer |

NeuMiss block, as introduced in Le Morvan et al. (2021). The criterion for training the model is the log loss, and we split the samples into training (70%), validation (20%), and testing (10%) sets, where the former two sets are used for training the MLP, and the performance metrics are computed on the latter set. We consider the classical Area Under the Curve (AUC) on the test set (hidden during training) as the performance metric for the binary classification task. In this section, we set the number of imputation rounds in F3I to $T=2$, and we train the MLPs for each imputation approach with the hyperparameter values reported in Table 10.

Figure 7 shows the results for MCAR (Assumption B.2), MAR (Assumption B.3), and MCAR (Assumption B.4) synthetic data with an approximate missingness frequency of 50% and varying values of $\beta\in\{0.25,0.5,0.75\}$. Figure 8 shows the corresponding results when the approximated missingness frequency is in $\{25\%,75\%\}$. There is a significant improvement in PCGrad-F3I over the baseline NeuMiss. However, the imputation by the mean remains the top contender on the synthetic Gaussian data sets, as already noticed for imputation in the previous paragraph, even if PCGrad-F3I is sometimes on par regarding classification performance.

### H.2 Real-world data sets (drug repurposing & handritten-digit recognition)

In addition to the synthetic Gaussian data sets, we also evaluate the performance of F3I on real-world data for drug repurposing or handwritten-digit recognition on the well-known MNIST data set (LeCun et al., 1998).

Drug repurposing aims to pair diseases and drugs based on their chemical, biological, and physical features. However, those features might be missing due to the incompleteness of medical databases or to a lack or failure of measurement. Table 11 reports the sizes of the considered drug repurposing data sets, which can be found online as indicated in their corresponding papers. A positive drug-disease pair is a therapeutic association (that is, the drug is known to treat the disease). In contrast, a negative one is associated with a failure in treating the disease or the emergence of toxic side effects.

#### H.2.1 Imputation quality and runtimes (drug repurposing task)

Drug repurposing aims to pair diseases and drugs based on their chemical, biological, and physical features. However, those features might be missing due to the incompleteness of medical databases or to a lack or failure of measurement. We consider five public drug repurposing data sets of varying sizes without missing values (see Table 11 in Appendix H.2). We add missing values with a MNAR Gaussian self-masking mechanism (Assumption B.4). We run each imputation method 100 times on the drug and the disease feature matrices with different random seeds. Note that the position of the missing values is the same across

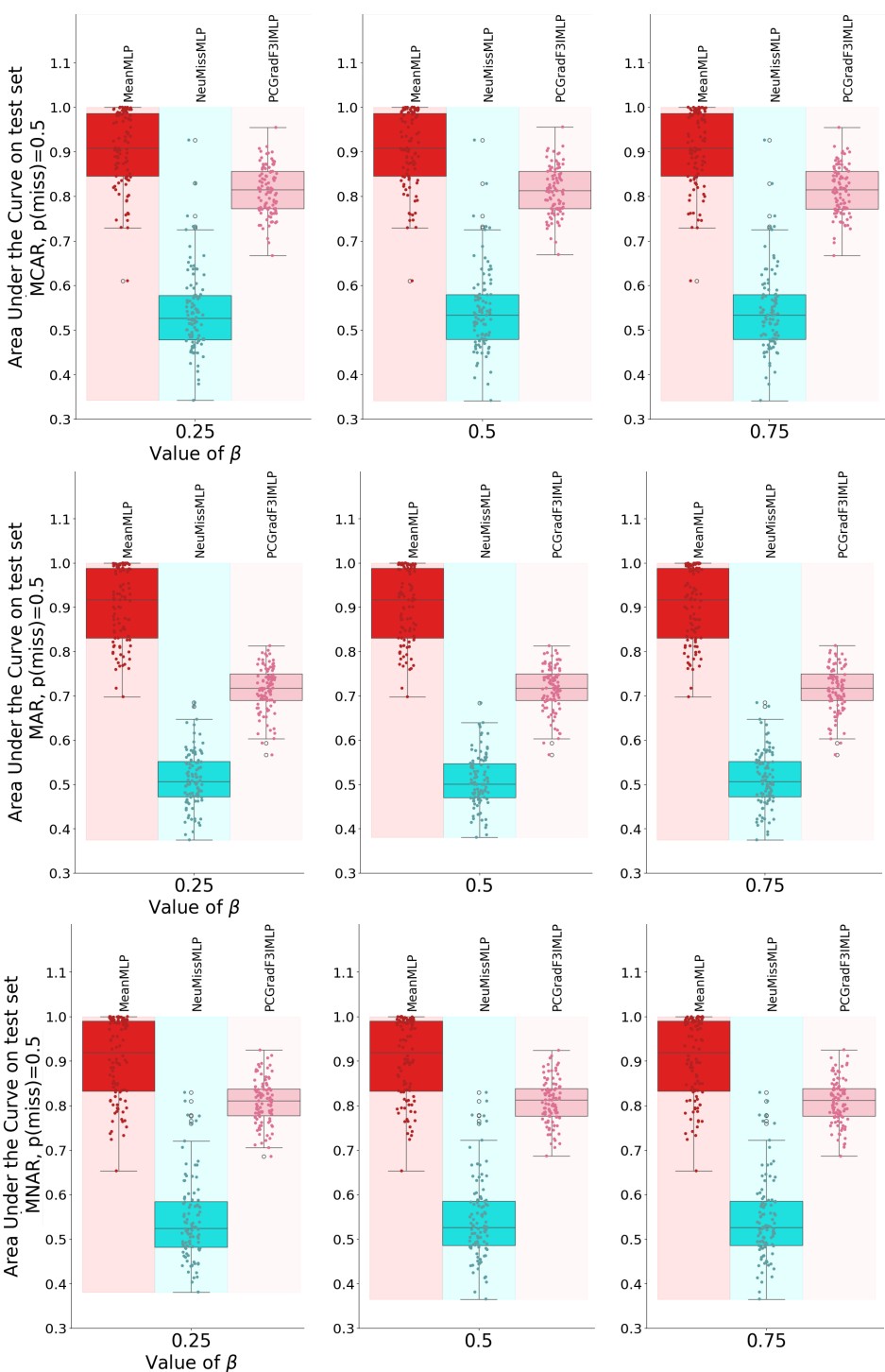

Figure 7: Joint-Imputation on a synthetic data set with MCAR (left), MAR (center) and MNAR (right) missing values and approximate missingness frequency $p^{\text{miss}} = 0.5$, for $\beta \in \{0.25, 0.5, 0.75\}$.

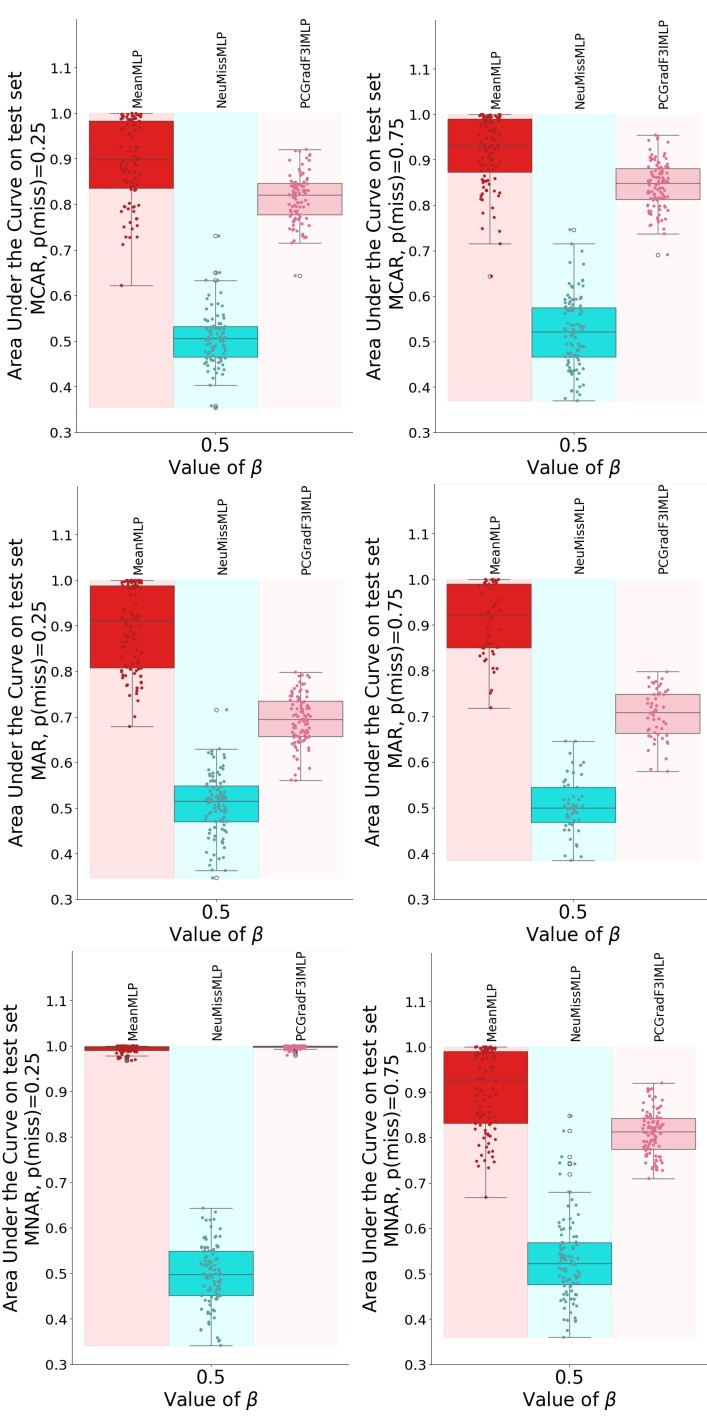

Figure 8: Joint-Imputation on a synthetic data set with MCAR (left), MAR (center) and MNAR (right) missing values and approximate missingness frequency $p^{\mathrm{miss}} \in \{0.25, 0.75\}$, for $\beta \in \{0.25, 0.5, 0.75\}$.

Table 11: Overview of the drug repurposing data sets: Cdataset (Luo et al., 2016), DNdataset (Gao et al., 2022), Gottlieb (Luo et al., 2016), PREDICT-Gottlieb (Gao et al., 2022) and TRANSCRIPT (Réda, 2023b) in the experimental study in Section H, with the number of drugs, drug features, diseases, disease features, along with the number of positive and negative drug-disease pairs.

| Name of the data set | $N_{\text{drugs}}$ | $F_{\text{drugs}}$ | $N_{\text{diseases}}$ | $F_{\text{users}}$ | Positive pairs | Negative pairs |
|---|---|---|---|---|---|---|
| Cdataset | 663 | 663 | 409 | 409 | 2,532 | 0 |
| DNdataset | 550 | 1,490 | 360 | 4,516 | 1,008 | 0 |
| Gottlieb | 593 | 593 | 313 | 313 | 1,933 | 0 |
| PREDICT-Gottlieb | 593 | 1,779 | 313 | 313 | 1,933 | 0 |
| TRANSCRIPT | 204 12,096 | 116 12,096 | 401 | 11 | | |

Table 12: Fine-tuned hyperparameter values using Optuna to train the MLP block for each imputation method (NeuMiss, Mean imputation, PCGradF3I) on the MNIST data set. K, T, $\beta$ and $\eta$ are F3I-specific parameters, whereas all remaining parameters are common to all three methods and belong to the MLP block.

| **Hyperparameter** | Number of epochs | MLP depth | K | T | $\beta$ | $\eta$ |
|---|---|---|---|---|---|---|
| **Value** | 5 | 5 layers | 17 | 13 | 0.71 | 0.053 |

runs. We considered as baselines the imputation by the feature-wise mean value (Mean), the MissForest algorithm (Stekhoven & Bühlmann, 2012), K-nearest neighbor (KNN) imputation with distance-proportional weights, where the weight is inversely proportional to the distance to the neighbor (Troyanskaya et al., 2001), an Optimal Transport-based imputer (Muzellec et al., 2020a) and finally not-MIWAE (Ipsen et al., 2021). Hyperparameter values are reported in Table 5.

Since the number of features $F \approx 12,000$ in the TRANSCRIPT dataset (Réda, 2023b) is prohibitive for most of the baselines, we reduce the number of features to $9,000$, selecting the features with the highest variance across drugs and diseases. Moreover, MissForest (Stekhoven & Bühlmann, 2012) and not-MIWAE (Ipsen et al., 2021) are too resource-consuming to be run on the largest data sets, DNdataset (Gao et al., 2022), PREDICT-Gottlieb (Gao et al., 2022) and TRANSCRIPT (Réda, 2023b). Figures 20-24 show the boxplots of average mean squared errors and runtimes of each algorithm across the 100 iterations for both drug and disease feature matrices. Table 18 shows the corresponding numerical results (average values ±standard deviations across the 100 iterations).

Overall, F3I has a runtime comparable to the fastest baselines, that is, the Optimal Transport-based imputer (Muzellec et al., 2020a) (OT in plots), the imputation by the mean value (Mean) and the k-nearest neighbor approach (Troyanskaya et al., 2001) (KNN), while having a performance in imputation which is on par with the best state-of-the-art algorithm MissForest (Stekhoven & Bühlmann, 2012), as reported by several prior works (Emmanuel et al., 2021; Joel et al., 2024). However, MissForest is several orders of magnitude slower than F3I and sometimes cannot be run at all (for instance, for the highly-dimensional TRANSCRIPT data set). The Optimal Transport imputer also performs well across the data sets and often competes with our contribution F3I in imputation and computational efficiency.

### H.2.2 Classification quality (handwritten-digit recognition task)

Again, we compare the performance of PCGrad-F3I with a simple mean imputation or NeuMiss (Le Morvan et al., 2020; 2021), chaining the corresponding imputation part with an MLP as previously done on synthetic data sets in Subsection H.1. The training procedure of the full architecture is provided in the code. We consider the MNIST dataset (LeCun et al., 1998), which comprises grayscale images of $25 \times 25$ pixels. We restrict our study to images annotated with class 0 or class 1 to get a binary classification problem. We remove pixels at random with probability 50% using a MCAR mechanism (Assumption B.2).

Table 13: Area Under the Curve (AUC) values (average ±standard deviation) in the testing subset in MNIST (hidden during the training phase) and corresponding tuned hyperparameter values (rounded up to the $2^{\text{nd}}$ decimal place for values in $\mathbb{R}$) for $N = 100$ iterations. NeuMiss has been trained on the same number of epochs and the same MLP architecture as PCGradF3I and the mean imputation followed by the MLP (Mean imputation). Those are the same results displayed in the third column of Table 4.

| Type | $p^{\text{miss}}$ | Algorithm | AUC |
|---|---|---|---|
| MCAR (Assumption B.2) | 50% | GRAPE (You et al., 2020) | **1.00 ±0.00** |
| | | K-NN (Troyanskaya et al., 2001) | 0.93 ±0.17 |
| | | Mean | 0.64 ±0.18 |
| | | NeuMiss (Le Morvan et al., 2020) | 0.99 ±0.07 |
| | | PCGradF3I (**ours**) | 0.99 ±0.09 |
| | | RF-GAP (Rhodes et al., 2023) | **1.00 ±0.00** |

**Hyperparameter tuning and importance.** We employed the Optuna framework to optimize our model and the baselines's hyperparameters (Akiba et al., 2019). The optimization process focused on tuning several key parameters: $\beta, \eta, T, K$, and the depth of the classifier MLP. For the hyperparameter search, we utilized Optuna's default Tree-structured Parzen Estimator-based sampler (Bergstra et al., 2011), conducting 50 trials to explore the parameter space. The dataset was evenly divided into three portions, with 34% allocated for training, 33% for validation, and 33% for testing. During the optimization process, we aimed to maximize the logarithm of the Area Under the Curve (AUC) scores from the Receiver Operating Characteristic (ROC) curve on the validation set. After identifying the optimal hyperparameter configuration, we constructed the final model. Final hyperparameter values for our model are reported in Table 12.

We also estimated the importance of each hyperparameter on the objective function using the functional analysis of variance, or fANOVA (Hutter et al., 2014). fANOVA estimates the percentage of variance in the classification performance on the validation set explained by each hyperparameter, given a regression tree and a hyperparameter space. Then, the larger the percentage, the greater impact on the classification performance on the validation set. The corresponding values are listed in Table 14.

Table 14: Percentages of variance of the Area Under the Curve (AUC) on the validation set explained by each hyperparameter finetuned on the MNIST data set for PCGradF3I. Hyperparameters are listed in the order of decreasing explained variance.

| Hyperparameter | Number of epochs | $\eta$ | K | T | $\beta$ | MLP depth |
|---|---|---|---|---|---|---|
| **Explained variance (%)** | 38.1 | 32.6 | 12.5 | 9.6 | 4.1 | 2.9 |

We observe that the number of training epochs for the MLP and the regularization factor $\eta$ for the weight vector $\boldsymbol{\alpha}$ account for approximately 38% and 33% respectively of the variability of the performance across the entire hyperparameter space. The number $K$ of nearest neighbours chosen for the convex combination is also relatively important. This is expected, as the number of neighbors controls the quality of the estimation of the data distribution in the imputation algorithm. Surprisingly, the number of iterations $T$ explains less than 10% of the variance in the performance, demonstrating that F3I probably reaches the early stopping criterion (Line 16 in Algorithm 1) very quickly, without exhausting the budget $T$. Another surprising observation is the relatively low importance for the classification performance of $\beta$ and the depth of the MLP. This shows that even a relatively simple classifier can achieve higher classification accuracy with good imputation quality on the MNIST data set restricted to the classes 0 and 1.

**Training.** This optimized model underwent training using the designated training set, followed by performance evaluation. We assessed its performance by measuring the AUC score of the ROC curve on the test set, repeating this evaluation process across 100 iterations to ensure robust results.

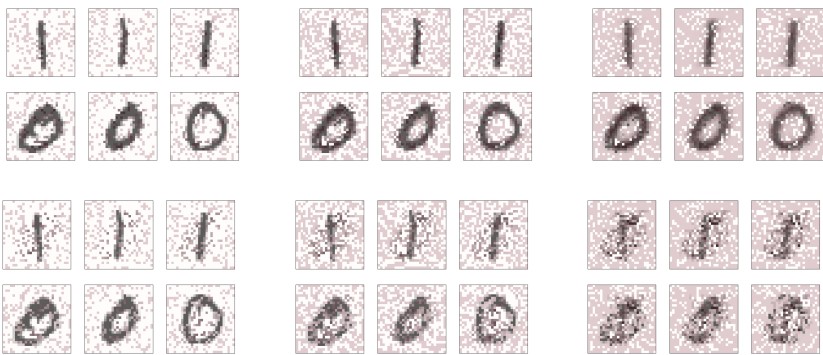

Figure 9: Imputed grayscale images by F3I (first two rows) or mean imputation (last two rows) for the first 6 samples (trained on the first 600 samples of MNIST with the hyperparameters in Table 12) with MCAR-missing pixels, with missingness frequencies in $\{25\%, 50\%, 75\%\}$. Columns 1 to 3 correspond to $p^{\text{miss}} = 25\%$, columns 4 to 6 to $p^{\text{miss}} = 50\%$, and columns 7 to 9 to $p^{\text{miss}} = 75\%$. Positions of red pixels represent missing pixels during the training phase which are imputed by either F3I or mean imputation.

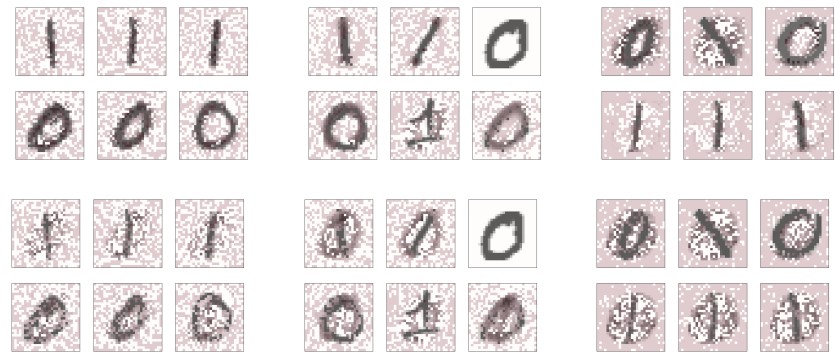

Figure 10: Imputed grayscale images by F3I (first two rows) or mean imputation (last two rows) for the first 6 samples (trained on the first 600 samples of MNIST with the hyperparameters in Table 12). Positions of red pixels represent missing pixels during the training phase which are imputed by either F3I or mean imputation. Columns 1 to 3 correspond to MCAR-missing pixels (Assumption B.2), columns 4 to 6 to MAR-missing pixels (Assumption B.3), and columns 7 to 9 to MNAR-missing pixels (Assumption B.4). Note that in one of the samples with MAR-missing pixels, no pixel is missing, which is due to randomness in the generation of missing pixels.

**Results.** We report the numerical results in Table 13. We also display the imputed images for the first 6 samples in MNIST by F3I or by mean imputation, trained on the first 600 samples. [4] We vary $p^{\text{miss}} \in \{25\%, 50\%, 75\%\}$ in Figure 9. Finally, we modify the missingness mechanism in Figure 10, switching the MCAR missingness mechanism to MAR (Assumption B.3) or MNAR (Assumption B.4).

As mentioned in the main text, PCGradF3I beats the mean imputation and NeuMiss regarding classification accuracy according to Table 13. It also preserves a good imputation of the MNIST images compared to the mean imputation, even when the number of missing values increases (see Figure 9). Even if one still can distinguish between ones and zeroes with the mean imputation, there is a higher confidence in the predicted labels when looking at F3I-imputed images. Moreover, F3I turns out to be more robust to the different types of missingness mechanisms compared to the mean imputation, as illustrated by Figure 10. For missing-completely-at-random pixels, both methods fare good regarding imputation. However, the performance of the mean imputation is limited in the case of MAR or MNAR-missing pixels (Columns 4 et 6), as most samples represent both a 0 and a 1.

---

[4]The NeuMiss network (Le Morvan et al., 2020) does not perform imputation, only classification or regression.

Table 15: Overview of the drug repurposing data set PREDICT for joint imputation-classification experiments, with the number of drugs, drug features, percentage of missing drug data, diseases, disease features, percentage of missing disease features, along with the number of positive and negative drug-disease pairs.

| Data set | $N_{\text{drugs}}$ | $F_{\text{drugs}}$ (% missing) | | $N_{\text{diseases}}$ | $F_{\text{users}}$ (% missing) | | Pos | Neg |
|---|---|---|---|---|---|---|---|---|
| PREDICT (Réda, 2023a) | 1,150 | 1,642 | (24) | 1,028 | 1,490 | (26) | 4,627 | 132 |
| PREDICT (reduced) | 175 | 326 | (36) | 175 | 215 | (60) | 454 | 0 |

Table 16: Fine-tuned hyperparameter values using Optuna to train the MLP block for each imputation method (NeuMiss, Mean imputation, PCGradF3I) on the PREDICT data set. K, T, $\beta$ and $\eta$ are F3I-specific parameters, whereas all remaining parameters are common to all three methods and belong to the MLP block.

| Hyperparameter | Number of epochs | Learning rate | MLP depth | K | T | $\beta$ | $\eta$ |
|---|---|---|---|---|---|---|---|
| Value | 10 | 0.01 | 1 layer | 12 | 25 | 0.246 | 0.008 |

### H.2.3 Classification quality (drug repurposing task)

**Joint imputation and repurposing** This time, we consider the drug (item) and disease (user) feature matrices, along with the drug-disease association class labels from another drug repurposing data set, which natively includes missing values in the drug and disease feature matrices. This data set, named PREDICT (Réda, 2023a), is further described in Table 15. All unknown drug-disease associations are labeled 0.5, whereas positive (respectively, negative) ones are labeled $+1$ (resp., $-1$). To restrict the computational cost, we restricted the data set to its first 500 ratings (*i.e.*, drug-disease pairs) and to the 350 features with highest variance across all drugs and diseases. We also add other missing values to the data set via a MCAR mechanism –as we might have lost some missing values when reducing the data set– and run a hyperparameter optimization procedure, similarly to what has been done on the MNIST data set (see Subsection H.2.2). See Table 16 for the selected hyperparameter values.

The corresponding numerical results compared to the mean imputation and NeuMiss (with the same architecture of MLPs) is displayed in Table 17. This table shows that PCGradF3I performs slightly better than NeuMiss on this very difficult data set, while being significantly better than the naive approach relying on the imputation by the mean value. Those results confirm what we observed on the MNIST data set (see Table 13).

Table 17: Area Under the Curve (AUC) values (average ±standard deviation) in the testing subset in PREDICT (hidden during the training phase) and corresponding tuned hyperparameter values (rounded up to the $2^{\text{nd}}$ decimal place for values in $\mathbb{R}$) for $N = 100$ iterations. NeuMiss has been trained on the same number of epochs and the same MLP architecture as PCGradF3I and the mean imputation followed by the MLP (Mean imputation). Those are the same results displayed in the fourth column of Table 4.

| Type | $p^{\text{miss}}$ | Algorithm | AUC |
|---|---|---|---|
| MCAR (Assumption B.2) | 50% | GRAPE (You et al., 2020) | 0.49 ±0.07 |
| | | K-NN (Troyanskaya et al., 2001) | 0.47 ±0.07 |
| | | Mean | 0.48 ±0.00 |
| | | NeuMiss (Le Morvan et al., 2020) | 0.50 ±0.01 |
| | | PCGradF3I (**ours**) | 0.51 ±0.01 |
| | | RF-GAP (Rhodes et al., 2023) | **0.53 ±0.13** |

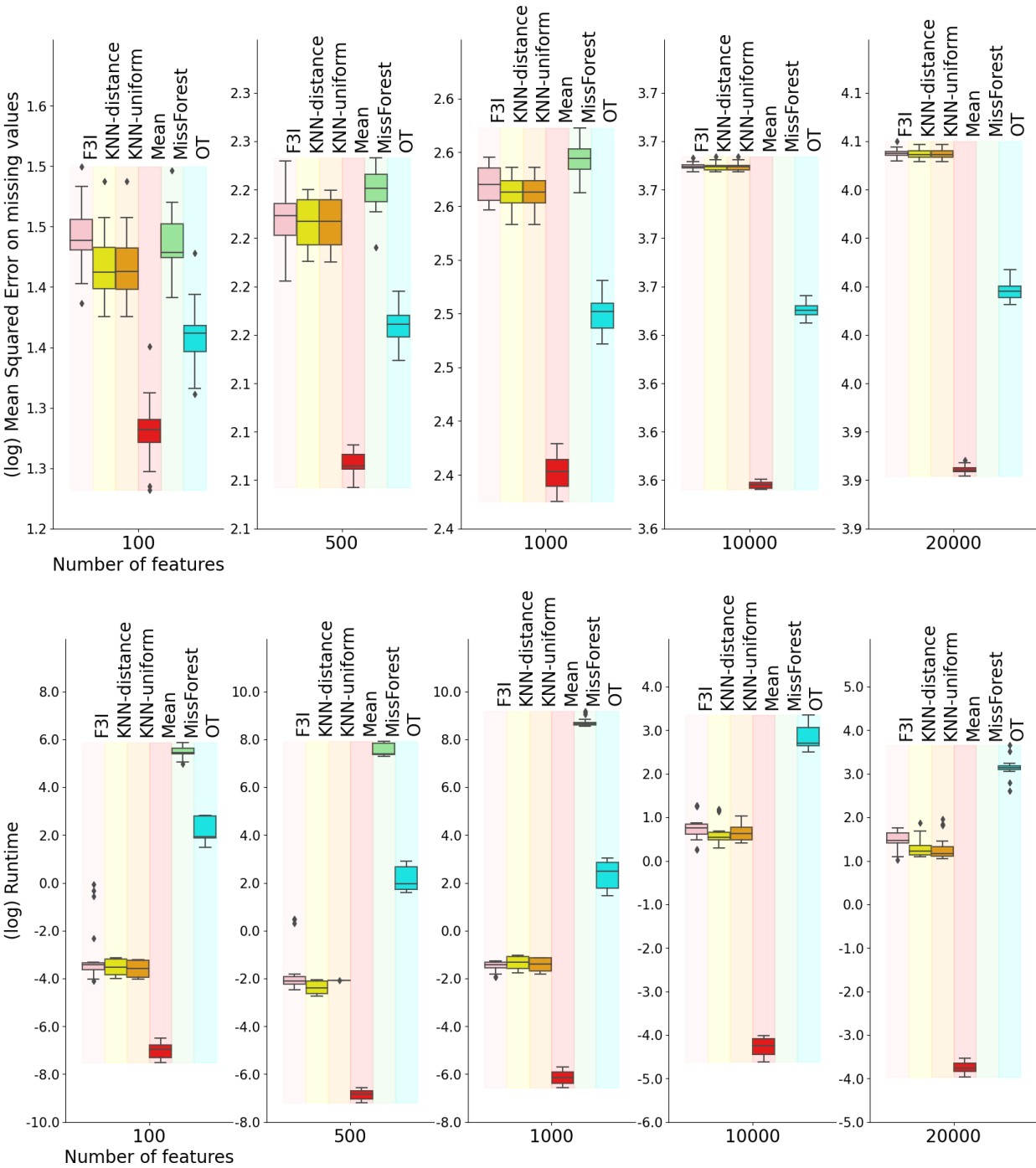

Figure 11: Imputation on 2 synthetic data sets × 10 different random seeds for generating missing values (MCAR, $p^{\mathrm{miss}} = 25\%$, K=3) for F3I, K-nearest neighbor imputers (Troyanskaya et al., 2001) (uniform or distance-based weights), mean imputation, MissForest (Stekhoven & Bühlmann, 2012) and Optimal-Transport imputer (Muzellec et al., 2020a).

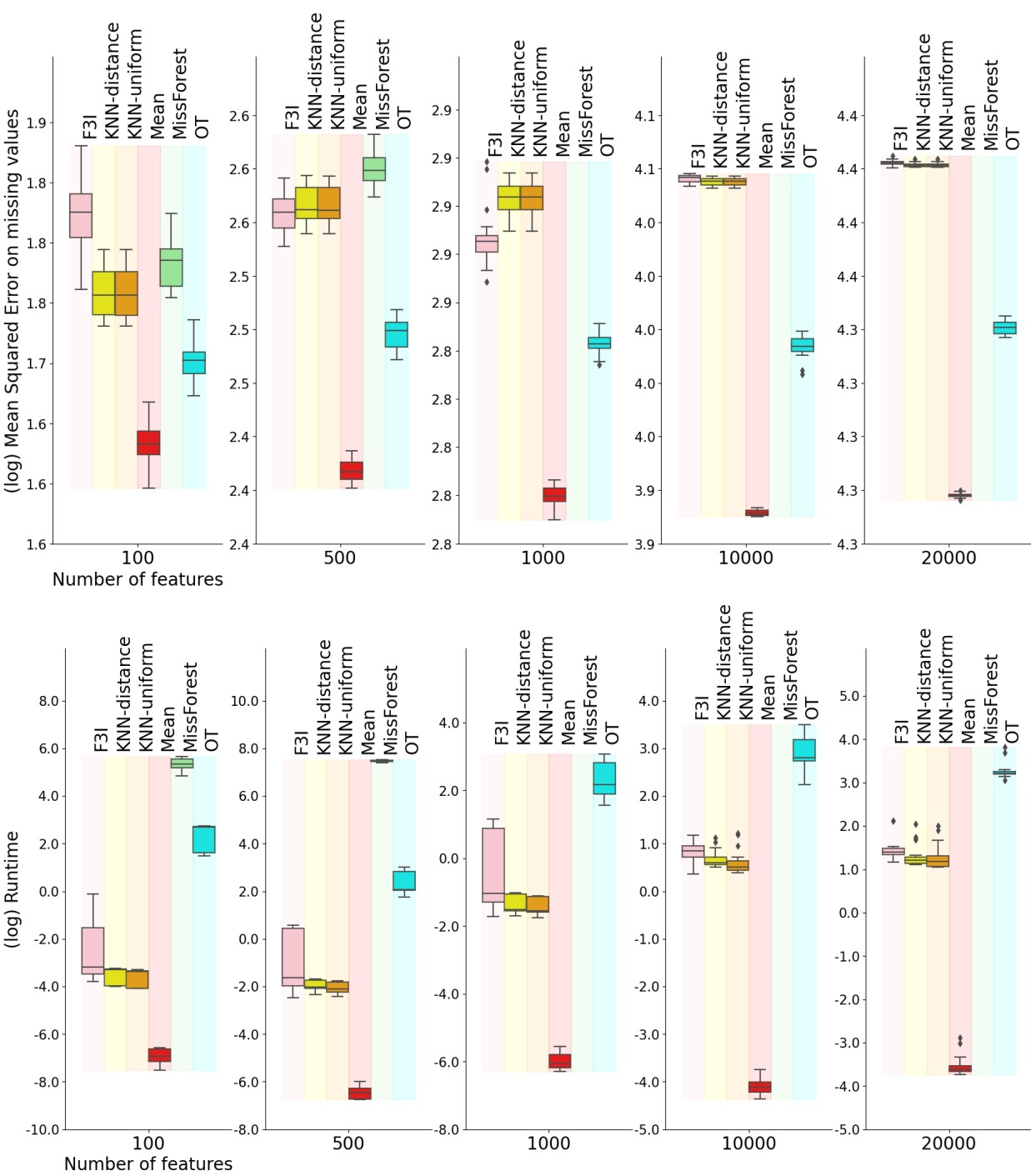

Figure 12: Imputation on 2 synthetic data sets × 10 different random seeds for generating missing values (MCAR, $p^{\text{miss}} = 50\%$, K=3) for F3I, K-nearest neighbor imputers (Troyanskaya et al., 2001) (uniform or distance-based weights), mean imputation, MissForest (Stekhoven & Bühlmann, 2012) and Optimal-Transport imputer (Muzellec et al., 2020a).

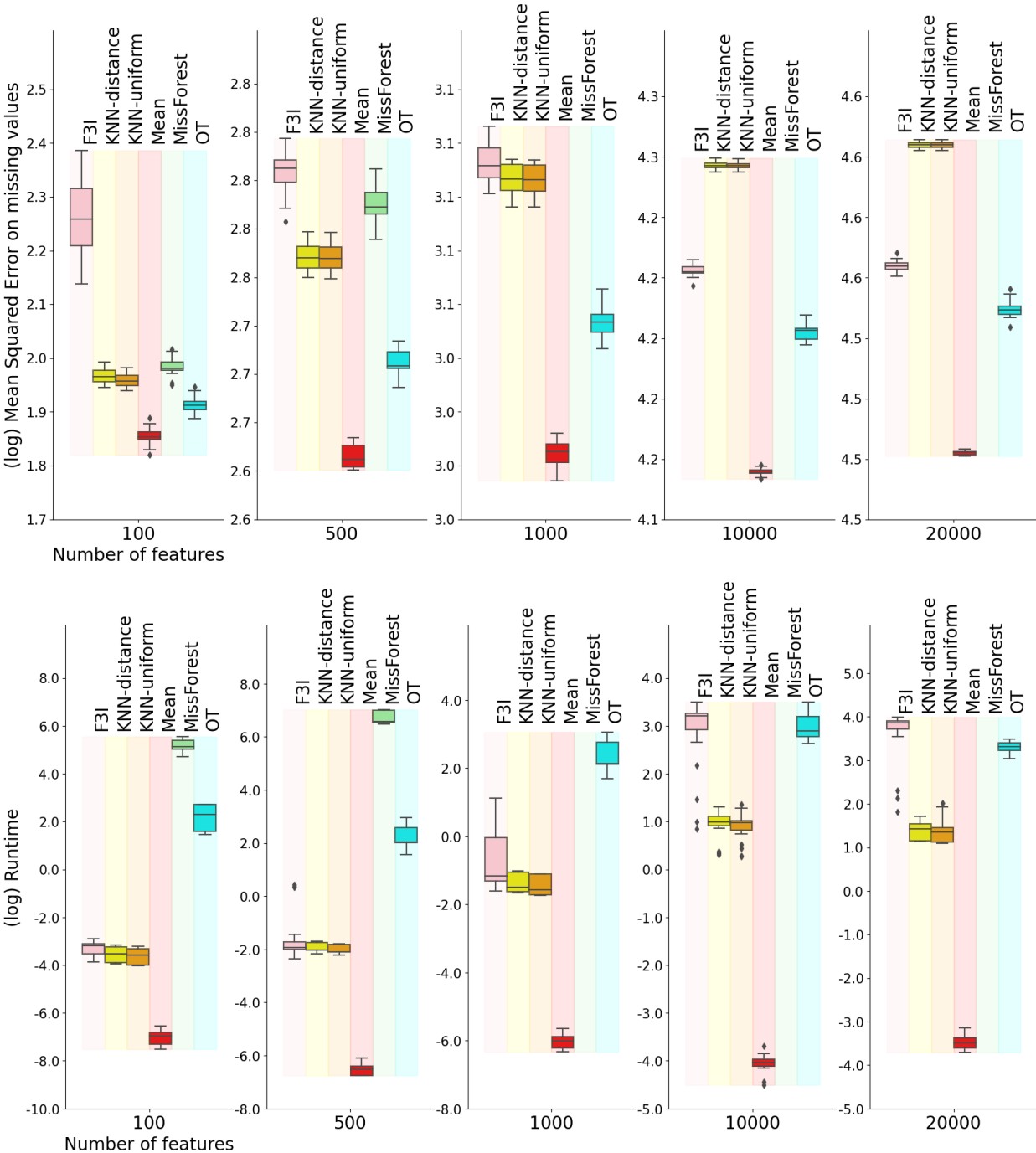

Figure 13: Imputation on 2 synthetic data sets × 10 different random seeds for generating missing values (MCAR, $p^{\mathrm{miss}} = 75\%$, K=3) for F3I, K-nearest neighbor imputers (Troyanskaya et al., 2001) (uniform or distance-based weights), mean imputation, MissForest (Stekhoven & Bühlmann, 2012) and Optimal-Transport imputer (Muzellec et al., 2020a).

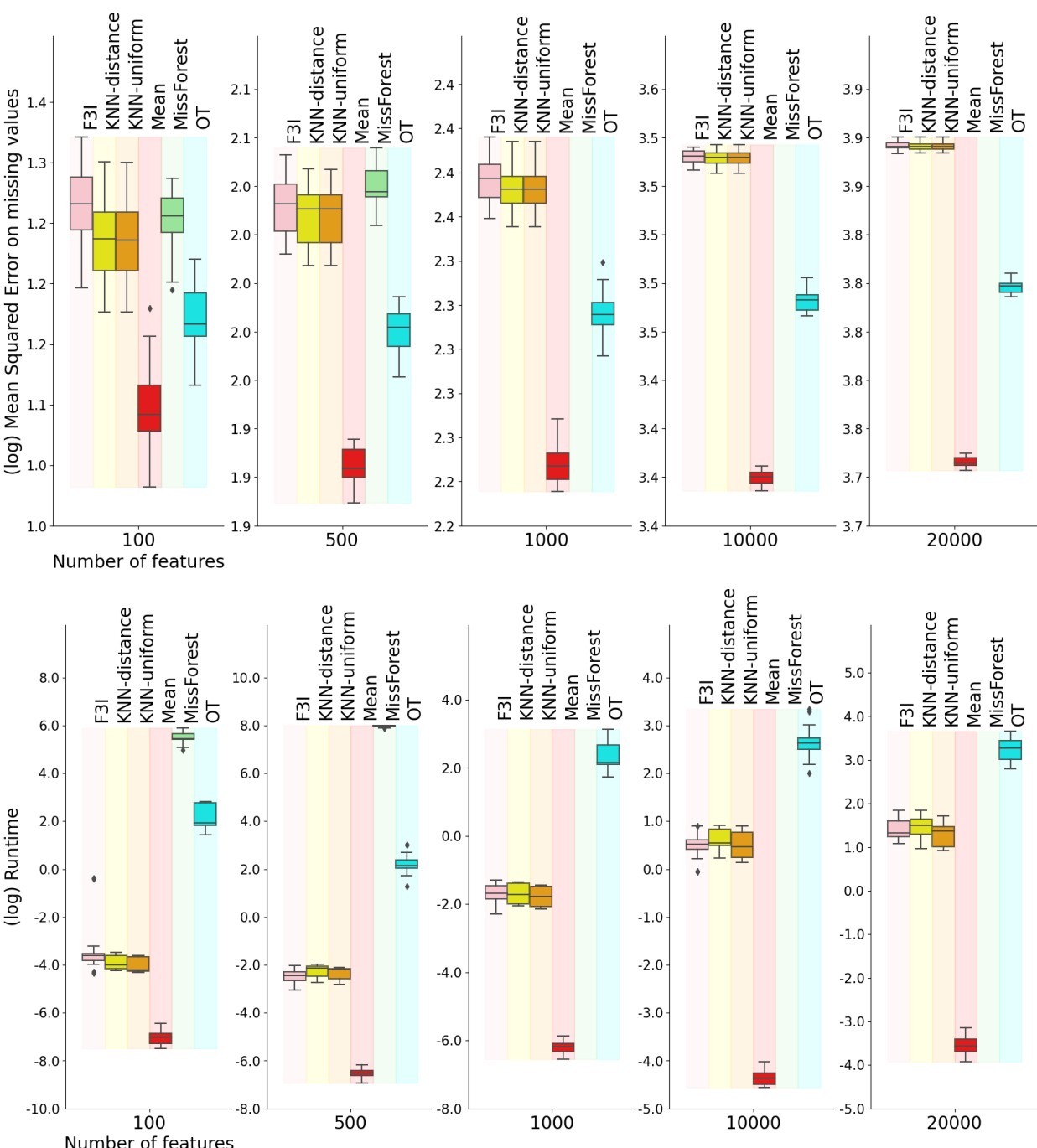

Figure 14: Imputation on 2 synthetic data sets × 10 different random seeds for generating missing values (MAR, $p^{\mathrm{miss}} = 25\%$, K=3) for F3I, K-nearest neighbor imputers (Troyanskaya et al., 2001) (uniform or distance-based weights), mean imputation, MissForest (Stekhoven & Bühlmann, 2012) and Optimal-Transport imputer (Muzellec et al., 2020a).

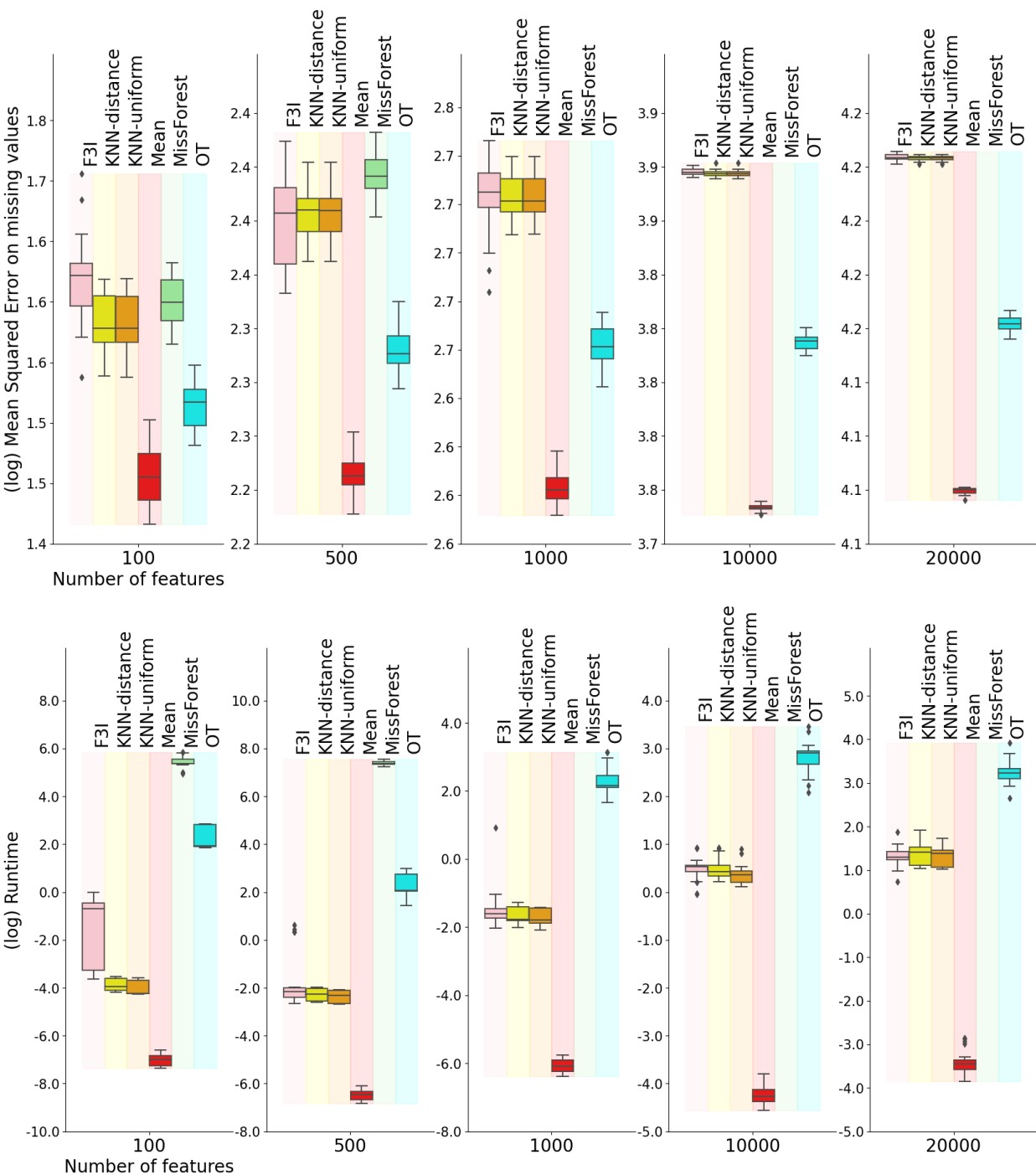

Figure 15: Imputation on 2 synthetic data sets × 10 different random seeds for generating missing values (MAR, $p^{\mathrm{miss}} = 50\%$, K=3) for F3I, K-nearest neighbor imputers (Troyanskaya et al., 2001) (uniform or distance-based weights), mean imputation, MissForest (Stekhoven & Bühlmann, 2012) and Optimal-Transport imputer (Muzellec et al., 2020a).

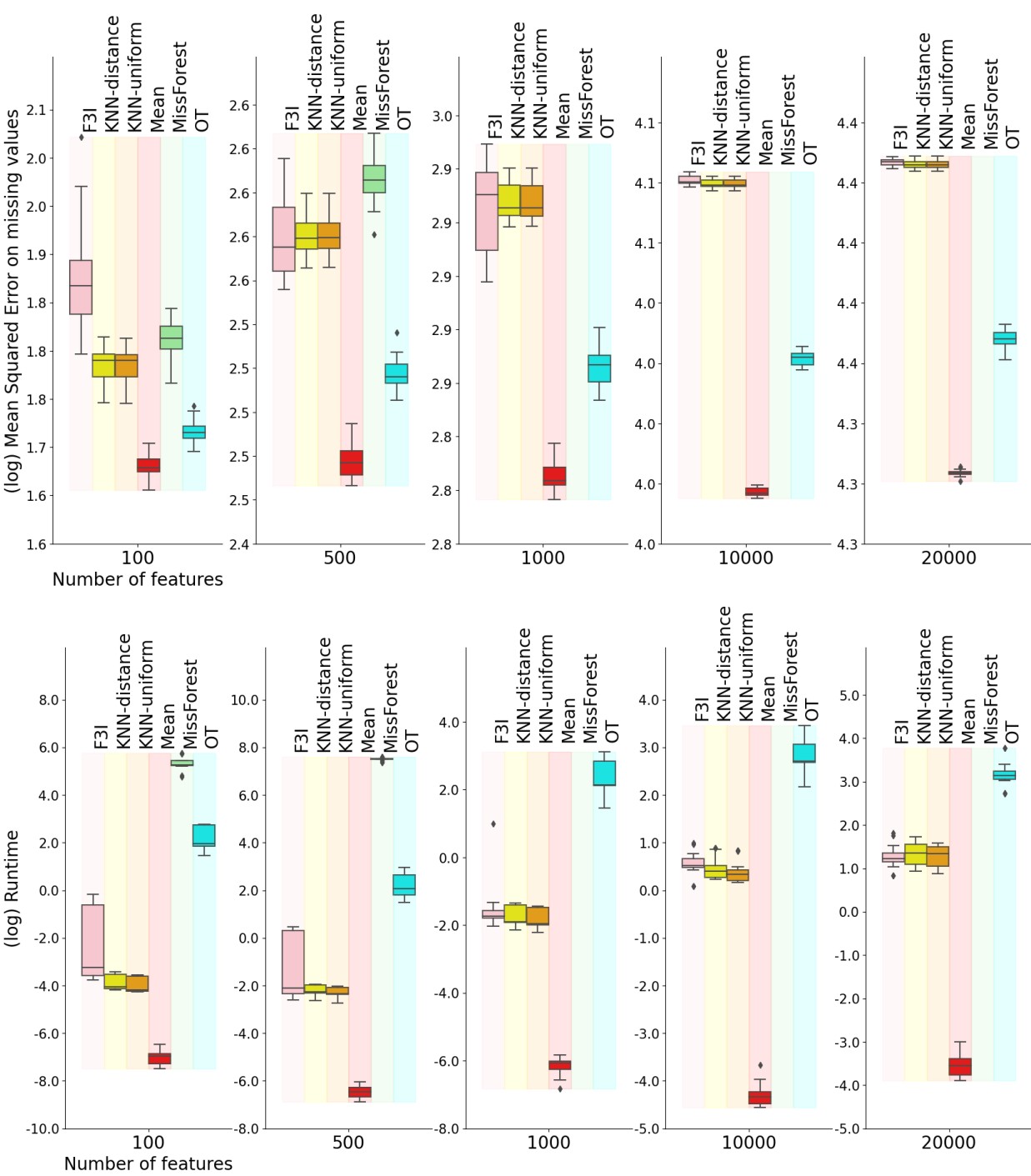

Figure 16: Imputation on 2 synthetic data sets × 10 different random seeds for generating missing values (MAR, $p^{\text{miss}} = 75\%$, K=3) for F3I, K-nearest neighbor imputers (Troyanskaya et al., 2001) (uniform or distance-based weights), mean imputation, MissForest (Stekhoven & Bühlmann, 2012) and Optimal-Transport imputer (Muzellec et al., 2020a).

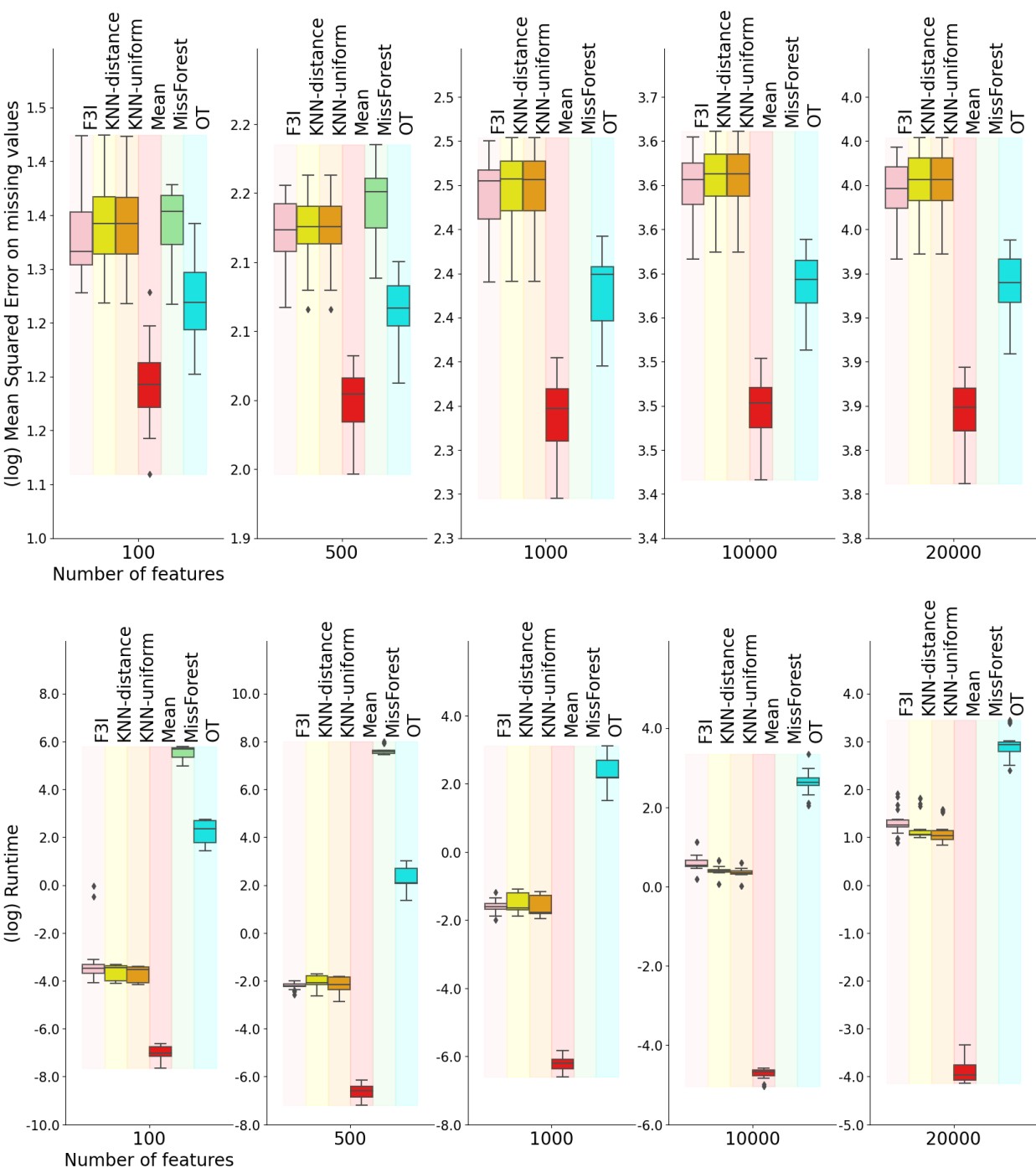

Figure 17: Imputation on 2 synthetic data sets × 10 different random seeds for generating missing values (MNAR, $p^{\text{miss}} = 25\%$, K=3) for F3I, K-nearest neighbor imputers (Troyanskaya et al., 2001) (uniform or distance-based weights), mean imputation, MissForest (Stekhoven & Bühlmann, 2012) and Optimal-Transport imputer (Muzellec et al., 2020a).

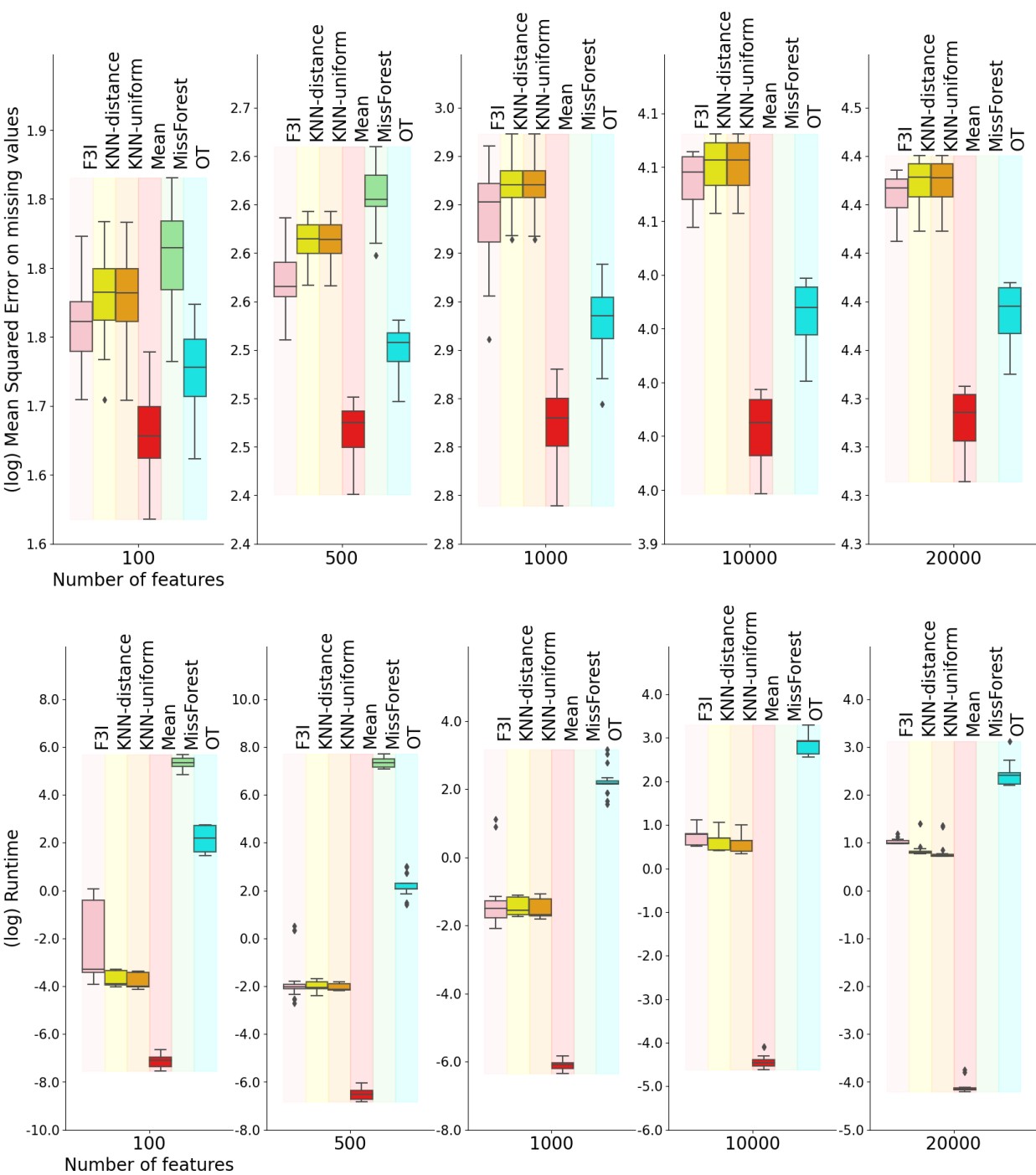

Figure 18: Imputation on 2 synthetic data sets × 10 different random seeds for generating missing values (MNAR, $p^{\mathrm{miss}} = 50\%$, K=3) for F3I, K-nearest neighbor imputers (Troyanskaya et al., 2001) (uniform or distance-based weights), mean imputation, MissForest (Stekhoven & Bühlmann, 2012) and Optimal-Transport imputer (Muzellec et al., 2020a).

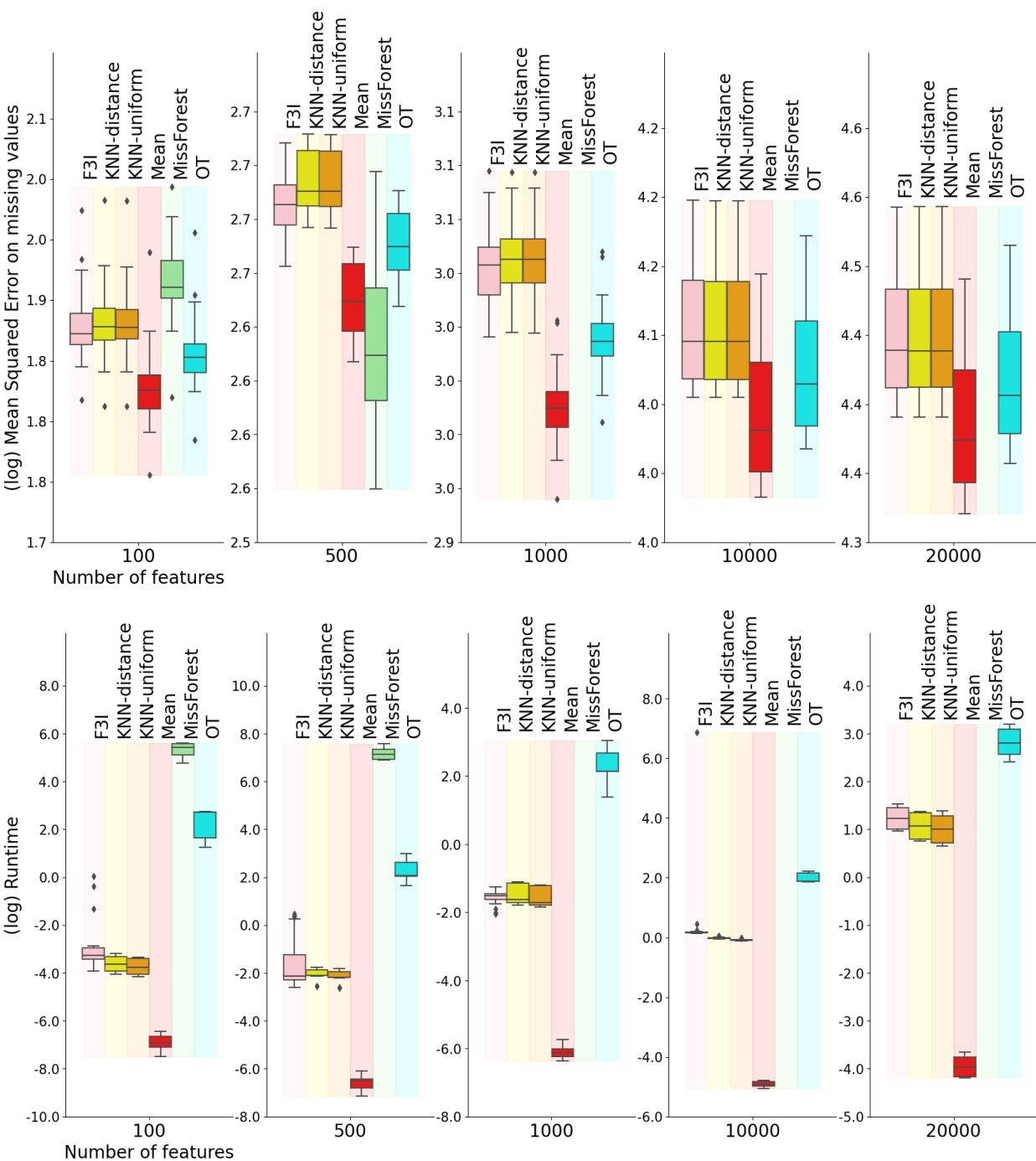

Figure 19: Imputation on 2 synthetic data sets × 10 different random seeds for generating missing values (MNAR, $p^{\mathrm{miss}} = 75\%$, K=3) for F3I, K-nearest neighbor imputers (Troyanskaya et al., 2001) (uniform or distance-based weights), mean imputation, MissForest (Stekhoven & Bühlmann, 2012) and Optimal-Transport imputer (Muzellec et al., 2020a).

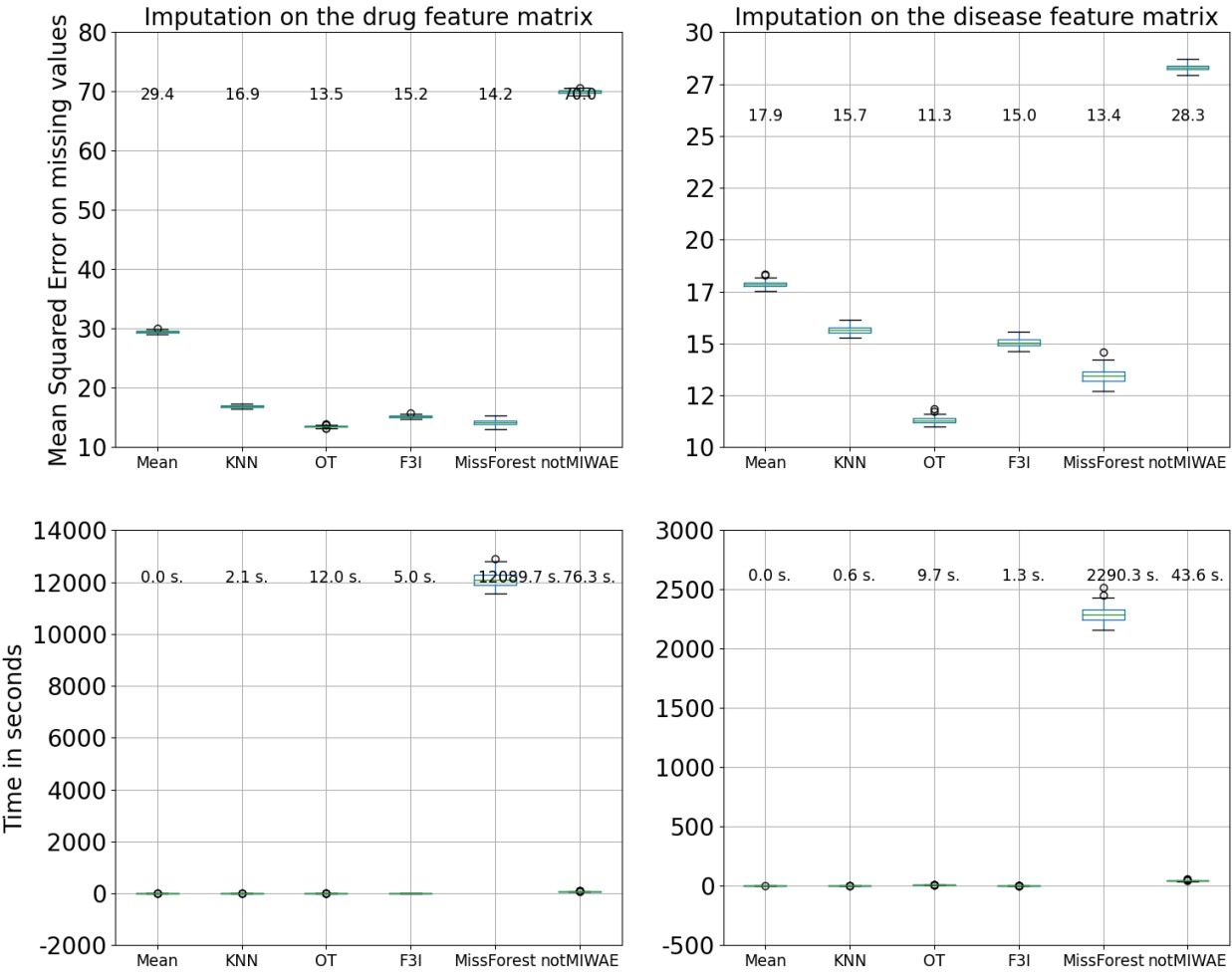

Figure 20: Imputation of missing values in the drug (left) and disease (right) feature matrices for F3I and its baselines in the Cdataset drug repurposing data set (Luo et al., 2016). The first row shows boxplots of mean-squared errors (MSE) across each algorithm's 100 iterations (with different random seeds). In contrast, the second row displays the runtimes (in seconds) across iterations for the imputation step. The average value of MSE and runtime is displayed above each corresponding boxplot. Abbreviations: OT: Optimal Transport-based imputer (Muzellec et al., 2020a), KNN: KNN imputer with distance-associated weights (Troyanskaya et al., 2001), Mean: imputation by the feature-wise mean value. Numerical results are in Table 18.

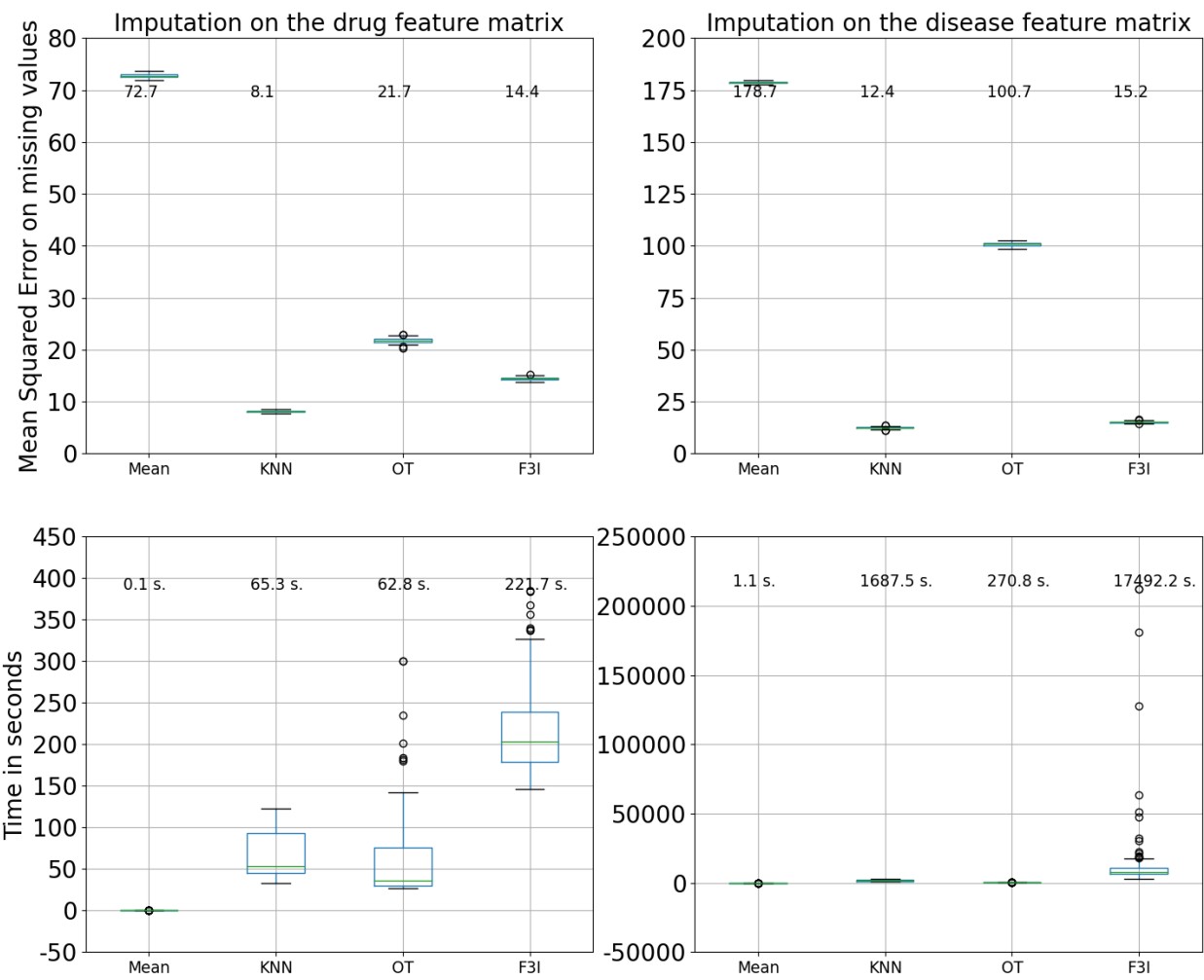

Figure 21: Imputation of missing values in the drug (left) and disease (right) feature matrices for F3I and its baselines in the DNdataset drug repurposing data set (Gao et al., 2022). The first row shows boxplots of mean-squared errors (MSE) across each algorithm's 100 iterations (with different random seeds). In contrast, the second row displays the runtimes (in seconds) across iterations for the imputation step. The average value of MSE and runtime is displayed above each corresponding boxplot. Abbreviations: OT: Optimal Transport-based imputer (Muzellec et al., 2020a), KNN: KNN imputer with distance-associated weights (Troyanskaya et al., 2001), Mean: imputation by the feature-wise mean value. Numerical results are in Table 18.

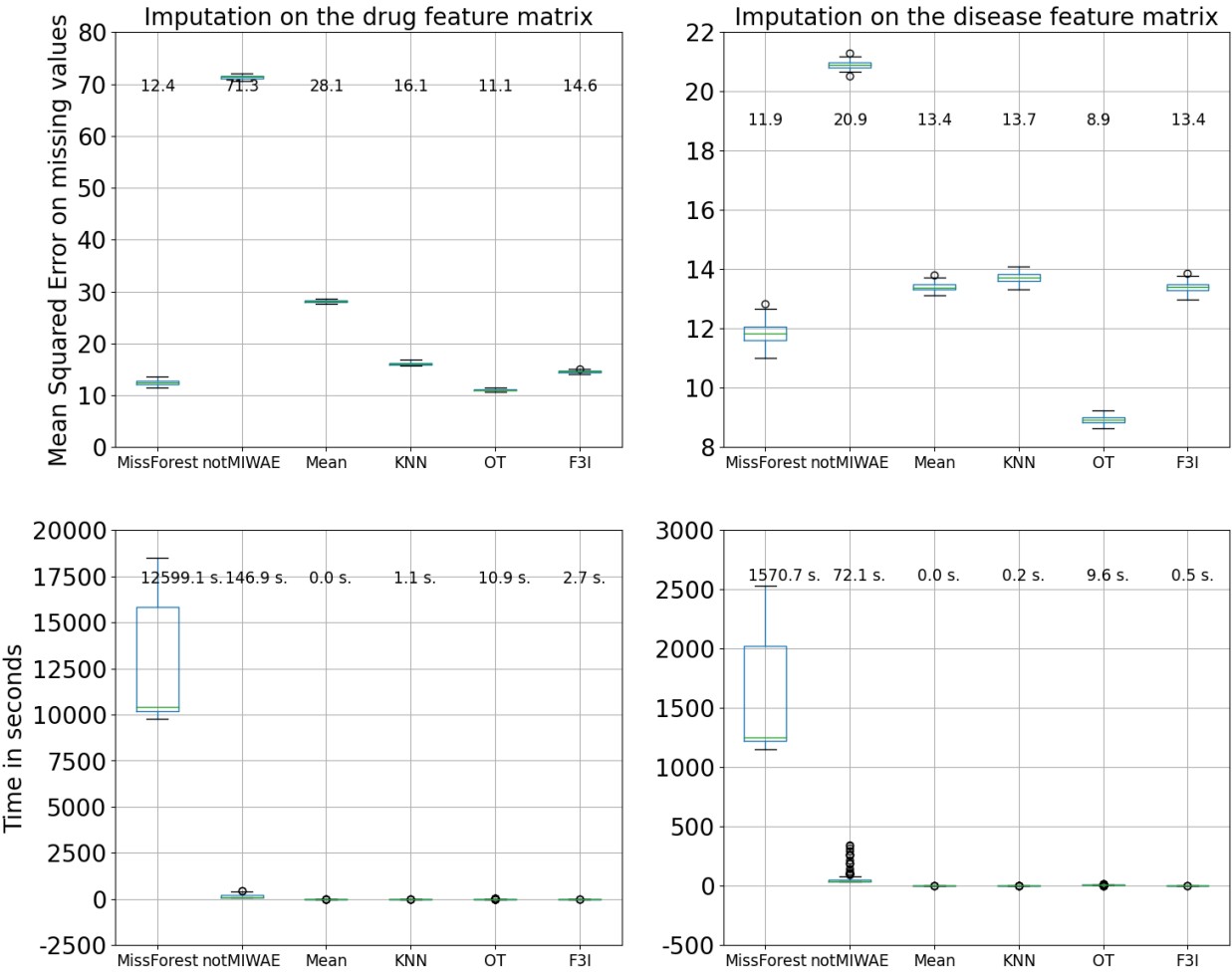

Figure 22: Imputation of missing values in the drug (left) and disease (right) feature matrices for F3I and its baselines in the Gottlieb drug repurposing data set (Luo et al., 2016). The first row shows boxplots of mean-squared errors (MSE) across each algorithm's 100 iterations (with different random seeds). In contrast, the second row displays the runtimes (in seconds) across iterations for the imputation step. The average value of MSE and runtime is displayed above each corresponding boxplot. Abbreviations: OT: Optimal Transport-based imputer (Muzellec et al., 2020a), KNN: KNN imputer with distance-associated weights (Troyanskaya et al., 2001), Mean: imputation by the feature-wise mean value. Numerical results are in Table 18.

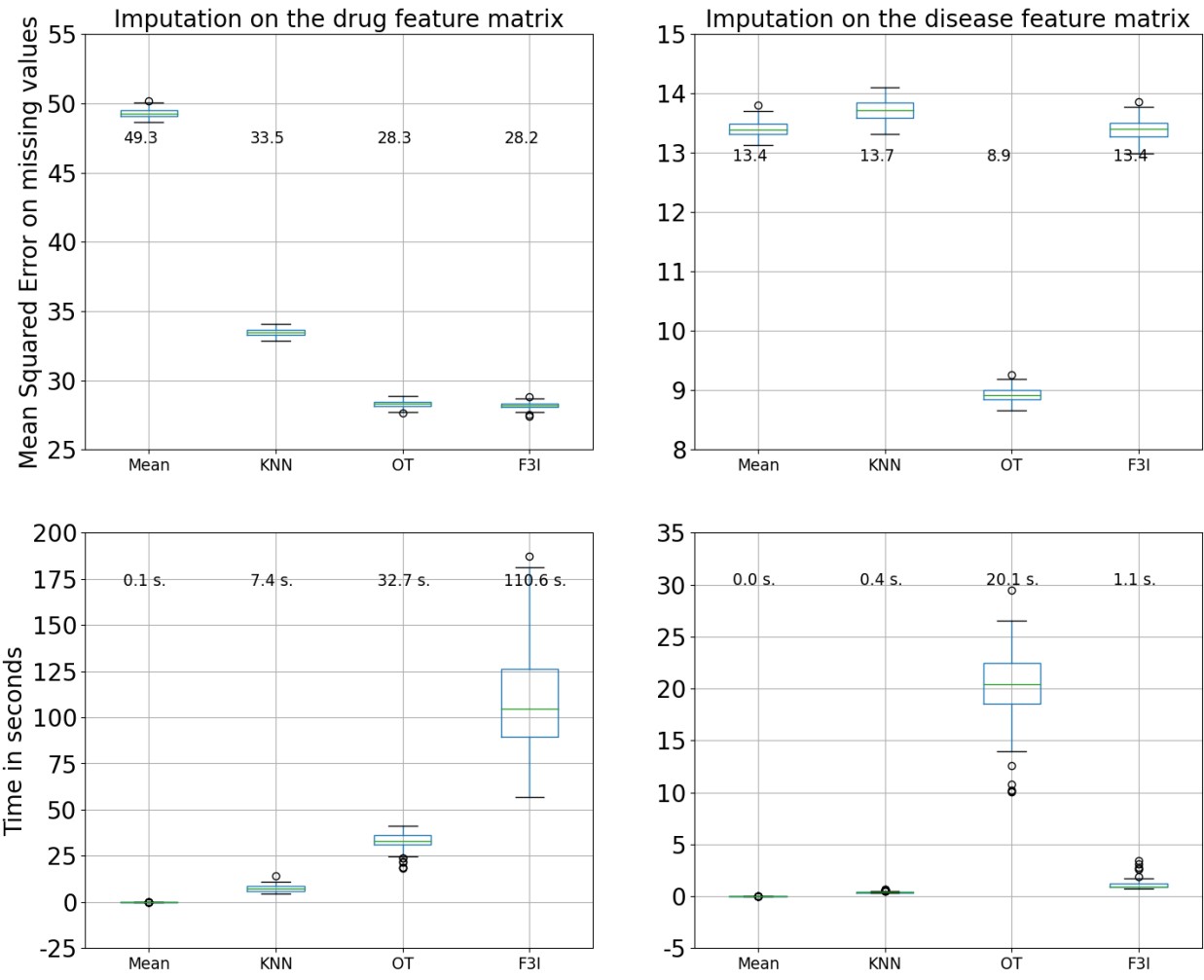

Figure 23: Imputation of missing values in the drug (left) and disease (right) feature matrices for F3I and its baselines in the PREDICT-Gottlieb drug repurposing data set (Gao et al., 2022). The first row shows boxplots of mean-squared errors (MSE) across each algorithm's 100 iterations (with different random seeds). In contrast, the second row displays the runtimes (in seconds) across iterations for the imputation step. The average value of MSE and runtime is displayed above each corresponding boxplot. Abbreviations: OT: Optimal Transport-based imputer (Muzellec et al., 2020a), KNN: KNN imputer with distance-associated weights (Troyanskaya et al., 2001), Mean: imputation by the feature-wise mean value. Numerical results are in Table 18.

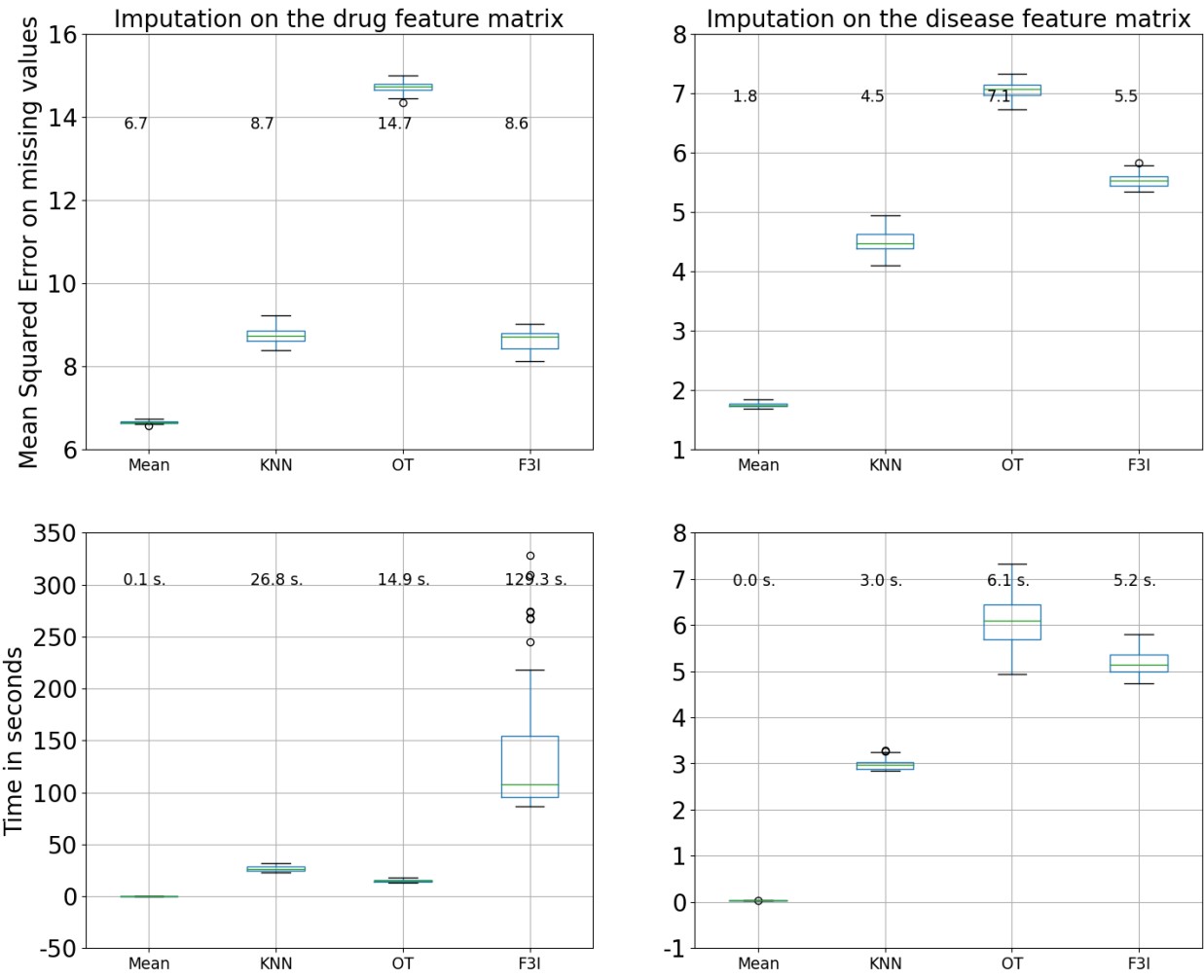

Figure 24: Imputation of missing values in the drug (left) and disease (right) feature matrices for F3I and its baselines in the TRANSCRIPT drug repurposing data set (Réda, 2023b), restricted to the 9,000 features with highest variance across samples. The first row shows boxplots of mean-squared errors (MSE) across each algorithm's 100 iterations (with different random seeds). In contrast, the second row displays the runtimes (in seconds) across iterations for the imputation step. The average value of MSE and runtime is displayed above each corresponding boxplot. Numerical results are in Table 18.

Table 18: Mean Squared Errors and runtimes (average ±standard deviation) of the imputation of missing values on drug feature matrices, for 100 random seeds, rounded to the closest $1^{st}$ decimal place. Best values are in bold type, second best values underlined.

| Data set | Algorithm | MSE ↓ | Runtime (sec.) ↓ |
|---|---|---|---|
| **Cdataset** | KNN | 16.9 ±0.2 | 2.0 ±0.2 |
| | Mean | 29.4 ±0.2 | **0.0 ±0.0** |
| | MissForest | 14.2 ±0.4 | 12,089.7 ±268.8 |
| | not-MIWAE | 70.0 ±0.3 | 76.3 ±3.8 |
| | Optimal Transport | **13.5 ±0.2** | 12.0 ±0.9 |
| | F3I | 15.2 ±0.2 | 5.0 ±0.9 |
| **DNdataset** | F3I (ours) | 14.4 ±0.3 | 221.7 ±59.3 |
| | KNN | **8.1 ±0.2** | 65.3 ±27.1 |
| | Mean | 72.7 ±0.4 | **0.1 ±0.1** |
| | MissForest | - | - |
| | not-MIWAE | - | - |
| | Optimal Transport | 21.7 ±0.5 | 62.8 ±55.1 |
| **Gottlieb** | KNN | 16.1 ±0.2 | 1.1 ±0.2 |
| | Mean | 28.1 ±0.2 | **0.0 ±0.0** |
| | MissForest | 12.4 ±0.4 | 12,599.1 ±3,115.5 |
| | not-MIWAE | 71.3 ±0.3 | 146.9 ±111.5 |
| | Optimal Transport | **11.1 ±0.2** | 10.9 ±2.6 |
| | F3I | 14.6 ±0.2 | 2.7 ±0.4 |
| **PREDICT-Gottlieb** | KNN | 33.5 ±0.3 | 7.4 ±1.7 |
| | Mean | 49.3 ±0.3 | **0.1 ±0.0** |
| | MissForest | - | - |
| | not-MIWAE | - | - |
| | Optimal Transport | 28.3 ±0.3 | 32.7 ±4.9 |
| | F3I | **28.2 ±0.2** | 110.6 ±28.2 |
| **TRANSCRIPT** | KNN | 8.7 ±0.2 | 26.8 ±2.4 |
| | Mean | **6.7 ±0.0** | **0.1 ±0.0** |
| | MissForest | - | - |
| | not-MIWAE | - | - |
| | Optimal Transport | 14.7 ±0.1 | 14.8 ±1.0 |
| | F3I | 8.6 ±0.2 | 129.3 ±52.9 |

### H.3 Imputation of mixed-type variables

A naive approach to extend readily F3I to mixed-type data–that is, with both categorical and continuous data. First, we replace Chebychev distance by Gower's distance (Gower, 1971) in the imputation model (see Algorithm 2). This allows F3I to compute the K nearest neighbors even when the point features categorical variables. Given two $F$-dimensional points points $\boldsymbol{x}$ and $\boldsymbol{y}$, Gower's distance between those two points is

$$d^{\text{Gower}}(\boldsymbol{x}, \boldsymbol{y}) := \frac{1}{F} \sum_{f \leq F} \text{dist}_f(x_f, y_f) \,,$$

where the distance function for a categorical variable $f$ is $\text{dist}_f(x, y) = \delta(x \neq y)$ and $\delta$ is the Kronecker symbol, whereas the distance function for a continuous variable $g$ is $\text{dist}_g(x, y) = |x - y|$.

Second, the kNN imputation approach for continuous variables–where missing values are estimated as weighted averages of the nearest neighbors–can naturally extend to categorical variables by adopting a consensus procedure. For each categorical value for a given variable in a given point, the weights of the K

nearest neighbors to the point featuring that value are summed. Then, the categorical value with highest weight sum is selected as the imputed value for the considered variable and point. Third, the kernel density estimator (which is currently a Gaussian kernel) might also be adapted to distributions on discrete sets, see for instance Rajagopalan & Lall (1995). However, the analysis of this mixed-type version of F3I is not straightforward, as the corresponding objective function G is no longer continuous due to the categorical imputation.

We implemented this extension of F3I, and applied it on the Titanic data set (Harrell, 2011) with the following categorical variables: 'Survived', 'Pclass', 'Sex'; continuous variables: 'Age', 'SibSp', 'Fare'. , using the Python packages `kdtree` (Kögl, 2013) and `gower` (Yan, 2019) to implement a KD Tree compatible with Gower's distance. Table 19 reports the average and standard deviations of Gower's distance (the smaller, the better) computed between the imputed data set and the complete Titanic data set across 10 iterations. Missing values were added with an MCAR mechanism with frequency $p^{\text{miss}} = 30\%$.

| Algorithm | Gower's distance |
|---|---|
| F3I | 0.30 ±0.15 |
| HyperImpute | 0.51 ±0.11 |

Table 19: Average and standard deviation of Gower's distance between the inputed data set and the complete Titanic dataset across 10 iterations.

On this data set, F3I achieves a significant improvement over HyperImpute, which was the top baseline in our experiments and can natively handle heterogeneous data.

# I  Experiments complementary to the main text

We report here tables of numerical results related to experiments described in the main text in Section 6.

## I.1  Imputation-only task

We study the imputation quality–without any downstream task. We resorted to the framework HyperImpute (Jarrett et al., 2022) to implement and run the benchmark for an imputation task across different performance metrics on the four standard data sets BreastCancer (Wolberg et al., 1993), Diabetes (from scikit-learn (Pedregosa et al., 2011)), HeartDisease (Janosi et al., 1989), Ionosphere (selva86, 2024), including only the top baselines based on Table 1.

First, we obtained more robust estimates of the performance, by averaging metrics over 100 runs instead of 10 runs as in the main text. We considered again the scenario MNAR in the framework HyperImpute to add missing values. We report in Table 20 the corresponding numerical results across 100 runs with different random seeds. Additionally, we also recorded the mean RMSE scores and runtime for 10 different seeds by running F3I and the top-3 baselines across the three different missingness mechanisms (corresponding to Assumptions B.2-B.4) on five different datasets, the results of which are reported in Table 21.

As written in the main text, those results on a larger set of runs confirm our observations mentioned in the main text, and show that F3I is competitive imputation-wise while being dramatically faster than baselines.

Table 20: Average and standard deviation values of imputation quality metrics (rounded to the closest second decimal place) and runtime across 100 different random seeds. HeartDisease has native missing values, which is why the Wasserstein distance cannot be computed. RMSE: root mean square error. MAE: mean average error. WD: Wasserstein distance. Runtime is in seconds. TDM failed on the Gottlieb data set. Bold type is the top performer, underline denotes the second best (and corresponding percentage of deterioration of performance across metrics compared to the top performer).

| Data set | | RMSE ↓ | MAE ↓ | WD ↓ | Runtime ↓ |
|---|---|---|---|---|---|
| BreastCancer | | | | | |
| F3I (ours) | | **0.08 ±0.03** | **0.03 ±0.01** | **0.07 ±0.02** | **0.18 ±0.06** |
| GAIN | | 0.27 ±0.03 | 0.10 ±0.02 | 0.24 ±0.05 | 45 ±27 |
| HyperImpute | (+213%) | 0.26 ±0.03 | 0.09 ±0.02 | 0.22 ±0.05 | 32 ±13 |
| MIRACLE | | 4.44 ±0.48 | 4.32 ±0.44 | 10.4 ±1.51 | 189 ±39 |
| NewImp | | 415 ±172 | 300 ±152 | 726 ±386 | 1,323 ±245 |
| HeartDisease | | | | | |
| F3I (ours) | | **0.14 ±0.05** | **0.07 ±0.03** | - | **0.13 ±0.04** |
| GAIN | | 0.30 ±0.07 | 0.18 ±0.05 | - | 15.89 ±4.47 |
| HyperImpute | (+79%) | 0.24 ±0.07 | 0.13 ±0.04 | - | 35 ±27 |
| MIRACLE | | 5.04 ±0.74 | 4.84 ±0.64 | - | 101 ±28 |
| NewImp | | 361 ±174 | 239 ±139 | - | 914 ±223 |
| Ionosphere | | | | | |
| F3I (ours) | (+11%) | 0.23 ±0.05 | 0.16 ±0.04 | 0.31 ±0.09 | **0.20 ±0.04** |
| GAIN | | 0.47 ±0.05 | 0.35 ±0.05 | 0.53 ±0.12 | 22 ±13 |
| HyperImpute | | **0.22 ±0.07** | **0.14 ±0.04** | **0.27 ±0.07** | 110 ±92 |
| MIRACLE | | 5.30 ±0.48 | 5.21 ±0.46 | 12.6 ±1.65 | 100 ±6 |
| NewImp | | 0.61 ±0.37 | 0.47 ±0.22 | 1.02 ±0.54 | 1,126 ±63 |

Table 21: Average runtime (in seconds) and RMSE score across 10 iterations across the three missingness mechanisms, for a missing rate of 30% when applicable (MCAR and MAR settings). RMSE values are rounded to the closest second decimal place.

| Data set | Algorithm | MNAR | | MCAR | | MAR | |
|---|---|---|---|---|---|---|---|
| | | RMSE | Time | RMSE | Time | RMSE | Time |
| BreastCancer | F3I (ours) | **0.10 ±0.03** | 0.14 | **0.05 ±0.02** | 0.20 | **0.09 ±0.02** | 0.20 |
| | GAIN | 0.27 ±0.01 | 34 | 0.13 ±0.03 | 34 | 0.34 ±0.01 | 43 |
| | HyperImpute | 0.26 ±0.02 | 7 | 0.10 ±0.04 | 8 | 0.33 ±0.02 | 10 |
| | kNN | 0.28 ±0.01 | **0.10** | 0.14 ±0.05 | **0.10** | 0.39 ±0.02 | **0.10** |
| | Remasker | 0.26 ±0.02 | 393 | 0.11 ±0.03 | 69 | 0.34 ±0.02 | 77 |
| Gottlieb | F3I (ours) | 0.04 ±0.03 | 2 | 0.04 ±0.00 | 3 | 0.04 ±0.00 | 3 |
| | GAIN | 0.04 ±0.00 | 103 | 0.04 ±0.00 | 64 | 0.04 ±0.00 | 119 |
| | HyperImpute | **0.02 ±0.00** | 44 | **0.03±0.00** | 12 | **0.03 ±0.00** | 10 |
| | kNN | 0.04 ±0.00 | **0.23** | 0.04 ±0.00 | **0.24** | 0.04 ±0.00 | **0.18** |
| | Remasker | 0.17 ±0.02 | 3,016 | 0.14 ±0.012 | 24.189 | 0.14 ±0.02 | 184 |
| HeartDisease | F3I (ours) | **0.14 ±0.03** | **0.10** | **0.10 ±0.01** | **0.10** | **0.18 ±0.03** | **0.10** |
| | GAIN | 0.30 ±0.04 | 34 | 0.24 ±0.04 | 8 | 0.36 ±0.05 | 9 |
| | HyperImpute | 0.24 ±0.04 | 17 | 0.13 ±0.03 | 8 | 0.29 ±0.06 | 7 |
| | kNN | 0.36 ±0.05 | **0.10** | 0.23 ±0.04 | **0.10** | 0.45 ±0.08 | **0.10** |
| | Remasker | 0.25 ±0.04 | 32 | 0.15 ±0.03 | 70 | 0.30 ±0.06 | 76 |
| Ionosphere | F3I (ours) | 0.21 ±0.04 | 0.19 | 0.22 ±0.03 | 0.37 | 0.25 ±0.07 | 0.67 |
| | GAIN | 0.48 ±0.03 | 50 | 0.48 ±0.03 | 45 | 0.47 ±0.05 | 42 |
| | HyperImpute | **0.20 ±0.04** | 99 | **0.23 ±0.04** | 22 | 0.25 ±0.08 | 36 |
| | kNN | 0.22 ±0.06 | **0.05** | 0.21 ±0.06 | **0.06** | **0.21 ±0.07** | **0.10** |
| | Remasker | 0.34 ±0.03 | 77 | 0.33 ±0.04 | 974 | 0.36 ±0.05 | 118 |

Table 22: Average of RMSE score over 10 seeds (rounded to the closest third decimal place) on the Ionosphere dataset for varying missing rates and missingness mechanisms. KNN is the kNN algorithm with distance-dependent weights.

| | $p^{\text{miss}}$ | **F3I (ours)** | KNN-Unif | HyperImpute | MIRACLE | GAIN | Remasker | GRAPE |
|---|---|---|---|---|---|---|---|---|
| M | 10% | 0.21 ±0.04 | 0.22 ±0.04 | **0.20 ±0.04** | 5.33 ±0.18 | 0.44 ±0.04 | 0.35 ±0.04 | 0.47 ±0.05 |
| C | 25% | 0.21 ±0.03 | 0.21 ±0.03 | **0.19 ±0.04** | 5.51 ±0.33 | 0.44 ±0.03 | 0.36 ±0.03 | 0.46 ±0.03 |
| A | 50% | **0.22 ±0.04** | 0.23 ±0.04 | **0.22 ±0.04** | 5.08 ±0.53 | 0.46 ±0.02 | 0.35 ±0.03 | 0.46 ±0.02 |
| R | 75% | **0.28 ±0.03** | 0.30 ±0.03 | **0.28 ±0.03** | 5.33 ±0.72 | 0.48 ±0.03 | 0.34 ±0.02 | 0.48 ±0.01 |
| | 90% | **0.35 ±0.02** | 0.41 ±0.02 | 0.36 ±0.03 | 4.89 ±0.75 | 0.48 ±0.027 | 0.36 ±0.04 | 0.44 ±0.02 |
| M | 10% | 0.20 ±0.06 | 0.22 ±0.04 | **0.17 ±0.05** | 5.69 ±0.16 | 0.40 ±0.06 | 0.33 ±0.05 | 0.45 ±0.06 |
| A | 25% | 0.19 ±0.04 | 0.26 ±0.03 | **0.18 ±0.04** | 5.46 ±0.19 | 0.47 ±0.05 | 0.31 ±0.03 | 0.43 ±0.03 |
| R | 50% | 0.23 ±0.03 | 0.29 ±0.04 | **0.22 ±0.03** | 5.35 ±0.24 | 0.46 ±0.04 | 0.32 ±0.04 | 0.43 ±0.04 |
| | 75% | **0.28 ±0.02** | 0.32 ±0.03 | **0.28 ±0.05** | 5.22 ±0.33 | 0.47 ±0.04 | 0.34 ±0.04 | 0.44 ±0.03 |
| | 90% | 0.37 ±0.03 | 0.38 ±0.05 | **0.35 ±0.03** | 5.26 ±0.41 | 0.45 ±0.03 | 0.37 ±0.03 | 0.44 ±0.02 |
| M | 10% | **0.19 ±0.05** | 0.25 ±0.02 | **0.19 ±0.05** | 5.48 ±0.17 | 0.42 ±0.04 | 0.34 ±0.05 | 0.44 ±0.04 |
| N | 25% | 0.20 ±0.04 | 0.20 ±0.04 | **0.19 ±0.04** | 5.30 ±0.24 | 0.45 ±0.04 | 0.32 ±0.04 | 0.45 ±0.03 |
| A | 50% | **0.23 ±0.03** | 0.24 ±0.03 | 0.22 ±0.03 | 5.11 ±0.46 | 0.46 ±0.02 | 0.32 ±0.03 | 0.45 ±0.02 |
| R | 75% | **0.28 ±0.02** | 0.31 ±0.03 | **0.28 ±0.05** | 4.94 ±0.67 | 0.48 ±0.03 | 0.33 ±0.03 | 0.44 ±0.01 |
| | 90% | 0.37 ±0.03 | 0.37 ±0.02 | **0.38 ±0.04** | 4.92 ±0.75 | 0.47 ±0.03 | 0.39 ±0.03 | 0.49 ±0.01 |

Table 23: Average of RMSE score over 10 seeds (rounded to the closest third decimal place) on the Breast Cancer dataset for varying missing rates and missingness mechanisms. KNN is the kNN algorithm with distance-dependent weights.

| | $p^{\text{miss}}$ | **F3I (ours)** | KNN-Unif | HyperImpute | MIRACLE | GAIN | Remasker | GRAPE |
|---|---|---|---|---|---|---|---|---|
| M | 10% | **0.03 ±0.01** | 0.15 ±0.03 | 0.05 ±0.03 | 4.19 ±0.11 | 0.08 ±0.04 | 0.06 ±0.03 | 0.28 ±0.04 |
| C | 25% | **0.04 ±0.01** | 0.21 ±0.03 | 0.09 ±0.04 | 4.10 ±0.14 | 0.12 ±0.03 | 0.11 ±0.04 | 0.28 ±0.02 |
| A | 50% | **0.07 ±0.02** | 0.21 ±0.02 | 0.14 ±0.03 | 4.01 ±0.23 | 0.18 ±0.02 | 0.15 ±0.03 | 0.31 ±0.01 |
| R | 75% | **0.08 ±0.01** | 0.23 ±0.01 | 0.20 ±0.01 | 3.89 ±0.35 | 0.21 ±0.01 | 0.20 ±0.01 | 0.31 ±0.01 |
| | 90% | **0.13 ±0.07** | 0.26 ±0.03 | 0.21 ±0.01 | 3.84 ±0.39 | 0.23 ±0.03 | 0.21 ±0.01 | 0.31 ±0.01 |
| M | 10% | **0.13 ±0.02** | 0.58 ±0.02 | 0.48 ±0.06 | 5.48 ±0.32 | 0.49 ±0.06 | 0.48 ±0.06 | 0.65 ±0.07 |
| A | 25% | **0.11 ±0.02** | 0.40 ±0.03 | 0.38 ±0.02 | 4.80 ±0.21 | 0.38 ±0.02 | 0.38 ±0.02 | 0.47 ±0.03 |
| R | 50% | **0.10 ±0.02** | 0.31 ±0.01 | 0.29 ±0.01 | 4.40 ±0.23 | 0.29 ±0.01 | 0.29 ±0.01 | 0.37 ±0.02 |
| | 75% | **0.10 ±0.02** | 0.26 ±0.00 | 0.25 ±0.01 | 4.23 ±0.31 | 0.25 ±0.01 | 0.25 ±0.01 | 0.33 ±0.01 |
| | 90% | **0.10 ±0.01** | 0.24 ±0.00 | 0.24 ±0.01 | 4.28 ±0.40 | 0.24 ±0.01 | 0.24 ±0.01 | 0.32 ±0.01 |
| M | 10% | **0.10 ±0.02** | 0.43 ±0.03 | 0.34 ±0.04 | 4.83 ±0.21 | 0.35 ±0.04 | 0.34 ±0.04 | 0.50 ±0.04 |
| N | 25% | **0.08 ±0.02** | 0.33 ±0.02 | 0.27 ±0.02 | 4.37 ±0.21 | 0.29 ±0.02 | 0.28 ±0.02 | 0.38 ±0.02 |
| A | 50% | **0.08 ±0.01** | 0.27 ±0.01 | 0.23 ±0.01 | 4.47 ±0.43 | 0.24 ±0.01 | 0.23 ±0.01 | 0.34 ±0.02 |
| R | 75% | **0.08 ±0.01** | 0.24 ±0.01 | 0.22 ±0.01 | 3.94 ±0.29 | 0.23 ±0.01 | 0.22 ±0.01 | 0.30 ±0.01 |
| | 90% | **0.10 ±0.24** | 0.27 ±0.02 | 0.23 ±0.01 | 3.83 ±0.44 | 0.24 ±0.01 | 0.23 ±0.01 | 0.32 ±0.01 |

## I.2 Joint imputation-binary classification task

We also looked at the trend in performance in PCGradF3I on the BreastCancer and Ionosphere data sets for increasing values of $\beta$, related to the importance of the classification task, in Table 24. As expected, increasing values of $\beta$ improve the performance of the classifier, reaching a plateau in the average AUC value.

Table 24: Average and standard deviation Area Under the Curve (AUC) values on a held-out testing set across runs on the joint imputation-classification task (MCAR scenario, $p^{\mathrm{miss}} = 0.5$). $\beta$ is the weight of the classification task in PCGradF3I. Bold type is the top performer, underline denotes the second best. Results for the BreastCancer data set are computed over 100 runs, over 60 runs for the Ionosphere data set.

| Data set / $\beta$ | 0.14 | 0.25 | 0.50 | 0.75 |
|---|---|---|---|---|
| PCGradF3I (Ionosphere) | 0.778 ±0.174 | **0.785 ±0.174** | **0.785 ±0.176** | 0.783 ±0.174 |
| PCGradF3I (BreastCancer) | 0.699 ±0.142 | **0.700 ±0.143** | **0.700 ±0.142** | **0.700 ±0.142** |

