# OpenReview forum: "Handling Missing Data in Downstream Tasks With Distribution-Preserving Guarantees"
_TMLR — Withdrawn by Authors_

### Review · Reviewer_bozn · 2025-11-20

**Summary Of Contributions:**

Contributions:
The authors propose a novel method for missing values imputation.
Method is essentially a modification of the KNN algorithm, in which the weights for nearest neighbours are learned.
The authors provide theoretical guarantees on the method's performance in terms of upper bounds for the MSE and regret for both the imputation task and the joint imputation-downstream task.

Strengths:
1) The main text is well-written and easy to follow.
2) The proposed method is theoretically supported.
3) The theoretical bounds are validated experimentally.
4) Empirical evaluation shows that the method stably achieves SOTA or near-SOTA performance, while being much faster.
5) The method is thoroughly ablated.

Weaknesses:
1) Presentation needs to be improved. Most of the tables and figures are badly formatted and hard to read. The authors should carefully go through each figure and table to ensure consistency in captions, labels, axis names, and titles. Layout, object sizes, axis scales, font sizes, and the overall visualization format should be refined.
   1) Section 2.2: The "Notation" subsection should be moved to section 3.
   2) Section 3.2: I think the definition of G should be formatted as an equation.
   3) Section 3.2: The sentense, "A practical value of h can be computed by explicitly finding the smallest positive root of a specific cubic equation, for instance, by using Cardano’s method (Cardano et al., 1968).", appears out of context. A reference to Appendix C should be provided instead.
   4) Section 3.3: Early stopping is introduced in the F3I algorithm; there should be at least one sentence in the main text explaining it.
   5) Table 1: Table overlaps with the page number.
   6) Table 2, Table 3: I believe "iterations" should be changed to "seeds".
   7) Table 3: The table should have the same form as the Table 2 (according to text). Furthermore, it is unclear why only one baseline is used. If including the other baselines is computationally expensive, it would be better to remove this table from the main text.
   8) Table 1, Table 2, Table 3: The tables are located far from their references in the text (actually in the middle of the section 8), making them difficult to find. If template constraints are an issue, the authors could: i) change format of the table; ii) move it's part to Appendix; iii) use boxplots/barplors as in HyperImpute/Miracle/NewImp papers.
   10) Figure 4: From them plots I can not see that "The blue points are always below the red lines".
   11) Figure 4, Figure 6: The labels are inconsistent (one label is a text and another is an expression).
   12) Figure 1, Figure 3, Figure 4: It is better to add titles with missingness type.
   13) Figure 5: Figure 5 is fully unreadable: i) Subplots are overlapped; ii) labels intersect with subplots borders; iii) labels fontsizes are too small; iv) the title positioned outside the plot (I think it should be centered); v) boxplots are too small to distinguish; vi) the y-axis label is unnecessarily abbreviated; vii) the missingness probability is in the title, whereas it was in the caption for previous figures.
   14) Table 7: i) Last column is not aligned; ii) column names are inconsistent (some of them are expressions, while others are text).
   15) Figure 7, Figure 8, Figure 11, Figure 12, Figure 13, Figure 14, Figure 15, Figure 16, Figure 17, Figure 18, Figure 19: the issues of Figure 5 are applicable.
   16) Table 10: The columns are misaligned, the table width does not match text width.
   17) Algorithm 1: the comments should be aligned;
   18) Figure 20, Figure 21, Figure 22, Figure 23, Figure 24: The most of the plots have one method with extreme values, making boxes for other methods unreadable. Presenting these results in a table is recommended.
2) The empirical evaluation is confusing. The benchmarking does not appear fully consistent with previous works.
   1) The authors state they follow the HyperImpute evaluation framework but use only 5 datasets, whereas HyperImpute used 12 (with an overlap of only 2). For a consistent comparison, it is recommended to use the same datasets as the HyperImpute paper.
   2) Since F3I is an improvement over a KNN algorithm, the main experiments should include KNN baselines (with both uniform and distance-dependent weights). For completeness, KNN baselines should be added to all applicable experiments.

**Additional Comments:**

Questions:
1) Equality (1) holds only when $\frac{D_0((x^0)_i(\alpha))}{D_0((x^0)_i)}$ is close to 1.
Are there any explicit or implicit constraints imposed by the F3I algorithm to keep this ratio close to 1?
2) Have authors observed an overfitting of the F3I without regularization?
3) What is the purpose of keeping the regularization term in the early stopping criterion?
4) Have authors observed changes in the KD-tree across iterations? For example, how many of the K closest neighbours initially move outside the K neighbours set on average in subsequent iterations?
5) How does the MSE loss behave through iterations on the synthetic and real world data?
6) What missing rate was used in the first experiment in section 6.1 (Table 1)?
7) While NewImp authors do not report RMSE, in their experiments MAE for all datasets/missing rates is less by several orders of magnitude compared to the values in the Table 1.
HyperImpute's performance is also lower than stated in the original paper: for example, the original paper reports WD<0.35 on diabet dataset for all missing rates (0.1, 0.3, 0.5, 0.7) (Figure 8), but in the Table 1 (which I understand uses a missing rate of 0.3?), the authors obtain WD≈0.54 on the same dataset. Why does the baseline performance not match the previously reported results? Did the authors use the original implementations? Is it possible to collect the previously reported metrics for all used baselines on a fixed dataset with a fixed missing mechanism and missing rate?
7) Do I correctly understand, that t, the Cumulative regret on G, and the upper bound in the Table 8 all are positive numbers?
8) Why, instead of optimizing $-G(\alpha^t, X^{t-1})$, do the authors propose to minimize $-(\alpha^t, \nabla_{\alpha}G(\alpha^t, X^{t-1}))$ (Section 3.3)?
9) In my opinion, F3I has two additional limitations, which should be added to the corresponding section: i) The KNN weights are not sample-specific; ii) F3I relies on KDE and KNN assumptions and is hence prone to the curse of dimensionality. Do the authors agree with that?

**Audience:**

Yes

**Audience Explanation:**

The proposed F3I method demostrates SOTA performance on the task of imputing missing values while being much faster than previous methods; hence, it would be interesting to the ML community.

**Broader Impact Concerns:**

-

**Claims And Evidence:**

Yes

**Claims Explanation:**

Despite my concerns about the benchmarking, the authors rigorously evaluate their proposed method across a variety of scenarios. These include diverse synthetic and real-world datasets with different missingness mechanisms and rates. The paper provides a clear motivation for the method's design and supports it with a detailed theoretical analysis. Furthermore, non-trivial statements are consistently supported by references to the relevant literature.

**Requested Changes:**

My concerns about the paper are described in the "Weaknesses" section.
Requested changes:
1) Fix the presentation issues.
2) Conduct additional benchmarking.

---

> ### Author Response · Authors · 2025-12-09
> **Rebuttal 1/4**
>
> We thank the reviewer for their thorough review, and their positive comments about our theoretical and empirical results. We updated a partial revision of the manuscript, and will upload the final version in the upcoming days. In this final version, we will address issues 12 and 14, report the results for the running experiments in the table below and in Tables 2-3 of the manuscript, and add our observations on the k-d tree.
>
> **1. ``Presentation needs to be improved [...].''**
>
> We included all the listed comments but issues 12 and 14 (that will be addressed in the next revision). Thank you!
>
> **2. ``The authors state they follow the HyperImpute evaluation framework but use only 5 datasets, whereas HyperImpute used 12 (with an overlap of only 2). For a consistent comparison, it is recommended to use the same datasets as the HyperImpute paper.''**
>
> The main reason why we did not test all the data sets present in the HyperImpute paper is because some of them have categorical/discrete data as features: e.g., Airfoil Self-Noise Dataset, Blood Transfusion Service Center Dataset, California Housing Dataset, Concrete Compressive Strength Dataset, Iris Dataset, and Letter Recognition Dataset. Our contribution F3I (in its original implementation) is not suitable for such data, as highlighted in the limitations. However, in the revision of the paper, we will add results on the Libras Movement Dataset, the Spambase Dataset, and the Wine Quality (red and white) Datasets. The numerical results we obtained so far are in the table below.
>
> | Data set     | Alg.                | RMSE $\downarrow$           | MAE $\downarrow$            | WD $\downarrow$             | Runtime $\downarrow$        |
> |--------------|---------------------|-----------------------------|-----------------------------|-----------------------------|-----------------------------|
> | Spambase     |**F3I (ours)**|*0.06 $\pm$0.01* |*0.02 $\pm$0.00* |*0.02 $\pm$0.00* | 26.04 $\pm$7.80             |
> |              | GAIN                |*0.06 $\pm$0.01* |*0.02 $\pm$0.00* | 0.04 $\pm$0.01              | 5.80 $\pm$1.45              |
> |              | GRAPE               | 0.07 $\pm$0.01              | 0.03 $\pm$0.00              | 0.04 $\pm$0.01              | 4,090 $\pm$691              |
> |              | KNN                 |**0.05 $\pm$0.00**    |**0.01 $\pm$0.00**    |*0.01 $\pm$0.00*    |*2.91 $\pm$0.66* |
> |              | KNN-Unif            |**0.05 $\pm$0.00**   |**0.01 $\pm$0.00**    |*0.02 $\pm$0.00* |**1.99 $\pm$0.25**    |
> |              | HyperImpute         |**0.05 $\pm$0.00**   |*0.02 $\pm$0.00* | 0.03 $\pm$0.01              | 22.01 $\pm$4.51             |
> |              | MIRACLE             |**0.05 $\pm$0.00**   | 0.03 $\pm$0.00              | 0.05 $\pm$0.00              | 51.76 $\pm$4.37             |
> |              | NewImp              | Running                     | Running                     | Running                     | Running                     |
> |              | Remasker            | Running                     | Running                     | Running                     | Running                     |
> |              | TDM                 | 0.22 $\pm$0.01              | 0.17 $\pm$0.01              | 0.38 $\pm$0.02              | 246.16 $\pm$76.29           |
> | Wine (White) |**F3I (ours)**| 0.11 $\pm$0.01              | 0.07 $\pm$0.01              | 0.06 $\pm$0.01              | 19.20 $\pm$4.51             |
> |              | GAIN                | 0.10 $\pm$0.01              | 0.07 $\pm$0.00              | 0.12 $\pm$0.01              | 5.84 $\pm$1.30              |
> |              | GRAPE               | 0.19 $\pm$0.01              | 0.15 $\pm$0.01              | 0.24 $\pm$0.02              | 1,047 $\pm$145              |
> |              | KNN                 |**0.08 $\pm$0.01**    |**0.05 $\pm$0.00**    |*0.05 $\pm$0.00* |*2.12 $\pm$0.34* |
> |              | KNN-Unif            |*0.09 $\pm$0.00*|*0.06 $\pm$0.00* | 0.06 $\pm$0.00              |**1.95 $\pm$0.34**   |
> |              | HyperImpute         |**0.08 $\pm$0.00**    |**0.05 $\pm$0.00**    |**0.04 $\pm$0.00**   | 21.53 $\pm$4.82             |
> |              | MIRACLE             |*0.09 $\pm$0.00*|*0.06 $\pm$0.00* | 0.07 $\pm$0.00              | 28.98 $\pm$6.82             |
> |              | NewImp              | Running                     | Running                     | Running                     | Running                     |
> |              | Remasker            | Running                     | Running                     | Running                     | Running                     |
> |              | TDM                 | 0.16 $\pm$0.01              | 0.13 $\pm$0.01              | 0.12 $\pm$0.02              | 170.92 $\pm$29.28           |

---

> ### Author Response · Authors · 2025-12-09
> **Rebuttal 2/4**
>
> | Data set     | Alg.                | RMSE $\downarrow$           | MAE $\downarrow$            | WD $\downarrow$             | Runtime $\downarrow$        |
> |--------------|---------------------|-----------------------------|-----------------------------|-----------------------------|-----------------------------|
> | Wine (Red)   |**F3I (ours)**| 0.13 $\pm$0.01              | 0.09 $\pm$0.01              | 0.09 $\pm$0.00              |*2.29 $\pm$0.43* |
> |              | GAIN                | 0.13 $\pm$0.01              | 0.09 $\pm$0.01              | 0.14 $\pm$0.01              | 5.90 $\pm$2.45              |
> |              | GRAPE               | 0.24 $\pm$0.01              | 0.19 $\pm$0.01              | 0.28 $\pm$0.02              | 431 $\pm$15                 |
> |              | KNN                 |**0.10 $\pm$0.00**    |**0.06 $\pm$0.00**    |*0.07 $\pm$0.01* |**0.17 $\pm$0.03**    |
> |              | KNN-Unif            |*0.11 $\pm$0.00* |*0.07 $\pm$0.00* | 0.09 $\pm$0.01              |**0.17 $\pm$0.02**    |
> |              | HyperImpute         |**0.10 $\pm$0.00**    |*0.07 $\pm$0.00* |**0.05 $\pm$0.00**    | 20.91 $\pm$5.04             |
> |              | MIRACLE             | 0.21 $\pm$0.01              | 0.16 $\pm$0.01              | 0.16 $\pm$0.02              | 11.42 $\pm$0.94             |
> |              | NewImp              | Running                     | Running                     | Running                     | Running                     |
> |              | Remasker            | Running                     | Running                     | Running                     | Running                     |
> |              | TDM                 | 0.19 $\pm$0.01              | 0.14 $\pm$0.01              | 0.10 $\pm$0.01              | 157.67 $\pm$15.86           |
> | Libras       |**F3I (ours)**| 0.08 $\pm$ 0.00             | 0.07 $\pm$ 0.00             | 0.06 $\pm$ 0.00             |*0.51 $\pm$ 0.15*|
> |              | GAIN                |*0.05 $\pm$ 0.00*|*0.05 $\pm$ 0.00*|*0.03 $\pm$ 0.00*| 36.06 $\pm$ 6.72            |
> |              | GRAPE               | Running                     | Running                     | Running                     | Running                     |
> |              | KNN                 | 0.09 $\pm$ 0.00             | 0.08 $\pm$ 0.01             | 0.07 $\pm$ 0.00             |**0.02 $\pm$ 0.00**   |
> |              | KNN-Unif            | 0.09 $\pm$ 0.00             | 0.08 $\pm$ 0.00             | 0.07 $\pm$ 0.00             |**0.02 $\pm$ 0.00**   |
> |              | HyperImpute         |**0.02 $\pm$ 0.00**   |**0.02 $\pm$ 0.00**   |**0.01 $\pm$ 0.00**  | 11.06 $\pm$ 1.28            |
> |              | MIRACLE             | 4.11 $\pm$ 0.27             | 9.80 $\pm$ 0.76             | 4.03 $\pm$ 0.26             | 22.01 $\pm$ 0.50            |
> |              | NewImp              | Running                     | Running                     | Running                     | Running                     |
> |              | Remasker            | 0.19 $\pm$ 0.01             | 0.24 $\pm$ 0.02             | 0.15 $\pm\pm$  0.01         | 2,123 $\pm$ 271             |
> |              | TDM                 | 0.31 $\pm$ 0.01             | 0.14 $\pm$ 0.01             | 0.25 $\pm$ 0.00             | 665.01 $\pm$ 106.04         |
>
> **3. ``Since F3I is an improvement over a KNN algorithm, the main experiments should include KNN baselines (with both uniform and distance-dependent weights). For completeness, KNN baselines should be added to all applicable experiments.''**
>
> We added and will add results related to KNN to Tables 1-3 in the main text.
>
> **4. ``[The first equality of the paper] holds only when [the ratio between densities] is close to 1. Are there any explicit or implicit constraints imposed by the F3I algorithm to keep this ratio close to 1?''**
>
> The approximation is based on the limited development of the $\log$ function around $0$, and allowed us to construct a differentiable objective function. In order to reduce the inaccuracy of the approximation, we added the early stopping criterion in the F3I algorithm so that the function G has nonnegative values, which means that the product of the ratios is greater than $\exp(N\eta \|\alpha\|^2_2) < \exp(4NK \|\alpha\|^2_2)$. We acknowledge that this condition does not ensure that each ratio is close to 1, but it ensures that the ratios across samples are not too low on average.

---

> ### Author Response · Authors · 2025-12-09
> **Rebuttal 3/4**
>
> **5. ``Have authors observed an overfitting of the F3I without regularization?''**
>
> We did not perform explicit experiments to investigate overfitting. The inclusion of the regularization term, governed by the parameter $\eta$, was solely for theoretical investigation, aimed at analyzing the precise range of values of $\eta$ that would maintain our established theoretical guarantees. Proposition C.2 establishes that our core theoretical guarantees are maintained even when the regularization parameter $\eta$ is $0$ (in which case there is no regularization), provided the data adheres to the assumed distributions.
>
> **6. ``What is the purpose of keeping the regularization term in the early stopping criterion?''**
>
> The regularization parameter is kept in the early stopping criterion to avoid early stopping due to numeric instability (see our response to the approximation made in constructing objective function $G$). In particular, it ensures that early stopping is not triggered when $G$ attains a small negative value due to numerical instability.
>
> **7. ``Have authors observed changes in the KD-tree across iterations? For example, how many of the K closest neighbours initially move outside the K neighbours set on average in subsequent iterations?''**
>
> We will perform this experiment in the upcoming days, and comment again once the experiment concludes.
>
> **8. ``How does the MSE loss behave through iterations on the synthetic and real world data?''**
>
> We monitored the value of the MSE over 4 iterations (not seeds) on the BreastCancer, Diabetes, Gottlieb, TRANSCRIPT and synthetic (100 samples, 100 features) data sets, with missing values added with the MCAR mechanism described in the manuscript with $p^\text{miss}=30\%$. We show below the absolute values of MSE. When there is no value before the fourth iteration, it means that F3I exited through the early stopping condition. Overall, the MSE decreases across iterations, which is consistent with our assumption that maximizing the objective function $G$ leads to a better imputation.
>
> | Data set / Iteration    | 1         | 2         | 3         | 4         |
> |-------------------------|-----------|-----------|-----------|-----------|
> | TRANSCRIPT              | 3.497     | 3.250     | 3.247     | -         |
> | Synthetic (100 samples) | 2.511     | 2.423     | -         | -         |
> | Diabetes                | 1.555     | 1.501     | 1.491     | -         |
> | Gottlieb                | 4.069     | 3.993     | -         | -         |
> | BreastCancer            | 8,359.276 | 8,329.815 | 8,322.047 | 8,321.185 |
>
> **9. ``What missing rate was used in the first experiment in section 6.1 (Table 1)?''**
>
> The missing rate used in Table 1 was 30\%. We added it to Section 6.1.
>
> **10. ``While NewImp authors do not report RMSE, in their experiments MAE for all datasets/missing rates is less by several orders of magnitude compared to the values in the Table 1. HyperImpute's performance is also lower than stated in the original paper: for example, the original paper reports WD$<$0.35 on Diabetes dataset for all missing rates (0.1, 0.3, 0.5, 0.7) (Figure 8), but in the Table 1 (which I understand uses a missing rate of 0.3?), the authors obtain WD$\approx$0.54 on the same dataset. Why does the baseline performance not match the previously reported results? Did the authors use the original implementations? Is it possible to collect the previously reported metrics for all used baselines on a fixed dataset with a fixed missing mechanism and missing rate?''**
>
> We unfortunately suspect that there is a reproducibility issue in both codes for NewImp and HyperImpute: the random seed used is not provided in the papers, leading to potentially distinct training/testing splits. In our experiments, we have used default/recommended parameter settings for NewImp and HyperImpute. Moreover, we ensured to run all runs with different random seeds and all algorithms for a given data set on the same machine to cope with this issue. Despite this precaution, we observed that, from one computer to the other, the results might (slightly) change in absolute values--however, the relative performance across data sets and algorithms is mostly preserved. As shown in our code, we used the implementation in HyperImpute for the testing framework and imputation plugin, and we wrote the plugin for NewImp based on the original implementation provided by the authors. Collecting all metrics from the original papers might lead to a bias, as the random seeds might not be the same, and might yield incomparable metrics.

---

> ### Author Response · Authors · 2025-12-09
> **Rebuttal 4/4**
>
> **11. ``Do I correctly understand, that t, the Cumulative regret on G, and the upper bound in the Table 8 all are positive numbers?''**
>
> Yes, they are.
>
> **12. ``Why, instead of optimizing $-G(\alpha^t,X^{t-1})$, do the authors propose to minimize $-\langle \alpha^t, \nabla_\alpha G(\alpha^t, X^{t-1})\rangle$ (Section 3.3)?''**
>
> As mentioned in Section 3.4, directly minimizing $-G(\alpha^t,X^{t-1})$ at each iteration might be a time-consuming step, even if the underlying problem is convex. Then, we aim at minimizing the cumulative regret on G defined at the top of page 6 instead, and leverage this to resort to online learners, which are much faster.
>
> **13. ``In my opinion, F3I has two additional limitations, which should be added to the corresponding section: i) The KNN weights are not sample-specific; ii) F3I relies on KDE and KNN assumptions and is hence prone to the curse of dimensionality. Do the authors agree with that?''**
>
> We agree, and added this to the paragraph on limitations and to the conclusion of the manuscript.

---

> ### Author Response · Authors · 2025-12-18
> **Update (1/3)**
>
> We updated the latest version of the revision. In particular, we addressed issues 12 and 14, and reported all missing results and the observations of the k-d tree below.
>
> **2. ``The authors state they follow the HyperImpute evaluation framework but use only 5 datasets, whereas HyperImpute used 12 (with an overlap of only 2). For a consistent comparison, it is recommended to use the same datasets as the HyperImpute paper.''**
>
> | Data set     | Alg.                | RMSE $\downarrow$           | MAE $\downarrow$            | WD $\downarrow$             | Runtime $\downarrow$        |
> |--------------|---------------------|-----------------------------|-----------------------------|-----------------------------|-----------------------------|
> | Spambase     |**F3I (ours)**|*0.06 $\pm$0.01* |*0.02 $\pm$0.00* |*0.02 $\pm$0.00* | 26.04 $\pm$7.80             |
> |              | GAIN                |*0.06 $\pm$0.01* |*0.02 $\pm$0.00* | 0.04 $\pm$0.01              | 5.80 $\pm$1.45              |
> |              | GRAPE               | 0.07 $\pm$0.01              | 0.03 $\pm$0.00              | 0.04 $\pm$0.01              | 4,090 $\pm$691              |
> |              | KNN                 |**0.05 $\pm$0.00**    |**0.01 $\pm$0.00**    |*0.01 $\pm$0.00*    |*2.91 $\pm$0.66* |
> |              | KNN-Unif            |**0.05 $\pm$0.00**   |**0.01 $\pm$0.00**    |*0.02 $\pm$0.00* |**1.99 $\pm$0.25**    |
> |              | HyperImpute         |**0.05 $\pm$0.00**   |*0.02 $\pm$0.00* | 0.03 $\pm$0.01              | 22.01 $\pm$4.51             |
> |              | MIRACLE             |**0.05 $\pm$0.00**   | 0.03 $\pm$0.00              | 0.05 $\pm$0.00              | 51.76 $\pm$4.37             |
> |              | NewImp              | 0.17 $\pm$0.07              | 0.11 $\pm$0.03              | 0.27 $\pm$0.07              | 54,190 $\pm$9,704           |
> |              | Remasker            | **0.05 $\pm$0.00**    | *0.02 $\pm$0.00*  | 0.03 $\pm$0.00              | 5,507 $\pm$676              |
> |              | TDM                 | 0.22 $\pm$0.01              | 0.17 $\pm$0.01              | 0.38 $\pm$0.02              | 246.16 $\pm$76.29           |
> | Wine (White) |**F3I (ours)**| 0.11 $\pm$0.01              | 0.07 $\pm$0.01              | 0.06 $\pm$0.01              | 19.20 $\pm$4.51             |
> |              | GAIN                | 0.10 $\pm$0.01              | 0.07 $\pm$0.00              | 0.12 $\pm$0.01              | 5.84 $\pm$1.30              |
> |              | GRAPE               | 0.19 $\pm$0.01              | 0.15 $\pm$0.01              | 0.24 $\pm$0.02              | 1,047 $\pm$145              |
> |              | KNN                 |**0.08 $\pm$0.01**    |**0.05 $\pm$0.00**    |*0.05 $\pm$0.00* |*2.12 $\pm$0.34* |
> |              | KNN-Unif            |*0.09 $\pm$0.00*|*0.06 $\pm$0.00* | 0.06 $\pm$0.00              |**1.95 $\pm$0.34**   |
> |              | HyperImpute         |**0.08 $\pm$0.00**    |**0.05 $\pm$0.00**    |**0.04 $\pm$0.00**   | 21.53 $\pm$4.82             |
> |              | MIRACLE             |*0.09 $\pm$0.00*|*0.06 $\pm$0.00* | 0.07 $\pm$0.00              | 28.98 $\pm$6.82             |
> |              | NewImp              | 0.38 $\pm$0.03              | 0.31 $\pm$0.02              | 0.75 $\pm$0.06              | 32,375 $\pm$,3509           |
> |              | Remasker            | **0.08 $\pm$0.00**     | **0.05 $\pm$0.00**    | 0.06 $\pm$0.00              | 1,340 $\pm$224
> |              | TDM                 | 0.16 $\pm$0.01              | 0.13 $\pm$0.01              | 0.12 $\pm$0.02              | 170.92 $\pm$29.28           |

---

> ### Author Response · Authors · 2025-12-18
> **Update (2/3)**
>
> | Data set     | Alg.                | RMSE $\downarrow$           | MAE $\downarrow$            | WD $\downarrow$             | Runtime $\downarrow$        |
> |--------------|---------------------|-----------------------------|-----------------------------|-----------------------------|-----------------------------|
> | Wine (Red)   |**F3I (ours)**| 0.13 $\pm$0.01              | 0.09 $\pm$0.01              | 0.09 $\pm$0.00              |*2.29 $\pm$0.43* |
> |              | GAIN                | 0.13 $\pm$0.01              | 0.09 $\pm$0.01              | 0.14 $\pm$0.01              | 5.90 $\pm$2.45              |
> |              | GRAPE               | 0.24 $\pm$0.01              | 0.19 $\pm$0.01              | 0.28 $\pm$0.02              | 431 $\pm$15                 |
> |              | KNN                 |**0.10 $\pm$0.00**    |**0.06 $\pm$0.00**    |*0.07 $\pm$0.01* |**0.17 $\pm$0.03**    |
> |              | KNN-Unif            |*0.11 $\pm$0.00* |*0.07 $\pm$0.00* | 0.09 $\pm$0.01              |**0.17 $\pm$0.02**    |
> |              | HyperImpute         |**0.10 $\pm$0.00**    |*0.07 $\pm$0.00* |**0.05 $\pm$0.00**    | 20.91 $\pm$5.04             |
> |              | MIRACLE             | 0.21 $\pm$0.01              | 0.16 $\pm$0.01              | 0.16 $\pm$0.02              | 11.42 $\pm$0.94             |
> |              | NewImp              | 0.50 $\pm$0.03              | 0.42 $\pm$0.03              | 1.01 $\pm$0.07              | 5,846 $\pm$377              |
> |              | Remasker            |**0.10 $\pm$0.00**    | *0.07 $\pm$0.00*  | 0.08 $\pm$0.01              | 639.05 $\pm$146.41
> |              | TDM                 | 0.19 $\pm$0.01              | 0.14 $\pm$0.01              | 0.10 $\pm$0.01              | 157.67 $\pm$15.86           |
> | Libras       |**F3I (ours)**| 0.08 $\pm$ 0.00             | 0.07 $\pm$ 0.00             | 0.06 $\pm$ 0.00             |*0.51 $\pm$ 0.15*|
> |              | GAIN                |*0.05 $\pm$ 0.00*|*0.05 $\pm$ 0.00*|*0.03 $\pm$ 0.00*| 36.06 $\pm$ 6.72            |
> |              | GRAPE               | 0.29 $\pm$ 0.01             | 0.14 $\pm$ 0.01             | 0.24 $\pm$ 0.01             | 7,039 $\pm$ 795
> |              | KNN                 | 0.09 $\pm$ 0.00             | 0.08 $\pm$ 0.01             | 0.07 $\pm$ 0.00             |**0.02 $\pm$ 0.00**   |
> |              | KNN-Unif            | 0.09 $\pm$ 0.00             | 0.08 $\pm$ 0.00             | 0.07 $\pm$ 0.00             |**0.02 $\pm$ 0.00**   |
> |              | HyperImpute         |**0.02 $\pm$ 0.00**   |**0.02 $\pm$ 0.00**   |**0.01 $\pm$ 0.00**  | 11.06 $\pm$ 1.28            |
> |              | MIRACLE             | 4.11 $\pm$ 0.27             | 9.80 $\pm$ 0.76             | 4.03 $\pm$ 0.26             | 22.01 $\pm$ 0.50            |
> |              | NewImp              | 0.31 $\pm$ 0.09             | 0.52 $\pm$ 0.12             | 0.23 $\pm$ 0.05             | 579.65 $\pm$ 67.32
> |              | Remasker            | 0.19 $\pm$ 0.01             | 0.24 $\pm$ 0.02             | 0.15 $\pm\pm$  0.01         | 2,123 $\pm$ 271             |
> |              | TDM                 | 0.31 $\pm$ 0.01             | 0.14 $\pm$ 0.01             | 0.25 $\pm$ 0.00             | 665.01 $\pm$ 106.04         |
>
> **7. ``Have authors observed changes in the KD-tree across iterations? For example, how many of the K closest neighbours initially move outside the K neighbours set on average in subsequent iterations?''**
>
> We tracked the evolution of the K-D tree for $K=5$ with a single random seed on some of the data sets (Wine Quality red and white, BreastCancer, Ionosphere, TRANSCRIPT, Gottlieb, Diabetes and Cdataset). We report in the table below a few metrics of interest: Number of points (%), which is the number and percentage of samples which neighborhood changed between the first and final iterations of F3I where the early stopping criterion was fulfilled; Number of changes, which is the average number of neighbor points across samples that differ between the first and final iterations (if a point is no longer part of the neighborhood and another enters it, this counts as two separate changes, which is why that number can be greater than $K=5$); Number of samples, which is the number of samples in the corresponding data set; and finally, Stopping time, which is the time when the early stopping criterion was fulfilled. From these observations, we observe that when the k-d tree does not change, it usually leads to F3I terminating early. When the k-d tree changes, the changes are restricted to a small part of the samples, but likely resets the neighborhood of a point, as the number of changes are greater than K.

---

> > ### Author Response · Authors · 2025-12-18
> > **Update (3/3)**
> >
> > | Data set     | Number of points (\%) | Number of changes | Number of samples | Stopping time |
> > |--------------|-----------------------|-------------------|-------------------|---------------|
> > | BreastCancer | 6 (1.05\%)            | 9.00              | 569               | 4             |
> > | Cdataset     | 0 (0\%)               | 0                 | 663               | 1             |
> > | Diabetes     | 0 (0\%)               | 0                 | 442               | 1             |
> > | Gottlieb     | 0 (0\%)               | 0                 | 593               | 1             |
> > | Ionosphere   | 0 (0\%)               | 0                 | 351               | 1             |
> > | TRANSCRIPT   | 59 (9.62\%)           | 3.08              | 613               | 3             |
> > | Wine (Red)   | 68 (4.25\%)           | 7.21              | 1,599             | 4             |
> > | Wine (White) | 0 (0\%)               | 0                 | 4,898             | 2             |

---

> ### Comment · Reviewer_bozn · 2025-12-18
> **Further questions on Rebuttal**
>
> 2. 1) I believe the text should clarify why certain previously used datasets were not considered in this study.
>    2) Interestingly, on the newly added datasets, the F3I method does not appear to be competitive.
>       Could the authors elaborate on this observation?
>
> 3.  1) The authors should add kNN method with uniform weights to Tables 2 and 3.
>     2) The new results indicate that F3I (and F3I-MLP) and kNN have nearly the same performance, with two notable exceptions (the Heart Disease and Breast Cancer datasets, where F3I is significantly superior). Given that k-NN methods are an order of magnitude faster, what is the motivation for choosing F3I over standard k-NN?
>     3) To improve clarity, I recommend grouping the k-NN and F3I results in the tables, as their current separation hinders direct comparison. Furthermore, these performance differences warrant a substantive discussion, potentially in a dedicated section of the manuscript.
>
> 8)  1) I am confused by the numbers provided in the table. My understanding is that the final MSE values should approximately equal the square of the mean values in Table 1. For instance, for the BreastCancer dataset, I expect an MSE around $(0.08)^{2} = 0.0064$, but the table reports $8,321.185$. Could you clarify this discrepancy?
>     2) The improvements across iterations appear quite minor. Does this indicate that most of the error is corrected during the first iteration?
>     3) I recommend adding the error at iteration 0 to the table to establish a baseline.
>     4) I also propose to add a joint table with corresponding values of the objetive $ G $.
>
> 10. I disagree with the authors' claim regarding reproducibility issues.
>     1) I was able to reproduce the HyperImpute results (RMSE and WD) on the Airfoil and Diabetes datasets in approximately 10 minutes.
>     2) The authors state that random seeds were not reported. However, a brief review of the HyperImpute code reveals that the random seed is explicitly set during the benchmarking process.
>     3) Furthermore, the explanation attributing the drastic performance degradation to the random seed seems implausible. If this were the primary cause, the authors would have observed high variance in their own results (Tables 1, 2, and 3), which is not the case.
>
>     I request that the authors provide a clear explanation for the performance degradation observed in the NewImp and HyperImpute methods.

---

> ### Author Response · Authors · 2025-12-19
> **Rebuttal (1/2)**
>
> Thank you for engaging with our rebuttal. We partially updated our manuscript according to your requests, and will update it again in the upcoming days before the deadline of December 21st, so that to let you answer to our comments if needed.
>
> **``2. I believe the text should clarify why certain previously used datasets were not considered in this study. Interestingly, on the newly added datasets, the F3I method does not appear to be competitive. Could the authors elaborate on this observation?''**
>
> We added our previous comment about categorical data in our revision, in the experimental study in Appendix. As observed in Table 1 in our manuscript, there are data sets where HyperImpute outperforms F3I, albeit the difference in performance is not as striking as when F3I outperforms HyperImpute (see the percentages computed in Table 1: in the results on the newly added data sets, the percentages of decrease in performance are respectively 20%, 37.5%, 30% and 75% in the Spambase, Wine red and white, and Libras data sets). We never claimed that F3I outperformed HyperImpute: the point of our experiments is that F3I trades for some decrease in imputation performance significant computational acceleration, theoretical guarantees and an increased performance in the combination of imputation with any downstream task. We note that F3I is still on par with the state-of-the-art, as it outperforms most of the recent baselines in imputation.
>
> **``3. The authors should add kNN method with uniform weights to Tables 2 and 3. The new results indicate that F3I (and F3I-MLP) and kNN have nearly the same performance, with two notable exceptions (the Heart Disease and Breast Cancer datasets, where F3I is significantly superior). Given that k-NN methods are an order of magnitude faster, what is the motivation for choosing F3I over standard k-NN? To improve clarity, I recommend grouping the k-NN and F3I results in the tables, as their current separation hinders direct comparison. Furthermore, these performance differences warrant a substantive discussion, potentially in a dedicated section of the manuscript.''**
>
> Due to a lack of space in the manuscript, we added the corresponding results for KNN with uniform weights in Appendix. As expected, KNN with uniform weights has the same or a worst performance than KNN with distance-dependent weights. We do not understand how F3I and KNN compare to F3I-MLP, as the first two algorithms are imputation-only and their performance is mainly measured with (R)MSE, whereas F3I-MLP is for the joint classification-imputation task, and is an ablated variant of PCGradF3I+MLP (our original contribution for joint classification-imputation tasks) which performance is measured with AUC. The advantage of F3I compared to KNN is (1) the theoretical analysis on Gaussian distributions, which can be extended to other distributions (see our rebuttal), and (2) the ability to adapt to the imputation depending on a downstream task (our contribution PCGradF3I), where it shows a clear improvement in performance for joint classification-imputation, and an on-par or vastly better performance on imputation-only tasks than KNN. Conceptually, it is also an answer to setting the weights of a KNN algorithm: previously, only the uniform weights or weights inversely proportional to the distance to neighbors were considered.
>
> **``8. I am confused by the numbers provided in the table. My understanding is that the final MSE values should approximately equal the square of the mean values in Table 1. For instance, for the BreastCancer dataset, I expect an MSE around $0.08^2 = 0.0064$, but the table reports $8,321.185$. Could you clarify this discrepancy? The improvements across iterations appear quite minor. Does this indicate that most of the error is corrected during the first iteration? I recommend adding the error at iteration 0 to the table to establish a baseline. I also propose to add a joint table with corresponding values of the objective $G$.''**
>
> Note that we did not use the HyperImpute framework for this experiment, so there might be discrepancies in the implementation of the evaluation procedure, because this framework does not allow us easily to store the intermediate MSE values for F3I. We used the test at the end of file *F3I.py*. Note that initial MSEs (at iteration 0) are exactly the ones obtained by KNN with uniform weights, as F3I uses this algorithm to derive the first guesses of imputed values. We will add in the upcoming days a table with values of the objective function $G$ for the table in (8).

---

> > ### Author Response · Authors · 2025-12-19
> > **Rebuttal (2/2)**
> >
> > **``10. I disagree with the authors' claim regarding reproducibility issues. I was able to reproduce the HyperImpute results (RMSE and WD) on the Airfoil and Diabetes datasets in approximately 10 minutes. The authors state that random seeds were not reported. However, a brief review of the HyperImpute code reveals that the random seed is explicitly set during the benchmarking process. Furthermore, the explanation attributing the drastic performance degradation to the random seed seems implausible. If this were the primary cause, the authors would have observed high variance in their own results (Tables 1, 2, and 3), which is not the case. I request that the authors provide a clear explanation for the performance degradation observed in the NewImp and HyperImpute methods.''**
> >
> > As previously mentioned and shown in our code in supplementary material, we used code from HyperImpute and the original implementation of NewImp (files *benchmarksHI.py* and *HyperImputeBenchmark.py*), and reported the used hyperparameter values in the experimental study in Appendix. The random seeds in HyperImpute were not reported in the paper, but we will use the exact same seeds as in their code in the upcoming days to try and reproduce their results, and investigate the difference in performance.

---

> ### Author Response · Authors · 2025-12-20
> **Update (1/3)**
>
> **``8. I am confused by the numbers provided in the table [...]''**
>
> We modified the HyperImpute code to obtain intermediate MSEs in F3I (MNAR, 30\%, $\eta=0.01$). Note that we ran the experiment on all 10 random seeds in Table 1, and showed the longest series of iterations (other series stop at the first iteration), so there is a little discrepancy between the final RMSE reported in the table below and the average value over 10 iterations. F3I returns the imputed values corresponding to the last time when G has a positive value.
>
> | Data set     | Metric/Iteration | 1        | 2                | 3                 | 4 |
> |--------------|------------------|----------|------------------|-------------------|---|
> | BreastCancer | RMSE             | 0.083741 | 0.043487         | -                 | - |
> |              | G                | N/A      | -1.429.10$^{-8}$ | -                 | - |
> | Diabetes     | RMSE             | 0.349037 | 0.352460         | -                 | - |
> |              | G                | N/A      | -7.705.10$^{-7}$ |                   | - |
> | Gottlieb     | RMSE             | 0.037501 | 0.038418         | 0.038418          | - |
> |              | G                | N/A      | 1.031.10$^{-10}$ | 0                 | - |
> | HeartDisease | RMSE             | 0.166427 | 0.156262         | 0.156266          | - |
> |              | G                | N/A      | 6.324.10$^{-7}$  | -1.302.10$^{-11}$ | - |
> | Ionosphere   | RMSE             | 0.183365 | 0.223894         | -                 | - |
> |              | G                | N/A      | -1.828.10$^{-7}$ | -                 | - |

---

> ### Author Response · Authors · 2025-12-20
> **Update (2/3)**
>
> **``10. I disagree with the authors' claim regarding reproducibility issues [...]''**
>
> We made a mistake when comparing our results to the main table in HyperImpute, so we modify our answer to this question (knowing that the previous version can be read when looking at the revisions of this comment). We compare our results to Figure 8b in HyperImpute, and note that across all common data sets and metrics, the only one with a discrepancy is the Diabetes data set, on the Wasserstein distance (WD) metric. Investigating our code, we found that we actually used the transpose of the feature matrix in this dataset (see file *HyperImputeBenchmark.py*), explaining why the RMSE was not impacted (as it is symmetrical) but the WD was. Since no other dataset was impacted by such a discrepancy and by checking our code for importing other datasets, this error seems restricted to the Diabetes dataset. We report the new results below for HyperImpute and our algorithm by running our file *HyperImputeBenchmark.py* in the MAR setting at 30\% missingness, and will modify the results for the Diabetes data set throughout the manuscript, hopefully by the deadline.
>
> | Data set | Alg.                | RMSE $\downarrow$  | MAE $\downarrow$   | WD $\downarrow$    | Runtime $\downarrow$ |
> |----------|---------------------|--------------------|--------------------|--------------------|----------------------|
> | Diabetes | **F3I (ours)** | 0.2272 $\pm$0.0552 | 0.1549 $\pm$0.0288 | 0.1096 $\pm$0.0251 | 0.5058 $\pm$0.2571   |
> |          | HyperImpute         | 0.2063 $\pm$0.0572 | 0.1182 $\pm$0.0187 | 0.0746 $\pm$0.0136 | 33.7499 $\pm$14.2511 |
>
> To ensure that we indeed obtained the same results as in HyperImpute, we ran the *experiments_02_model_performance.ipynb* notebook from the HyperImpute GitHub (function *compare_models* on the imported Diabetes data set) and obtained the following results (only on RMSE and WD, since we had to modify the HyperImpute framework to report the MAE and runtime):
>
> | Data set | Alg.        | RMSE $\downarrow$  | MAE $\downarrow$   |
> |----------|-------------|--------------------|--------------------|
> | Diabetes | HyperImpute | 0.2034 $\pm$0.0523 | 0.0704 $\pm$0.0074 |
>
> Up to the third decimal place, those results are equal. For NewImp, looking at their code (https://github.com/JustusvLiebig/NewImp/blob/main/exper\_wgf.py), the benchmark seems to only run a single random seed iteration (with random seed equal to 4 by default), and does not provide the 9 other random seeds.
>
> Please us know if this new rebuttal answers all of your concerns. Thank you for engaging with us and improving our manuscript.

---

> > ### Author Response · Authors · 2025-12-20
> > **Update (3/3)**
> >
> > We have updated the manuscript with the new results for Diabetes in the MNAR setting at 30\% missingness in Table 1.

---

> > ### Comment · Reviewer_bozn · 2025-12-20
> >
> > 2) The F3I method does not demonstrate superiority over the KNN baselines, despite being significantly slower.
> >
> > 3) 1) The F3I method is primarily a modification of uniform-weight KNN. Consequently, one would expect it to demonstrate consistently better performance, but this is not the case. For example, on RMSE:
> >     Uniform KNN outperforms F3I by approximately 20% on the Spambase, Wine (White), and Wine (Red) datasets.
> >     Performance is nearly identical on the Diabetes, Gottlieb, and Ionosphere datasets.
> >     F3I outperforms KNN on the Libras (≈10%) and Ionosphere (≈5%) datasets.
> >     KNN performs significantly worse on the Diabetes (≈262%) and Heart Disease (≈185%) datasets.
> >    2) The F3IMLP variant can be directly compared to KNN-MLP. As with most single-imputation tasks in the experiments, their performance is nearly identical.
> >    3) The standard KNN algorithm also has theoretical bounds on error (for example, [1, 2]), so I do not understand how the theoretical analysis of F3I on Gaussian distributions can be considered an advantage over KNN.
> >    4) From my point of view, the only advantage of the F3I method over KNN is that it can be trained jointly with a classifier on a classification-imputation task.
> >    5) I still think a dedicated paragraph or section discussing KNN and F3I performance is needed.
> >
> > 8. 1) I do not understand how the updated table relates to the one in the previous answer. Could conceptually minor differences in the implementation really lead to a 10,000-fold difference in RMSE results?
> >    2) I still believe metrics for the 0th iteration should be included in the results.
> >    3) Why is the value for G after the first iteration listed as N/A?
> >    4) The optimization of G does not appear to lead to improvements in RMSE; it often even increases the error. Could the authors comment on this? Might this behavior be related to overfitting, given that the early stopping criterion is computed on the training data? I suggest the authors try a train-validation strategy.
> >
> > 10) I believe there are other issues within the benchmarking process beyond the authors' statement that only a single metric (WD) on a single dataset (Diabetes) for a single method (HyperImpute) was affected.
> >     1) In the current revision, the reported RMSE metric is 0.34, but in the comments, the authors reported a value of 0.20 after a fix that should not have affected RMSE. Does this mean other changes were made?
> >     2) I obtained results of 0.0534 ± 0.006 RMSE (≈420% difference from Table 1) and 0.0383 ± 0.0034 WD (≈630% difference from Table 1) for the HyperImpute method on the BreastCancer dataset.
> >     I did this by simply changing the input dataset in the benchmarking code (https://colab.research.google.com/drive/1zGm4VeXsJ-0x6A5_icnknE7mbJ0knUig?usp=sharing).
> >     This discrepancy makes me skeptical about the validity of the paper's results.
> >     3) I still do not understand why NewImp shows much lower performance than reported in the original paper.
> >     4) The same question applies to the MIRACLE method: the MIRACLE paper reports that it is consistently better than KNN, but the authors' results indicate that MIRACLE is consistently and significantly worse than KNN.
> >
> > [1] T. Cover and P. Hart, "Nearest neighbor pattern classification," in IEEE Transactions on Information Theory, vol. 13, no. 1, pp. 21-27, January 1967, doi: 10.1109/TIT.1967.1053964.
> > keywords: {Bayes methods;Posterior probability;Convergence;Auditory displays;Random variables;Loss measurement;Extraterrestrial measurements;Density measurement;Accuracy;Visualization}.
> >
> > [2] Bax, X. (2019). Speculate-correct error bounds for k-nearest neighbor classifiers. Machine Learning, 108(12), 2087-2111.

---

> ### Author Response · Authors · 2025-12-20
> **Rebuttal**
>
> Thank you for your swift reply.
>
> 2. We agree with points 1-2, and note that, similarly, no clear superiority can be shown between uniform-weights and distance-dependent KNN, and that, moreover, our method can show large percentages of improvement on some datasets (as you remarked). As a consequence, our method is another answer to the setting of weights in KNN algorithms. As for point 3, those bounds are for classification (that is, on the probability of error in assigning a label to a point) and not for regression (~ imputation). As such, we maintain that our bounds are novel, and can indeed be used to infer bounds for the uniform-weights KNN algorithm. We agree with point 4. We will write a paragraph by the deadline on the comparative performance between KNN and F3I.
>
> 8. Those are not minor differences in the implementation. Our implementation of the MNAR setting are different, as we used the Gaussian self-masking defined in Le Morvan et al, whereas in HyperImpute, they use a logistic masking model. As for point 2, we will add metrics for iteration 0 in Table 1. G is not defined at iteration 0 because its definition relies on a prior imputation of the points, and otherwise its value is trivial (comparing the same uniform-weights KNN imputation). The increase of the MSE is often linked to a negative value of G, which is why we implemented the early stopping criterion. This early stopping criterion could be refined to ensure the increase of the MSE.
>
> 10. The current revision for Diabetes in Table 1 for HyperImpute is 0.23. No other changes beyond the transposition of the feature matrix at line 539 of file *HyperImputeBenchmark.py* were made. We note that the missingness mechanism in the previous comment was MAR, to match the setting of the main table in the HyperImpute paper. We will investigate points 2-4, and come back with an update.

---

> ### Author Response · Authors · 2025-12-20
> **Update (1/2)**
>
> 2. We added to the revision the paragraph on the comparative performance between KNN and F3I in section H.1.2.
>
> 3. The metrics obtained for iteration 0 are the same as those with the KNN with uniform weights...
>
> 4. With regards to point 4, here are the results on RMSE obtained on the notebook in HyperImpute for MIRACLE and KNN with uniform weights (MNAR, 30%, 10 random seeds) which are compatible with our results in Table 1 (including some variability induced by different random seeds).
>
> Breast cancer: KNN unif: 0.0801 +/- 0.0024, MIRACLE: 3.9074 +/- 0.2494
>
> Diabetes: KNN unif: 0.2143 +/- 0.0207, MIRACLE: 4.0993 +/- 0.0932
>
> Wine Red:  KNN unif: 0.1084 +/- 0.0041, MIRACLE: 0.2097 +/- 0.007
>
> Wine White: KNN unif: 0.0896 +/- 0.0038, MIRACLE: 0.1138 +/- 0.031
>
> Moreover, the values of MIRACLE are compatible with those reported in the NewImp paper.

---

> > ### Author Response · Authors · 2025-12-20
> > **Update (2/2)**
> >
> > 4. As for points 2-3, given that we needed a few days to investigate the Diabetes data set, it seems that we won't make it in time for December 21st, 2025. We will withdraw the paper, check one by one every dataset/method/setting, and resubmit it later. In the meantime, thank you very much for the thorough review and the productive exchange, which helped improving our manuscript.

---

### Review · Reviewer_yHSf · 2025-11-28

**Summary Of Contributions:**

This paper proposes an imputation method called F3I, designed to improve imputation performance based on iterative refinements of a K-nearest neighbor imputation approach. F3I can also be jointly trained with a downstream classifier of any architecture.
The authors also derive upper bounds for imputation under MCAR, MAR, and MNAR missingness mechanisms, and conduct extensive experiments to evaluate the performance of F3I and validate the theoretical claims.

**Additional Comments:**

I am not very familiar with the research field of data imputation. I have tried to understand the techniques and go through the experiments, and wrote my comments, though my comments may not be professional. I hope the AE can place greater weight on the other reviewers' comments when making a judgment.

**Audience:**

Yes

**Audience Explanation:**

1. Missing values are a common scenario in datasets, so the authors address a very important problem in machine learning.

2. The paper appears comprehensive and concise. Practitioners seeking to improve model performance on biased datasets (datasets with many missing values) may find this paper relevant.

**Broader Impact Concerns:**

This paper uses public datasets for the experiments, so I believe there are no ethical concerns associated with this work.

**Claims And Evidence:**

Yes

**Claims Explanation:**

1. The authors derive rigorous theoretical bounds for their proposed approach, and the experiments convincingly demonstrate that F3I outperforms other baseline methods in most cases.

2. The authors propose a joint imputation-classification framework that simultaneously optimizes both imputation performance and downstream model performance, offering strong practical utility.

3. The experiments are thorough, taking into account both synthetic missing values and real-world datasets with naturally occurring missing data.

**Requested Changes:**

My main concerns are related to the experimental design and presentation:
- F3I is a KNN-based imputation method. However, the authors do not appear to compare it against other KNN-based imputation approaches mentioned in the Related Work section.
- In Table 2, HyperImpute shows stronger overall performance than F3I, while F3I performs best on other datasets. Could the authors elaborate on this observation?
- Regarding the joint training approach in Section 5, it would be helpful to see an ablation study where the model is trained on the imputed dataset without joint training. How would the performance compare in that scenario?
- It would also be valuable to evaluate F3I on more challenging datasets (such as CIFAR-10) or with more complex model architectures (like DNNs or CNNs instead of MLPs) to better assess its scalability and robustness.

Minor suggestion:  Moving the main results (Table 1 to Table 4) before the Limitations section would improve the readability of the paper.

---

> ### Author Response · Authors · 2025-12-09
> **Rebuttal**
>
> We thank the reviewer for taking the time to review our paper, and for their positive feedback about our theoretical work and experimental study. We updated a partial revision of the manuscript, and will upload the final version in the upcoming days. In the final version, we will add new experiments regarding more complex real-world data sets and classifier architectures.
>
> **1. ``F3I is a KNN-based imputation method. However, the authors do not appear to compare it against other KNN-based imputation approaches mentioned in the Related Work section.''**
>
> See the answer to **Question 3** to Reviewer bozn. We added experiments on the KNN algorithms with uniform and distance-dependent (that is, inversely proportional to the distance to the neighbor) weights to Tables 1-3 in the main text.
>
> **2. ``In Table 2, HyperImpute shows stronger overall performance than F3I, while F3I performs best on other datasets. Could the authors elaborate on this observation?''**
>
> As illustrated in Table 1, HyperImpute might be on top, depending on the data set. However, note that the relative increase in performance (expressed in \% in Table 1) of HyperImpute compared to F3I is much lower--with metrics almost similar up to $10^{-2}$--than the increase in performance of F3I compared to HyperImpute on other data sets. This is confirmed in Tables 2 and 3, where HyperImpute is on top in Table 2 (Ionosphere data set), but often not significantly better than F3I, whereas F3I cleraly outperforms HyperImpute in Table 3 (BreastCancer data set).
>
> **3. ``Regarding the joint training approach in Section 5, it would be helpful to see an ablation study where the model is trained on the imputed dataset without joint training. How would the performance compare in that scenario?''**
>
> We create a method F3IMLP, which sequentially and separately applies F3I for imputation, and then the MLP for prediction. We compare with the results shown in Table 4 for PCGradF3I (the joint model) in the manuscript. As expected, the performance is lesser (but still high for some of the data sets) than the joint training approach.
>
> | Task     | Alg./Data set             | BreastCancer   | Ionosphere     | MNIST          | PREDICT        |
> |----------|---------------------------|----------------|----------------|----------------|----------------|
> | Joint    |**PCGradF3I (ours)** | 0.70 $\pm$0.14 | 0.77 $\pm$0.14 | 0.99 $\pm$0.09 | 0.51 $\pm$0.01 |
> | Separate | F3IMLP                    | 0.53 $\pm$0.13 | 0.70 $\pm$0.19 | 0.96 $\pm$0.13 | 0.49 $\pm$0.06        |
>
> **4. ``It would also be valuable to evaluate F3I on more challenging datasets (such as CIFAR-10) or with more complex model architectures (like DNNs or CNNs instead of MLPs) to better assess its scalability and robustness.''**
>
> We will perform this experiment in the upcoming days, and comment again once the experiment concludes.

---

> > ### Author Response · Authors · 2025-12-18
> > **Update**
> >
> > We updated the latest version of the revision, and we provide below the results for the experiments on the more complex model and dataset.
> >
> > **4. ``It would also be valuable to evaluate F3I on more challenging datasets (such as CIFAR-10) or with more complex model architectures (like DNNs or CNNs instead of MLPs) to better assess its scalability and robustness.''**
> >
> > Due to the time constraints on the rebuttal, we focused on Imagenette (available on GitHub at fastai/imagenette) which is a subset of ImageNet and which is smaller than CIFAR-10, with two of the 10 available labels, n01440764 and n02102040. After resizing each image to 32x32, we applied a MCAR missingness procedure with probability $50\%$ on each pixel (meaning that, if a pixel is missing, all RGB channels have missing values). We considered as classifier the following CNN architecture: 3 iterations of two convolutional layers and ReLU activations followed by a Max-Pool layer, and then three linear layers with ReLU activations, where the final number of features is 2 (the number of classes). We aimed at minimizing the cross-enthropy loss. We ran the joint imputation-classification experiments 10 times on some of the fastest methods in Table 4 in the manuscript. We reported the AUC metric in the table below. PCGradF3I still performs better than NeuMiss, but slightly worse than KNN imputation. By looking at the trend in AUC, we notice that PCGradF3I overfits at 10 epochs (meaning that the training loss still decreases whereas the validation loss increases), so we assume that for images, we might need to develop a more stringent criterion for early stopping than the one currently implemented in F3I.
> >
> > *Average and standard deviation of Area Under the Curve (AUC) and Accuracy (ACC) values (rounded to the closest second decimal place) across 10 runs on the joint imputation-classification task on Imagenette-2-labels (MCAR scenario, $p^\text{miss}=50\%$). Hyperparameter values: $K=20$, $T=10$, 10 epochs, batch size: 128, validation set size: 10\%, learning rate: 0.001. We compare the AUCs with the AUC value obtained when the classifier has access to the complete data without missing values.*
> > | Metric | (no missing values) | KNN                     | PCGradF3I                  | PCGradF3I      | PCGradF3I      | NeuMiss       |
> > |--------|---------------------|-------------------------|----------------------------|----------------|----------------|---------------|
> > |        |                     |                         | ($\beta=0$)                | ($\beta=0.5$)  | ($\beta=1$)    |               |
> > | AUC    | 0.79 $\pm$0.10      | **0.65 $\pm$0.16** | *0.63 $\pm$0.15* | 0.61 $\pm$0.13 | 0.62 $\pm$0.12 | 0.51$\pm$0.01 |

---

### Review · Reviewer_gu2i · 2025-12-07

**Summary Of Contributions:**

In this work, the authors present a novel method for missing data imputation entitled F3I (Fast Iterative Improvement for Imputation). The crux of the method is to iteratively improve the weights for a linear combination of $K$ data samples from the reference (non-missing) sample obtained via a $K$\-nearest-neighbor search. The weights are determined by applying the online learning algorithm AdaHedge with a modified objective function that seeks to maximize the likelihood of the data under a multivariate Gaussian kernel density fitted to a reference set of initial guesses of the imputed values. The weight vector is then iteratively updated until a maximum number of iterations or early stopping criterion (based on the density ratio) is reached. The article also includes a fairly extensive theoretical analysis of the algorithm proving high probability bounds on the imputation error. The main advantages of the authors’ approach are (i) it’s theoretical guarantees for distribution-preserving imputation subject to a number of assumptions about the data distribution and missingness mechanism, and (ii) it’s ability to achieve imputation quality competitive to other state-of-the-art algorithms at a much lower computational cost.

### Strengths

- The manuscript is of very high quality and was clearly prepared with great care. The theoretical analysis and empirical results are exceptionally thorough and meticulously presented. The writing is also generally very clear.

- Although the proposed method does not consistently achieve better imputation performance than the baselines, it does appear to be far more computationally efficient in most cases.

- The core concept of the method is quite straightforward and elegant.

- The theoretical bounds on imputation performance are novel and potentially useful for some application domains.

- The authors also provide empirical verification of the theoretical bounds which is a nice touch.

- The code is relatively well documented and nicely organized.


### Weaknesses

- The results of the empirical analysis are somewhat mixed. The proposed method (as well as many of the other baselines) are outperformed by mean imputation in the MCAR and MAR experiments, which is a bit surprising. The authors do address this in the Appendix by pointing out that the KNN-imputers implicitly assume a kind of MNAR mechanism.

- Related to the above point, the fact that the proposed method is consistently outperformed by HyperImpute in Table 2 deserves more discussion.

- The theoretical analysis, while comprehensive, is based on some pretty generous assumptions (e.g. Gaussianity of the data distribution) that likely will not hold in many (if not most) practical contexts.

- The method relies on fitting a nonparametric kernel density to the reference (imputed) dataset which may not scale well to large, high-dimensional datasets, and is of course very sensitive to the choice of kernel density estimator and its hyperparameters.

**Audience:**

Yes

**Audience Explanation:**

Imputation of missing data is a fundamental and widespread problem in many application domains and thus should be of interest to many practitioners in the audience of TMLR. This manuscript clearly falls within the scope of the journal.

**Broader Impact Concerns:**

I don’t see any issues with broader impact that are specific to this study rather than the practice of data imputation more generally. Of course there is always the general problem with imputation that it may create privacy concerns when applied to anonymized datasets.

**Claims And Evidence:**

Yes

**Claims Explanation:**

The manuscript is overall very meticulously prepared and carefully worded, I did not spot any major technical errors or inconsistent claims.

I read through the proofs and assessed them for sensibility, but I am not qualified to do a thorough assessment. Hopefully one of the other reviewers will be able to check them in more detail.

I was able to download and read through the code and did not spot any obvious issues. Unfortunately, I have not yet managed to actually run it due to issues with installing the dependencies. In particular, there seems to be an implicit dependency on Cython that is not mentioned in the README.

The experimental design is sound and there are no major inconsistencies or errors in the presentation of the empirical results. I only have a few minor points to note here:

1. Some of the box plots have scaling issues on the y-axis; i.e. it’s very hard to identify differences between most of the methods because one of the methods has error values that are an order of magnitude higher (see e.g. Figs 11-20). The authors should consider either using a log-scale or excluding these outliers from the plots.

2. In section H.2.2, it is not mentioned whether or how the hyperparameters of the baselines were optimized. I would argue that it’s an error to perform hyperparameter optimization on only one method (the proposed one) and not the baselines.

3. Figs 11-24 are not referenced in the text, and in some cases they probably should be. For example, in Figs. 11-19, the mean consistently outperforms the other methods, even on some of the MNAR tasks. This is only briefly alluded to in section H.1.4.

4. The authors also do not provide much detail in the text on how the kernel density is estimated. In the code, it appears to use the built-in density estimator provided by the scikit-learn KDTree. I think this could be better documented and discussed in the paper.

**Requested Changes:**

I only have minor suggested changes. They can all be considered optional and can be incorporated at the discretion of the authors.

### Section 2

- “untractable” (not a word) -> “intractable”

- “preserves the data distribution”
    The distribution of the observed data? Or the (unknown) true distribution implied by the data generating mechanism? Some clarification would be helpful here.


### Section 3.1

- “which are in essence unavailable”
    They are not “in essence” unavailable, they are just unavailable. Or perhaps “typically unavailable”.

- “First, we assume that each value in the full data matrix is drawn from independent fixed-variance Gaussian distributions (Assumption B.1).”
    Please move this sentence to the start of the following paragraph.

- “If mf i = 1, then the **coefficient** at position (i, f )…”
    I think you mean **covariate** here, unless I am misunderstanding something.

- “drawn from Bernoulli law with fixed mean…” -> “drawn from *a Bernoulli distribution* with fixed mean…”
    You can’t sample from a law.


### Proof of theorem 4.4

Please consider adding a half-sentence reminder to the reader as to what is meant by the “gradient trick”.

### Section 6.1

The results in Table 2 show that HyperImpute consistently outperforms F3I. This should be discussed in the text.

### Appendix A

The section on out-of-sample imputation is quite important and should be moved to the main text, possibly in the limitations or discussion section.

### Section H.2

There is insufficient detail given about the types of data in each dataset; e.g. what they represent, how they relate to each other, and how well they satisfy the distributional assumptions of the proposed method.

### General remarks

- I would suggest replacing “real-life data” with “real-world data” throughout the text as it is more appropriate in this context.

---

> ### Author Response · Authors · 2025-12-09
> **Rebuttal**
>
> Thank you very much for the time spent in reviewing our manuscript and the kind comments about our theoretical and empirical results. We updated a partial revision of the manuscript, and will upload the final version in the upcoming days. In particular, we will address the issue with the box plots and the description of the real-world data sets in the future revision.
>
> **1. ``Related to the above point, the fact that the proposed method is consistently outperformed by HyperImpute in Table 2 deserves more discussion. [...] The results in Table 2 show that HyperImpute consistently outperforms F3I. This should be discussed in the text.''**
>
> Please refer to our answer to \textbf{Question 2} to Reviewer yHSf.
>
> **2. ``The theoretical analysis, while comprehensive, is based on some pretty generous assumptions (e.g. Gaussianity of the data distribution) that likely will not hold in many (if not most) practical contexts.''**
>
> As mentioned in the paragraphs on limitations of the manuscript, we acknowledge the restriction of the theoretical results regarding data distribution. However, we think that those results nonetheless remain of interest, as the structure of the proof is still valid as long as an equivalent of the Corollary G.6. is found for the desired data distribution. Moreover, Gaussian distributions are the most common in the literature on statistics, and as such, finding upper bounds for this type of distribution remains valuable.
>
> **3. ``The method relies on fitting a nonparametric kernel density to the reference (imputed) dataset which may not scale well to large, high-dimensional datasets, and is of course very sensitive to the choice of kernel density estimator and its hyperparameters.''**
>
> We acknowledge the restriction regarding the choice of the kernel density estimator, and we added this information to the paragraph on limitations in the current revision.
>
> **4. ``Unfortunately, I have not yet managed to actually run it due to issues with installing the dependencies. In particular, there seems to be an implicit dependency on Cython that is not mentioned in the README.''**
>
> Please provide us the exact error and the missing dependency. We will add it to the README file.
>
> **5. ``In section H.2.2, it is not mentioned whether or how the hyperparameters of the baselines were optimized. I would argue that it’s an error to perform hyperparameter optimization on only one method (the proposed one) and not the baselines.''**
>
> All baselines had their hyperparameters finetuned using Optuna, as we did for F3I (see file *hyperparamOpt\_NeuMiss.py* for instance in the code). We added this information to Appendix H in the revision.
>
> **6. ``The authors also do not provide much detail in the text on how the kernel density is estimated. In the code, it appears to use the built-in density estimator provided by the scikit-learn KDTree. I think this could be better documented and discussed in the paper.''**
>
> We added a paragraph to the manuscript in Appendix A. Thank you!

---

> > ### Author Response · Authors · 2025-12-18
> > **Update**
> >
> > We updated the latest version of the revision.

---

### Review · Reviewer_TdJw · 2025-12-10

**Summary Of Contributions:**

The paper introduces F3I, an imputation method that iteratively refines K-Nearest Neighbor (KNN) weights using an online learner (AdaHedge) to maximize a distribution-preserving objective function estimated via Kernel Density Estimation (KDE). The authors extend this approach with PCGrad-F3I for joint training with downstream classification tasks and derive theoretical upper bounds on imputation quality and regret under MCAR, MAR, and Gaussian self-masking MNAR settings. Empirical evaluations on synthetic and real-world datasets, including drug repurposing and MNIST, demonstrate that F3I achieves accuracy comparable to state-of-the-art deep generative models while requiring significantly less computation time.

**Audience:**

Yes

**Audience Explanation:**

Handling Missing Not At Random (MNAR) data is a massive headache for everyone, and seeing a method that is computationally cheap compared to diffusion or GAN models is really appealing. The mix of classical KNN imputation with online learning theory is a fresh angle. It’s a solid contribution for the community.

**Broader Impact Concerns:**

No concern

**Claims And Evidence:**

Yes

**Claims Explanation:**

The claims are mostly supported. I'm overall very positive about the paper, except for a few weaknesses that I noticed:
- The theoretical side relies heavily on the data being drawn from independent fixed-variance Gaussian distributions (Assumption B.1), which is a huge idealization for real-world data.
- Forcing a single weight vector $\alpha$ to be shared across every single sample feels like a strict constraint that ignores local geometry? Unless we assume the local geometry around all data points to be identical (which is extremely unlikely in moderate dimensions), it does not seem to make sense to me to have a single $\alpha$ weight. Have the author considered using per-sample weight (but in this case, the formulation (1) no longer makes sense)?
- I also have doubts about the methodology: you are optimizing against a static reference density $D_0$ that comes from a basic uniform imputation3. If that initial guess is biased, aren't we just optimizing towards a bad target? Have the authors considered evolving $D_0$ as well as optimization goes on?

**Requested Changes:**

Could the authors comment on the weaknesses that I mentioned above?

---

> ### Author Response · Authors · 2025-12-10
> **Rebuttal**
>
> Thank you for the time spent on reviewing our manuscript, and the positive comments regarding our contributions. We will try to address your concerns below.
>
> **1. ``The theoretical side relies heavily on the data being drawn from independent fixed-variance Gaussian distributions (Assumption B.1), which is a huge idealization for real-world data.''**
>
> Please refer to our answer to **Question 2** to Reviewer gu2i. The bottom line is that, albeit the current assumptions are strong, we believe that our proof strategy can be reused for any target distribution.
>
> **2. ``Forcing a single weight vector to be shared across every single sample feels like a strict constraint that ignores local geometry? Unless we assume the local geometry around all data points to be identical (which is extremely unlikely in moderate dimensions), it does not seem to make sense to me to have a single weight. Have the author considered using per-sample weight (but in this case, the formulation (1) no longer makes sense)?''**
>
> Please refer to our answer to **Question 13** to Reviewer bozn and our discussion in the revised manuscript. Sample-specific weights introduce an additional tractability challenge, that could perhaps be solved by clustering samples into a fixed number of groups, and then apply independently F3I to each cluster of samples. This is a direction that could be investigated in the future, in order to make theoretical results match this setting. However, note that, despite the strong constraint on shared weights, F3I still succeeds in being competitive in practice.
>
> **3. ``I also have doubts about the methodology: you are optimizing against a static reference density that comes from a basic uniform imputation. If that initial guess is biased, aren't we just optimizing towards a bad target? Have the authors considered evolving as well as optimization goes on?''**
>
> As discussed in page 6 of the manuscript and in our discussion, considering an evolving target distribution $D_t$ at iteration $t$ instead of $D_0$ would probably be beneficial both empirically and theoretically (removing a term linear in $t$). However, this would require extra steps for controlling the gap between $D_{t-1}$ and $D_t$, and might also be more computationally expensive, as we have to build again the k-d tree at the start of each iteration.

---

> > ### Author Response · Authors · 2025-12-18
> > **Update**
> >
> > We updated the latest version of the revision.

---

### Note · Authors · 2025-12-20

**Comment:**

We need time to review the results and their consistency with respect to prior works. We would like to thank all reviewers and the editor for taking the time to consider our manuscript and providing useful feedback. Happy end-of-year holidays.

**Withdrawal Confirmation:**

I have read and agree with the venue's withdrawal policy on behalf of myself and my co-authors.